# Uncertainty Estimation for Heterophilic Graphs Through the Lens of Information Theory

Dominik Fuchsgruber [* 1]   Tom Wollschläger [* 1]   Johannes Bordne [1]   Stephan Günnemann [1]

## Abstract

While uncertainty estimation for graphs recently gained traction, most methods rely on homophily and deteriorate in heterophilic settings. We address this by analyzing message passing neural networks from an information-theoretic perspective and developing a suitable analog to data processing inequality to quantify information throughout the model's layers. In contrast to non-graph domains, information about the node-level prediction target can *increase* with model depth if a node's features are semantically different from its neighbors. Therefore, on heterophilic graphs, the latent embeddings of an MPNN each provide different information about the data distribution – different from homophilic settings. This reveals that considering all node representations simultaneously is a key design principle for epistemic uncertainty estimation on graphs beyond homophily. We empirically confirm this with a simple post-hoc density estimator on the joint node embedding space that provides state-of-the-art uncertainty on heterophilic graphs. At the same time, it matches prior work on homophilic graphs without explicitly exploiting homophily through post-processing.

## 1. Introduction

Trusting a machine learning model is critical for real-world use, especially in high-risk application domains (Xu & Saleh, 2021; Rabe et al., 2021). While most uncertainty modeling focuses on computer vision models (Gawlikowski et al., 2023), recently, also interdependent data modalities

like graphs gained traction (Stadler et al., 2021; Fuchsgruber et al., 2024b; Trivedi et al., 2024; Wang et al., 2024). Many approaches assume homophily – that nodes preferably connect to nodes of the same class – either explicitly by using graph diffusion (Stadler et al., 2021; Fuchsgruber et al., 2024a; Wu et al., 2023) or implicitly through the chosen backbone model (Bazhenov et al., 2022).

In heterophilic settings, when nodes primarily connect to nodes of a different class, these methods and their uncertainty estimates deteriorate. One reason is that they often explicitly rely on smoothing to propagate (un)certainty within clusters in the graph. This can be beneficial when links exist predominantly between semantically similar nodes but it is less justified on heterophilic graphs.

In this work, we approach uncertainty estimation from an information-theoretic perspective. Taking inspiration from Tishby & Zaslavsky (2015), we study Message Passing Neural Networks (MPNNs) in terms of the information its latent node representations provide about the target variable. By formulating a suitable graphical model, we derive an analog to the Data Processing Inequality (DPI) that applies to settings with independent and identically distributed (i.i.d.) data. It enables us to propose a novel definition of homophily based on the semantically relevant information a node's neighbors provide that is different from its own. We prove that in homophilic graphs where the information of adjacent nodes is similar to a node's own features, the information gained through considering deeper layers diminishes. In heterophilic graphs, adjacent nodes exhibit different semantics that are captured by representations at different layers. This reveals a key principle for uncertainty quantification in MPNNs under heterophily: They should *jointly* consider all node representations.

We realize this design principle through **J**oint **L**atent **D**ensity **E**stimation (JLDE) on the embeddings in an MPNN using a simple KNN-based estimator. We are the first to study uncertainty estimation in node classification beyond homophilic graphs and systematically compare our principle to established estimators on model architectures that perform well on heterophilic data. It provides state-of-the-art epistemic uncertainty across different datasets, distribution shifts, and MPNN backbones, confirming the validity

---
*Equal contribution   [1]School of Computation, Information & Technology and Munich Data Science Institute, Technical University of Munich. Correspondence to: Dominik Fuchsgruber <d.fuchsgruber@tum.de>, Tom Wollschläger <t.wollschlaeger@tum.de>.

*Proceedings of the 42nd International Conference on Machine Learning*, Vancouver, Canada. PMLR 267, 2025. Copyright 2025 by the author(s).

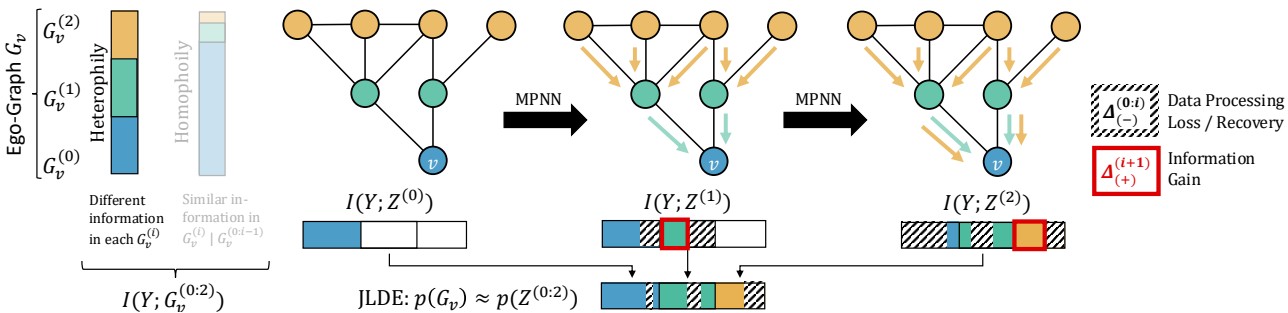

*Figure 1.* Information propagation in MPNNs. In the $i$-th iteration, information about $i$-hop neighbors of the anchor node $v$ is *gained* ($\Delta_{(+)}^{(i)}$) while information about neighbors at smaller distances may be lost to processing ($\Delta_{(-)}^{(0:i-1)}$). In heterophilic graphs, the $\mathcal{G}_v^{(i)}$ are semantically different and each latent representation $Z^{(i)}$ often contains different information about the target Y. Therefore, density-based uncertainty must be estimated jointly from all $Z^{(i)}$ to fully capture all available information about the data distribution.

and broad applicability of our findings. Our principle also matches the performance of other model-agnostic uncertainty estimates on homophilic graphs without explicitly facilitating homophily.[1]

## 2. Background

**Transductive Node Classification.** We define a graph $\mathcal{G} = (\mathcal{V}, \mathcal{E}, \mathcal{F})$ as a set of $n$ vertices $\mathcal{V}$ that are connected by $m$ undirected edges $\mathcal{E} \subseteq \{\{u, v\} | u, v \in \mathcal{V}\}$ represented with a symmetric adjacency matrix $\boldsymbol{A} \in \{0, 1\}^{n \times n}$. Each node is associated with features $\boldsymbol{F} \in \mathbb{R}^{n \times d}$. Nodes are assigned a label $\boldsymbol{y} \in \{1, \dots, c\}^n$ and the task is to infer these labels from a subset of labeled nodes $\mathcal{V}_{\text{train}}$.

**Mutual Information** (MI) measures the dependence between two random variables X, Y and is defined as:

$$I(X; Y) = \int p(X, Y) \log \frac{p(X, Y)}{p(X)p(Y)} dx dy. \quad (1)$$

It is symmetric $I(Y; X) = I(X; Y)$ and non-negative $I(X; Y) \geq 0$, with equality only if X and Y are independent. MI quantifies the difference between the joint distribution over two random variables is and the product distribution of their marginals. By conditioning on additional random variable(s) Z, $I(X; Y \mid Z)$ measures the information between X and Y that is not already in Z. We use MI to track the information of a MPNN's embeddings regarding the label by representing its inputs, target, and hidden representations as random variables similar to (Tishby & Zaslavsky, 2015).

**Homophily and Heterophily** describe how semantically similar adjacent nodes are. Many definitions measure similarity through class labels with some recent work also considering node features (Luan et al., 2022), see Appendix C.1. Instead, in Section 4.3, we define homophily as the semantically relevant mutual information between a node's features and the features of its neighbors.

---

[1]We provide our code at cs.cit.tum.de/daml/heterophilic-uncertainty/

**Ego Graphs.** For each node $v \in \mathcal{V}$, we can define an *ego-graph* $\mathcal{G}_v$ centered at this node. This ego-graph can be decomposed into disjoint subsets $\mathcal{G}_v = \sqcup_i \mathcal{G}_v^{(i)}$ where the ego-graph of $i$-th order $\mathcal{G}_v^{(i)}$ contains all nodes with a shortest-path distance of $i$-hops to the center node $v$. Consequently, $\mathcal{G}_v^{(0)}$ contains just the node $v$ and its features.

**Message Passing Neural Networks.** Most popular Graph Neural Networks (GNNs) can be formalized as MPNNs (Kipf & Welling, 2017; Hamilton et al., 2018; Veličković et al., 2018; Brody et al., 2022). They are closely related to the ego-graph perspective as they iteratively update node representations by first aggregating the representations of a node $v$'s neighbors $\mathcal{N}(v)$ and then combining them with their own representation $\boldsymbol{z}_v$.

$$\boldsymbol{a}_v^{(k)} = \text{AGGREGATE}^{(k)} \left( \{ \boldsymbol{z}_u^{(k-1)} | u \in \mathcal{N}(v) \} \right),$$
$$\boldsymbol{z}_v^{(k)} = \text{COMBINE}^{(k)} \left( \boldsymbol{z}_v^{k-1}, \boldsymbol{a}_v^{(k)} \right). \quad (2)$$

Therefore, the representation of a node $v$ after $i$ iterations $\boldsymbol{z}_v^{(i)}$ depends on the ego graphs up to $i$-th order $\mathcal{G}_v^{(0:i)}$.

**Uncertainty in Machine Learning** is often disentangled into aleatoric uncertainty $u^{\text{alea}}$ and epistemic uncertainty $u^{\text{epi}}$. The former arises from irreducible sources inherent to the data (e.g. measurement noise). The latter is rooted in a lack of knowledge and can be reduced, e.g. with additional data. It can be quantified as inverse to (an estimate of) the data distribution $p(X)^{-1}$, as a model should be confident near its training data (Qazaz, 1996).

## 3. Related Work

**Heterophilic Node Classification** is a sub-field of node classification specifically dealing with heterophilic graphs. While this problem has been studied exhaustively by previous work (see Appendix B), a recent benchmark on heterophilic node classification reveals that much of the success of GNNs dedicated to heterophilic graphs can be attributed

to hyperparameter tuning and issues regarding dataset leakage (Platonov et al., 2023). In particular, their systematic evaluation shows that many GNNs, like GCN (Kipf & Welling, 2017) or graph attention (Veličković et al., 2018; Brody et al., 2022; Mustafa & Burkholz, 2024) originally designed for homophilic settings, are highly effective and often outperform designated models in heterophilic classification problems. Importantly, Platonov et al. (2023) augment these homophilic GNNs to accommodate for heterophily: They introduce residual connections, increase the model complexity through deep feature transformations, and use LayerNorm (details in Appendix C.2). We build on these insights and verify JLDE by applying it to these models as they are the most effective in heterophilic settings.

**Uncertainty in i.i.d. Classification** can be estimated under different paradigms. A family of sampling-based Bayesian estimators uses an information-theoretic decomposition by approximating the posterior over the model weights (Hüllermeier & Waegeman, 2021) from which multiple samples are drawn during inference. Here, epistemic uncertainty is quantified as the deviation of individual samples from their expectation. Bayesian Neural Networks (BNNs) (Blundell et al., 2015; Dusenberry et al., 2020; Farquhar et al., 2020) explicitly model their weights as normal distributions to sample from the posterior. Other approaches are training an ensemble (Lakshminarayanan et al., 2017; Wen et al., 2020), Stochastic Centering (Thiagarajan et al., 2022), using Monte-Carlo dropout (MCD) Gal & Ghahramani (2016), or data augmentations during inference (Wang et al., 2019). Sampling-free estimators are typically deterministic and can be computed with a single forward pass. In evidential learning, epistemic uncertainty is estimated by predicting a second-order Dirichlet distribution (Sensoy et al., 2018; Malinin & Gales, 2018). Another method is to quantify epistemic uncertainty as distance from or similarity to the training data. Examples include Gaussian Processes (GPs) (Ng et al., 2018; Zhi et al., 2023) that measure similarity with a kernel function, or methods that directly estimate the data density in distance-preserving models (Liu et al., 2020a; Mukhoti et al., 2021a;b). Grathwohl et al. (2019) propose an energy-based interpretation of a classifier (EBM) that can be used for that purpose as well (Liu et al., 2020b). Posterior Networks (Charpentier et al., 2020) combine this with evidential learning by computing distance-based Bayesian evidence updates. Recently, an information-theoretic perspective on uncertainty estimation was proposed by Benkert et al. (2024). They argue that uncertainty estimation benefits from jointly considering all latent representations throughout a model to mitigate information loss. They introduce linear classification heads at each layer that are trained jointly with the backbone classifier which prevents post-hoc uncertainty estimation. In contrast, we derive an MPNN-based DPI to reveal the importance of jointly modeling all hid-

den representations especially in heterophilic problems, and confirm this insight with a simple post-hoc estimate.

**Uncertainty in Homophilic Node Classification** is a recently emerging field. Many approaches transfer concepts from i.d.d. problems to the graph domain: DropEdge applies MCD to the edges (Rong et al., 2020; Hasanzadeh et al., 2020), Graph Δ-UQ (Trivedi et al., 2024) uses Stochastic Centering in GNNs, and other methods propose GPs for graphs (Ng et al., 2018; Zhi et al., 2023; Liu et al., 2020c). Often, such approaches facilitate graph diffusion to improve uncertainty estimates. Graph Posterior Network (GPN) (Stadler et al., 2021) diffuses evidence using PageRank (Page, 1999), and graph EBMs like GNNSafe (Wu et al., 2023) or GEBM (Fuchsgruber et al., 2024b) employ label propagation. These diffusion kernels exploit the homophily in the graph, which empirically transfers well to uncertainty estimates. Despite the popularity of non-homophilic classification in the literature, to the best of our knowledge, there is no work on uncertainty quantification that explicitly targets heterophilic graphs, or even deliberately abstains from using homophily. Furthermore, none of the aforementioned works studies how these estimators behave when their (often only implicitly stated) homophily assumptions are violated. Our work addresses this gap by studying uncertainty *without assuming homophily* and empirically confirming that existing estimators are not well-suited for these problems.

# 4. Uncertainty Estimation in MPNNs through Information Theory

We approach MPNNs from the perspective of information theory and track the informativeness of its latent representations throughout the model.

## 4.1. Information in NNs for I.I.D. Data

The seminal work of Tishby & Zaslavsky (2015) phrases training a NN to predict a target variable Y from an input representation X as equivalent to finding a so-called sufficient statistic $f^*(X)$ that fully describes Y and at the same time retains minimal information about X:

$$f^*(X) = \underset{f:I(Y;f(X))=I(Y;X)}{\arg\min} I(X;f(X)). \tag{3}$$

The optimal mapping $f^*$ of Equation (3), therefore, is maximally informative about the target while at the same time discarding all information contained in X that is uninformative regarding the learning problem. In practice, the solution of Equation (3) is unattainable and methods instead realize a trade-off between the informativeness of latent representations about the target and the compression of the input representation $I(X;Z^{(i)})$. For example, NNs benefit from learning a maximally compressed representation that loses as little predictive information as possible.

The layer-wise processing of the input data by a NN induces a Markov Chain of hidden representations $Z^{(i)}$ as depicted in Figure 2. Even though these transitions are deterministic, this perspective allows describing the information that each hidden representation $Z^{(i)}$ retains about the true label $Y$ (and input X) with the data processing inequality (DPI) (Cover, 1999): It states that processing a random variable can only decrease the information about its parents:

$$\begin{aligned} I(Y; Z^{(i+1)}) &= I(Y; Z^{(i)}) - I(Y; Z^{(i)} \mid Z^{(i+1)}) \\ &\leq I(Y; Z^{(i)}). \end{aligned} \quad (4)$$

The DPI Equation (4) holds with equality *only* if no information about Y is lost when the $i + 1$-th layer transforms $Z^{(i)}$ into $Z^{(i+1)}$. This information loss is quantified by $I(Y; Z^{(i)} \mid Z^{(i+1)})$. Intuitively, it measures the information about Y that is provided by $Z^{(i)}$ given $Z^{(i+1)}$. This is exactly the information about Y that is described by $Z^{(i)}$ but not anymore by $Z^{(i+1)}$. A similar statement about information loss regarding X follows from the DPI as well.

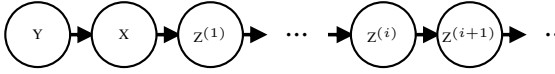

*Figure 2.* NNs for i.i.d. data induce a Markov Chain of random variables $Z^{(i)}$ that correspond to its hidden representations.

Benkert et al. (2024) link this information loss to problems with density-based epistemic uncertainty estimators. These quantify uncertainty with an estimate of $p(X)$ that is often directly computed from NN representations at the penultimate level $Z^{(L-1)}$ or from the logits $Z^{(L)}$ (Wu et al., 2023; Liu et al., 2020a). Because of the DPI, deep representations discard crucial information about both input and target. This is sometimes also referred to as feature collapse (Van Amersfoort et al., 2021). One remedy to this problem is enforcing distance preservation constraints on the NN (Mukhoti et al., 2021a; Liu et al., 2020a; Van Amersfoort et al., 2020). However, this is restrictive regarding the model architecture, prevents post-hoc uncertainty estimation, and often comes at decreased performance (Benkert et al., 2024).

### 4.2. A Data Processing Equality for MPNNs

For problems on graphs, the input X is twofold as it contains both node features $\boldsymbol{F}$ as well as the graph structure $\mathcal{E}$. Furthermore, node features and targets of individual nodes depend on those of other nodes. Therefore, the information-theoretic analysis of (Tishby & Zaslavsky, 2015) does not apply to MPNNs in general. We close this gap by devising a suitable probabilistic model that tracks the information about a node's label Y in the hidden representations of an MPNN in general. Importantly, we do not make any assumptions about how the MPNN layers are realized. We then derive an analog to the DPI which reveals that the information can also *increase* at deeper layers in the network.

Further, we argue that on heterophilic graphs, each latent embedding contains different information about the target variable. We verify this with a simple density-based uncertainty estimator on the joint representations of all layers that provides state-of-the-art epistemic uncertainty.

We track the information about each node $v$'s representation throughout the model individually. To that end, we represent the graph as an ego-graph $\mathcal{G}_v$ with $v$ as an anchor and analyze the informativeness of its representations regarding the label Y of this anchor node.

For MPNNs, in each iteration, only the representations of $v$'s direct neighbors $\mathcal{N}(v)$ propagate information to update $v$'s hidden representation. It is therefore useful to decompose the input, each node in the ego-graph $\mathcal{G}_v$, into disjoint sets $\mathcal{G}_v^{(k)}$ according to their shortest-path distance to $v$.

The information about Y is not only encoded in $v$'s representation but can also be retained in the latent representation of all other nodes in the graph. To represent this, we also introduce random variables that correspond to the representations of all nodes in each $\mathcal{G}_v^{(k)}$. We write $Z_k^{(i)}$ for the representations of the nodes in $\mathcal{G}_v^{(k)}$ after the $i$-th layer. The unprocessed information is denoted with $G_v^{(k)}$. Since $\mathcal{G}_v^{(0)}$ contains just $v$ itself, the random variables $Z_0^{(i)}$ track $v$'s representation over the MPNN layers which we denote with $Z^{(i)}$ for brevity and consistency with the DPI (Equation (4)).

The resulting probabilistic model of MPNNs is depicted in Figure 3. In contrast to the i.i.d. case, it is not a Markov Chain. Nonetheless, its structure reflects intuitive properties about MPNNs that allow relating the information of the node representations $Z^{(i)}$ between subsequent layers to each other, similar to the DPI in Equation (4).

1. For any layer $i$, the label Y and the representation $Z^{(i)}$ are conditionally independent given $G_v^{(0:i)}$. Consequently, the mutual information $I(Y; Z^{(i)} \mid G_v^{(0:i)}) = 0$ and the sequence of random variables $Y \to G_v^{(0:i)} \to Z^{(i)}$ form a Markov Chain. Because of the (i.i.d.) DPI, information about Y that was present in $G_v^{(0:i)}$ may be lost in $Z^{(i)}$ through processing by the $i$ MPNN layers.

2. $Z^{(i)}$ and $G_v^{(i+1)}$ are conditionally independent given $G_v^{(0:i)}$. Therefore, $Z^{(i)}$ can not contain any *additional information* that the $(i + 1)$-hop neighbors $\mathcal{G}_v^{(i+1)}$ of $v$ provide and that was not already present in $\mathcal{G}_v^{(0:i)}$.

3. In contrast to NNs for i.i.d. data, $Z^{(i)}$ and $G_v^{(0:i)}$ are *not* conditionally independent given $Z^{(i-1)}$. The same holds for $Z^{(i)}$ and Y. That means that information that is lost through previous layers *can* be recovered in deeper representations. Intuitively, the other nodes in the graph may retain this information and can transfer it back to $v$ in subsequent MPNN iterations.

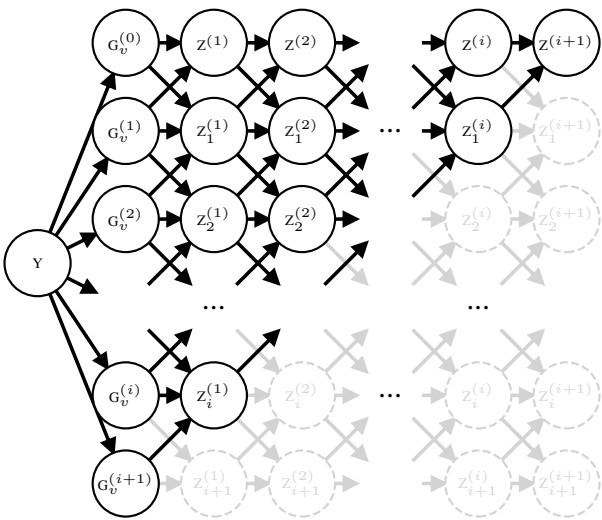

*Figure 3.* Probabilistic model of how the information is processed in MPNNs. The $i$-th layer updates the representations of each $k$-order ego graph $\mathcal{G}_v^{(k)}$ described by random variables $Z_k^{(i)}$.

The graphical model in Figure 3 also allows to condense these properties into a novel analog to the DPI for MPNNs. The difference in informativeness between latent representations decomposes additively into: (i) $\Delta_{(-)}^{(0:i)}$, the information loss / recovery regarding $\mathcal{G}_v^{(0:i)}$, and (ii) $\Delta_{(+)}^{(i+1)}$, the realized information gain through $v$'s extended receptive field in the $(i+1)$-th MPNN iteration.

**Theorem 4.1** (Data Processing Equality for MPNNs). *Let $Z^{(i)}$ be random variables corresponding to the hidden representations of a node $v$ in an MPNN according to Equation (2) after the $i$-th layer. Let $G_v = \sqcup_i G_v^{(i)}$ and $Y$ random variables representing the corresponding ego-graph and label. The information the subsequent representation $Z^{(i+1)}$ holds about $Y$ decomposes additively into the information in $Z^{(i)}$, their relative information $\Delta_{(-)}^{(0:i)}$ w.r.t. $G_v^{(0:i)}$, and the information gain $\Delta_{(+)}^{(i+1)}$ w.r.t. $G_v^{(i+1)} \mid G_v^{(0:i)}$.*

$$I(Y; Z^{(i+1)}) = I(Y; Z^{(i)}) - \Delta_{(-)}^{(0:i)} + \Delta_{(+)}^{(i+1)}. \quad (5)$$

First, we analyze $\Delta_{(-)}^{(0:i)}$. It measures how much information from $G_v^{(0:i)}$ is lost through the $(i+1)$-th MPNN layer while also accounting for the recovery of previously lost information retained in $v$'s neighbors and transferred back to $Z^{(i+1)}$.

**Definition 4.1.** Let $Z^{(i)}$, $Z^{(i+1)}$, $G_v^{(i)}$, and $Y$ be as in Theorem 4.1. The relative information about $Y$ between $Z^{(i)}$ and $Z^{(i+1)}$ contained in $G_v^{(0:i)}$ is defined as:

$$\Delta_{(-)}^{(0:i)} := I(Y; G_v^{(0:i)} \mid Z^{(i+1)}) - I(Y; G_v^{(0:i)} \mid Z^{(i)}).$$

$\Delta_{(-)}^{(0:i)}$ can roughly be seen as an analog to the information

loss term $I(Y; Z^{(i)} \mid Z^{(i+1)})$ of the DPI in Equation (4). Notably, the information from $G_v^{(0:i)}$ that is captured in $v$'s representation does not necessarily have to decrease with depth, which contrasts the DPI. If $v$'s neighbors recover more information than the $i$ MPNN layer successively lose in $v$'s embedding, $\Delta_{(-)}^{(0:i)}$ assumes a negative value, effectively resulting in *information gain*. This gain is naturally bounded by the information that all $G_v^{(0:i)}$ provide about $Y$.

**Proposition 4.1.** *Let $Z^{(i)}$, $Z^{(i+1)}$, $G_v^{(i)}$, and $Y$ be as in Theorem 4.1. The relative information about $Y$ that $Z^{(i+1)}$ and $Z^{(i)}$ hold in terms of the information in $G_v^{(0:i)}$ is bounded by $-I(Y; G_{0:i})$.*

$$-\Delta_{(-)}^{(0:i)} \leq I(Y; G_v^{(0:i)}).$$

The information gain term $\Delta_{(+)}^{(i+1)}$ has no correspondence in the DPI. It reflects the increasing receptive field of the node $v$ at deeper layers: With each additional layer, the representation $Z^{(i+1)}$ also depends on $G_v^{(i+1)}$. The resulting *additional* information $I(Y; G_v^{(i+1)} \mid G_v^{(0:i)})$ that is not lost through the $(i+1)$-th is given as:

**Definition 4.2.** Let $Z^{(i)}$, $Z^{(i+1)}$, $G_v^{(i)}$, and $Y$ be as in Theorem 4.1. The information gain about $Y$ in $Z^{(i+1)}$ that is provided by $G_v^{(i+1)}$ is defined as:

$$\Delta_{(+)}^{(i+1)} := I(Y; G_v^{(i+1)} \mid G_v^{(0:i)})$$
$$- I(Y; G_v^{(i+1)} \mid G_v^{(0:i)}, Z^{(i+1)}).$$

Intuitively, $\Delta_{(+)}^{(i+1)}$ takes all available additional information in $I(Y; G_v^{(i+1)} \mid G_v^{(0:i)})$ and subtracts what is lost through processing this information into $Z^{(i+1)}$. Similar to the DPI, this loss is described by $I(Y; G_v^{(i+1)} \mid G_v^{(0:i)}, Z^{(i+1)})$. Therefore, the gain $\Delta_{(+)}^{(i+1)}$ is trivially upper bounded $I(Y; G_v^{(i+1)} \mid G_v^{(0:i)})$ exactly when no information is lost. In the worst case, all information is lost and consequently, the information gain is lower bounded by 0 (see Proposition A.1).

### 4.3. An Information-based Definition of Homophily

We propose a novel definition of homophily grounded in information theory that enables studying how homophily affects the information in the latent representations of MPNNs.

**Definition 4.3.** Let $G_v^{(i)}$ and $Y$ be as in Theorem 4.1. We define the information homophily between the $(i+1)$-hop neighbors $G_v^{(i+1)}$ and the $i$-th order ego graph $G_v^{(0:i)}$ as:

$$h_v^{(i+1)} := I(G_v^{(i+1)}; G_v^{(0:i)}) - I(G_v^{(i+1)}; G_v^{(0:i)} \mid Y).$$

$h_v^{(i+1)}$ measures how much *semantically relevant* information is shared between the $(i+1)$-hop neighbors of

a node and the ego-graphs up to distance $i$. The term $I(G_v^{(i+1)}; G_v^{(0:i)} \mid Y)$ quantifies the shared information that is not described by the label Y and, therefore, irrelevant for the task. By subtracting this term, our definition of homophily only considers the shared information that is also meaningful in terms of Y. For example, in image classification, an image's background color does not affect its class and therefore also does not contribute to the information homophily according to this definition.

Other than existing notions of homophily (Appendix C.1), $h_v^{(i+1)}$ describes per-node homophily at each distance $i$ individually. For example, $h_v^{(1)}$ compares a node's own features to those of its neighbors similarly to other definitions. Instead of quantifying homophily through node labels or features alone, it only concerns the similarity of node features that are semantically relevant to the label Y. As computing the mutual information between random variables requires access to their joint distribution or samples from it, information homophily can typically not be computed for real data. However, it directly affects the upper bound on the information gain $\Delta_{(+)}^{(i+1)}$ each $Z^{(i+1)}$ can realize over the previous MPNN embeddings:

**Proposition 4.2.** *Let* $G_v^{(i)}$ *and* Y *be as in Theorem 4.1. The attainable information gain* $\Delta_{(+)}^{(i+1)}$ *decreases with homophily* $h_v^{(i+1)}$:

$$\Delta_{(+)}^{(i+1)} \leq I(Y; G_v^{(i+1)} \mid G_v^{(0:i)}) = I(Y; G_v^{(i+1)}) - h_v^{(i+1)}.$$

Proposition 4.2 reveals that with increasing information homophily, each subsequent hidden representation can realize less information gain about the label. Intuitively, the semantic similarity between $G_v^{(0:i)}$ and $G_v^{(i+1)}$ prohibits the latent representation $Z^{(i+1)}$ to contain information that is not already contained in $Z^{(0:i)}$. The more heterophilic a graph is, the more of the semantically relevant information $I(Y; G_v^{(i+1)})$ is exclusive to the $(i+1)$-hop neighbors and can only first be captured by $Z^{(i+1)}$.

### 4.4. Uncertainty Estimation in MPNNs Through Joint Latent Density Estimation (JLDE)

Proposition 4.2 implies that in homophilic graphs, the information gain of deeper embeddings over more shallow ones diminishes as nodes are semantically similar to their neighbors. Therefore, it suffices to estimate the data density from a single latent representation (Fuchsgruber et al., 2024b).

Our analysis is especially insightful for heterophilic graphs. Since nodes in the $i$-th order ego graph $G_v^{(i)}$ preferably connect to nodes with different semantics, $G_v^{(i+1)}$ likely contains information about Y that is different from the information in $G_v^{(i)}$ and the information gain $\Delta_{(+)}^{(i+1)}$ can assume large values. Consequently, each hidden representation

provides different semantic information: $I(Y; Z^{(0:L)}) \gg I(Y; Z^{(i)})$. Effective density-based epistemic uncertainty should consider all $Z^{(i)}$ jointly.

We confirm the validity of this insight by opting for the most simple realization: We quantify epistemic uncertainty using a KNN-based density (Loftsgaarden & Quesenberry, 1965) on the concatenated latent representations $z^{(1)}, \dots z^{(L)}$ (see Appendix C.2). While our primary goal is to empirically motivate the principle of **J**oint **L**atent **D**ensity **E**stimation (JLDE), we conjecture that future work can benefit from more sophisticated density models.

$$u^{\text{epi}}(v) = -\log p(\mathcal{G}_v) \approx -\log p_\theta \left( \mathop{\|}_{i=1}^{L} z^{(i)} \right) \quad (6)$$

$$\propto \sum_{u \in \text{k-NN}(\|_{i=1}^{L} \tilde{z}_v^{(i)})} \left| \left( \mathop{\|}_{i=1}^{L} \tilde{z}_v^{(i)} \right) - \left( \mathop{\|}_{i=1}^{L} \tilde{z}_u^{(i)} \right) \right|_2 \quad (7)$$

JLDE can be applied post-hoc to any MPNN and enables uncertainty quantification with a single forward pass. Importantly, in contrast to prior work, it does not rely on assumptions regarding homophily, for example by post-processing uncertainty using graph diffusion (Stadler et al., 2021; Wu et al., 2023; Fuchsgruber et al., 2024b).

## 5. Experiments

We evaluate JLDE by exposing it to distribution shifts proposed by previous work (Stadler et al., 2021; Wu et al., 2023; Fuchsgruber et al., 2024b), each of which corresponds to a different kind of anomaly. Like Shchur et al. (2019) and Stadler et al. (2021), we average all results over 10 dataset splits with 10 model initializations each and report the standard deviations in Appendix E.

### 5.1. Setup

**Datasets and Distribution Shifts.** We consider these three distribution shifts:

(i) **Leave-Out-Class (LoC).** We exclude nodes in a subset of classes from the training set $\mathcal{V}_\mathcal{D}$ and treat them as out-of-distribution (o.o.d.) during inference.

(ii) **Near Feature Perturbations.** We randomly sample a subset of designated o.o.d. nodes and replace their features with noise during inference. We emulate a distribution shift with semantics similar to the training data through Bernoulli noise for datasets with bag-of-word features (CoraML, PubMed, Chamaleon and Squirrel) and Gaussian noise $\mathcal{N}(\mu, \Sigma)$ for graphs with continuous features (Amazon Ratings, Roman Empire). Mean $\mu$ and diagonal covariance $\Sigma$ are the maximum likelihood estimates from the dataset.

(iii) **Far Feature Perturbations** are generated by replacing the features of designated o.o.d. nodes with noise from $\mathcal{N}(\mathbf{0}, \mathbf{I})$. For bag-of-word features, this distribution shift can be seen as out-of-domain.

We build on the insights of Platonov et al. (2023) regarding GNN evaluation on heterophilic data by primarily focusing on the novel heterophilic *Amazon Ratings* and *Roman Empire* datasets. The three other heterophilic datasets are binary classification problems that do not permit a LoC distribution shift. Additionally, we evaluate our method on the revisions of the heterophilic *Squirrel* and *Chameleon* datasets. We also study the two homophilic citation networks *CoraML* (Bandyopadhyay et al., 2005) and *PubMed* (Namata et al., 2012) to compare JLDE to uncertainty estimators that explicitly leverage the homophily in the data.

**Backbone Models.** JLDE can be applied post-hoc to any MPNN backbone. Since uncertainty estimation improves with predictive performance (Fort et al., 2021), we study the best-performing architectures according to Platonov et al. (2023) and our benchmark (Appendix D): Augmented versions of GCN (*Res-GCN*), GATv2 with explicit separation between the own representation of a node and an aggregate of information from its neighbors (*Res-GAT-Sep*), *GATE*, and *GPRGNN*. Additionally we compare JLDE to estimates on *FAGCN*, *FSGNN* as GNNs for heterophilic graphs. See Appendix C.2 for details.

**Uncertainty Estimators.** We compare JLDE to different uncertainty estimators: *GPN* and *SGCN* quantify uncertainty using Evidential Learning. Bayesian GCNs (*BGCN*s) model the weights of a GCN as Gaussians. For other baselines, we use a logit-based EBM as epistemic uncertainty. Additionally, we compare to the following other model-agnostic methods: An ensemble of 10 (*Ens.*), Monte-Carlo Dropout (*MCD*), energy-based uncertainty from logits (*EBM*), as well as the graph-specific EBMs GNN-*Safe* and *GEBM*.

**Metrics.** Like previous work (Stadler et al., 2021; Fuchsgruber et al., 2024b), our main proxy for the quality of an uncertainty is o.o.d. detection. We report the AUC-ROC and AUC-PR metrics to the binary classification problem of distinguishing between in-distribution (i.d.) and o.o.d. data. We supply how well an uncertainty estimator identifies erroneous predictions in Appendix E.3. Since JLDE is a post-hoc estimator it does not affect the backbone's (softmax) predictions and the associated aleatoric uncertainty (quantified as the maximal softmax response) and its calibration. Both are backbone-dependent and we report them only for completeness. While beyond the scope of this work, we remark that our framework permits other post-hoc improvements like temperature scaling (Guo et al., 2017). We report the Expected Calibration Error (ECE) (Naeini et al., 2015) of each backbone in Appendix E.4 for completeness.

## 5.2. Results

**Out-of-Distribution Detection.** Table 1 shows the o.o.d. detection performance of different epistemic and aleatoric uncertainty estimators. Post-hoc methods are applied to a Res-GCN backbone (full results in Table 7). On heterophilic data, JLDE has high efficacy on all distribution shifts. On the homophilic CoraML, it compares well against other model-agnostic estimates. This underlines that the principle of JLDE is especially effective for heterophilic graphs where the information of the latent node representation after each layer can vary significantly. At the same time, in homophilic settings, it provides high-quality uncertainty without resorting to post-processing with (homophilic) smoothing kernels like GPN, Safe, or GEBM.

*Table 1.* O.o.d. detection using different uncertainty estimators (best and runner-up). On heterophilic graphs, we perform the best while being competitive with other model-agnostic estimates on homophilic data. Grey cells indicate numbers associated with JLDE.

| | Model | LoC AUC-ROC (Alea. / Epi.) | Acc.↑ | Near-Features AUC-ROC (Alea. / Epi.) | Acc.↑ | Far-Features AUC-ROC (Alea. / Epi.) | Acc.↑ |
|---|---|---|---|---|---|---|---|
| Roman Empire | GPN | 60.3/64.0 | 48.6 | 59.9/66.4 | 38.1 | 54.8/77.3 | 38.0 |
| | SGCN | 60.0/56.5 | 58.3 | 55.7/54.1 | 45.5 | 12.5/7.1 | 30.6 |
| | BGCN | 54.9/60.7 | 53.0 | 58.1/46.2 | 42.4 | 38.3/73.6 | 28.8 |
| | GPRGNN | 66.5/63.4 | 78.1 | 86.5/87.9 | 64.8 | 17.6/79.1 | 61.0 |
| | FSGNN | 73.0/70.2 | 86.6 | 70.5/76.3 | 71.6 | 21.9/3.7 | 56.9 |
| | FAGCN | 72.7/68.8 | 83.6 | 77.8/85.6 | 68.0 | 23.9/11.2 | 44.3 |
| | Ens. | 76.4/76.2 | | 78.4/78.2 | | 89.2/86.9 | |
| | MCD | 76.3/75.9 | | 75.8/72.5 | | 82.7/72.9 | |
| | EBM | 73.3/72.5 | 87.6 | 74.5/81.3 | 72.3 | 83.0/92.0 | 70.6 |
| | Safe | 73.3/64.8 | | 74.5/73.4 | | 83.0/81.0 | |
| | GEBM | 73.3/53.6 | | 74.5/66.4 | | 83.0/80.9 | |
| | JLDE | 73.3/76.9 | | 74.5/93.8 | | 83.0/100.0 | |
| Amazon Ratings | GPN | 48.7/51.5 | 54.2 | 54.3/52.4 | 46.8 | 52.6/58.5 | 47.0 |
| | SGCN | 47.8/46.6 | 55.5 | 54.0/54.2 | 47.2 | 27.7/22.8 | 33.5 |
| | BGCN | 52.7/49.0 | 51.4 | 54.1/46.6 | 44.2 | 44.5/72.1 | 33.5 |
| | GPRGNN | 52.5/52.1 | 57.9 | 62.1/70.0 | 50.6 | 10.9/48.6 | 34.1 |
| | FSGNN | 54.4/53.6 | 56.8 | 45.1/38.6 | 49.1 | 12.7/2.2 | 38.6 |
| | FAGCN | 53.1/53.3 | 58.1 | 59.8/62.9 | 51.2 | 17.9/12.0 | 31.7 |
| | Ens. | 51.4/57.9 | | 60.3/63.6 | | 74.3/50.4 | |
| | MCD | 49.2/53.5 | | 56.6/58.2 | | 72.9/33.2 | |
| | EBM | 48.8/48.9 | 56.0 | 54.8/56.1 | 48.2 | 78.2/94.4 | 47.3 |
| | Safe | 48.8/48.0 | | 54.8/55.3 | | 78.2/67.2 | |
| | GEBM | 48.9/49.9 | | 54.8/47.1 | | 78.2/58.9 | |
| | JLDE | 48.8/61.2 | | 54.8/65.4 | | 78.2/99.2 | |
| CoraML | GPN | 82.8/81.9 | 88.9 | 55.4/53.8 | 80.9 | 49.1/61.4 | 81.1 |
| | SGCN | 86.3/87.2 | 89.4 | 63.7/66.9 | 82.0 | 16.8/12.2 | 34.4 |
| | BGCN | 85.3/75.1 | 88.3 | 59.8/57.7 | 81.3 | 28.5/34.3 | 34.1 |
| | GPRGNN | 85.1/87.4 | 90.2 | 61.0/65.9 | 84.1 | 17.5/4.8 | 37.3 |
| | FSGNN | 82.7/81.8 | 87.5 | 63.5/69.8 | 71.7 | 6.8/1.6 | 40.2 |
| | FAGCN | 85.0/86.4 | 88.9 | 68.7/76.0 | 81.4 | 11.2/1.2 | 40.4 |
| | Ens. | 81.3/81.4 | | 52.7/52.8 | | 84.7/78.4 | |
| | MCD | 79.6/78.1 | | 54.6/55.1 | | 88.0/64.0 | |
| | EBM | 76.9/76.0 | 88.3 | 52.1/53.3 | 79.6 | 85.0/93.5 | 77.1 |
| | Safe | 76.9/80.7 | | 52.1/51.0 | | 85.0/72.0 | |
| | GEBM | 76.9/81.4 | | 52.1/53.3 | | 85.0/80.5 | |
| | JLDE | 76.9/80.8 | | 52.1/54.9 | | 85.0/95.7 | |

**Misclassification Detection.** Like prior work (Stadler et al., 2021; Fuchsgruber et al., 2024b), we find that aleatoric uncertainty estimators are better equipped for misclassification detection than epistemic uncertainty (Tables 13 and 14). JLDE performs similarly to other epistemic estimates.

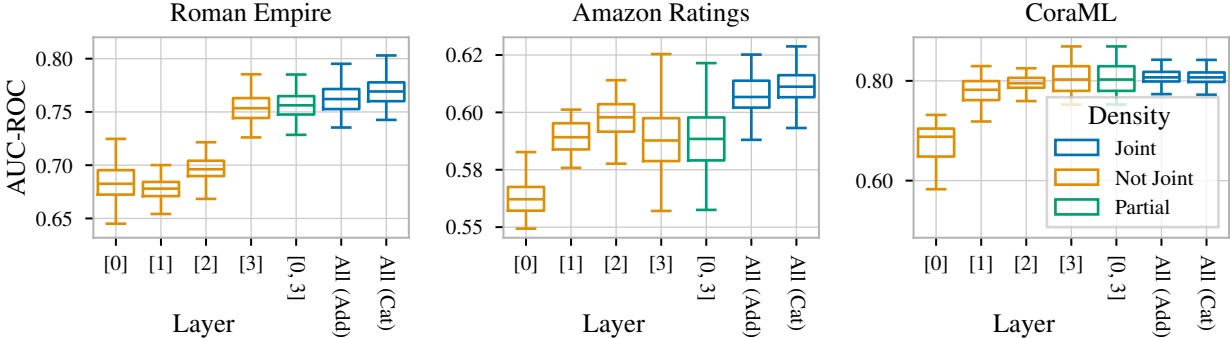

*Figure 4.* O.o.d. detection AUC-ROC (↑) under a leave-out-classes distribution shift when estimating the density from different layers of a Res-GCN backbone. On heterophilic datasets (Amazon Ratings, Roman Empire), joint density estimation performs best as different layer representations provide different information.

## 5.3. Ablations

**Joint Density Estimation.** We empirically verify our analysis regarding the informativeness of latent node representations in different MPNN layers by estimating the data density from (i) each latent representation $z^{(i)}$ individually, (ii) the concatenation of only the first and last representations $z^{(1)}$ and $z^{(L)}$, (iii) each latent representation $z^{(i)}$ independently and averaging the uncertainty estimates (All (Add)), and (iv) to the concatenated latent representations of all layers as in Equation (6) (All (Cat)).

Figure 4 confirms the claims of Section 4.2: For heterophilic graphs, each latent representation $z^{(i)}$ can contain different information originating from different $k$-hop neighborhoods of each node. The information can both decrease and also increase with depth as $\Delta_{(-)}^{(0:i)}$ can assume negative and positive values. For heterophilic problems, only *jointly* considering all latent representations results in a faithful representation of the data density $p(\mathcal{G}_v)$ and, consequently, high-quality uncertainty. The advantage in modeling a joint distribution over aggregating per-layer density suggests an interplay between the information contained in each $z^{(i)}$ that may be further exploited with more sophisticated methods to obtain even better approximations of the data density.

For the homophilic CoraML graph, we also confirm our hypothesis that each additional $k$-hop neighborhood contributes only information similar to the previous node representation $z^{(k-1)}$. Each MPNN iteration amplifies this information and there is no benefit in considering shallow representations as well. Similarly, JLDE's performance matches estimating epistemic uncertainty from the last layer only. This finding aligns well with the success of uncertainty estimators that are derived from deep layers like GEBM in homophilic problems. Nonetheless, including shallow representations in the density estimation does not notably harm the performance of JLDE's on homophilic graphs.

**Backbones.** The principle of JLDE can be applied post-hoc to any MPNN. In Tables 9 to 11, we evaluate the o.o.d.

*Table 2.* O.o.d. detection rank of model-agnostic epistemic estimators averaged over all distribution shifts (best and runner-up).

| | Model | Roman Empire | Amazon Ratings | Chameleon | Squirrel | CoraML | PubMed |
|---|---|---|---|---|---|---|---|
| Res-GCN | Ens. | 2.5 | 2.8 | 3.8 | 3.5 | 2.8 | 2.5 |
| | MCD | 4.2 | 3.8 | 4.2 | 4.8 | 4.2 | 4.8 |
| | EBM | 3.0 | 4.0 | 3.8 | 4.0 | 4.2 | 4.2 |
| | Safe | 4.5 | 5.0 | 4.0 | 3.8 | 4.8 | 4.0 |
| | GEBM | 5.8 | 4.5 | 2.8 | 2.5 | 2.8 | 2.8 |
| | JLDE | 1.0 | 1.0 | 2.5 | 2.5 | 2.2 | 2.8 |
| Res-GAT-Sep | Ens. | 1.8 | 2.8 | 5.5 | 2.8 | 2.8 | 2.5 |
| | MCD | 3.2 | 3.8 | 4.0 | 4.0 | 4.0 | 4.5 |
| | EBM | 3.2 | 3.8 | 3.5 | 4.2 | 4.2 | 4.0 |
| | Safe | 4.8 | 4.2 | 3.5 | 5.0 | 3.8 | 4.5 |
| | GEBM | 6.0 | 5.5 | 1.8 | 2.8 | 3.2 | 2.0 |
| | JLDE | 2.0 | 1.0 | 2.8 | 2.2 | 3.0 | 3.5 |
| Res-GATE | Ens. | 2.8 | 2.8 | 4.5 | 3.8 | 3.2 | 3.2 |
| | MCD | 4.0 | 4.2 | 4.0 | 3.8 | 4.5 | 4.8 |
| | EBM | 3.2 | 4.2 | 4.0 | 4.2 | 4.5 | 4.0 |
| | Safe | 4.5 | 3.2 | 4.2 | 4.2 | 4.8 | 4.5 |
| | GEBM | 5.5 | 5.5 | 2.0 | 3.0 | 3.0 | 3.0 |
| | JLDE | 1.0 | 1.0 | 2.2 | 2.0 | 1.0 | 1.5 |
| GPRGNN | Ens. | 3.2 | 2.8 | 5.2 | 3.5 | 5.2 | 3.5 |
| | MCD | 4.2 | 3.5 | 3.8 | 2.8 | 4.8 | 4.8 |
| | EBM | 3.8 | 4.5 | 4.2 | 4.2 | 3.5 | 3.8 |
| | Safe | 3.2 | 5.2 | 4.0 | 4.8 | 2.8 | 3.5 |
| | GEBM | 5.2 | 2.8 | 2.0 | 2.5 | 1.8 | 1.8 |
| | JLDE | 1.2 | 2.2 | 1.8 | 3.2 | 3.0 | 3.8 |

detection performance of JLDE on different MPNN backbones under different distribution shifts. JLDE outperforms other baselines in terms of AUC-ROC. This is also summarized by Table 2 that shows the rank (↓) of each estimator averaged over all three distribution shifts. Therefore, our theoretical analysis applies to a broad range of MPNNs on which the concept of JLDE provides high-quality epistemic uncertainty for both heterophilic and homophilic graphs.

## 6. Limitations

We propose a design principle for epistemic uncertainty on heterophilic graphs but do not aim to improve aleatoric uncertainty or its calibration. To verify the applicability of this principle, we opt for the most simple realization of JLDE but conjecture that more sophisticated methods for combining hidden representations and estimating their den-

sity likely further improve the resulting uncertainty estimate. Also, we focus our evaluation on node classification but the information-theoretic perspective and its implications apply to regression problems and other non-i.i.d. domains that can be cast into graphs as well.

## 7. Conclusion

We introduce JLDE as a design principle for uncertainty estimation on heterophilic graphs that is applicable to a broad range of MPNNs. Through an information-theoretic analysis of MPNNs, we derive an analog to the DPI that enables tracking the information over the layers of the network. It reveals that in heterophilic graphs, each embedding provides unique information about the data distribution. Leveraging this insight even with a simple density estimator consistently outperforms existing approaches on different datasets, distribution shifts, and backbones. We believe that our work provides the theoretical groundwork for advancing uncertainty estimation on graphs beyond the homophily assumption of prior work.

## Impact Statement

We conduct a theoretical analysis of message-passing neural networks and derive design principles for accurate uncertainty estimation on heterophilic graphs. This contributes to the advancement of reliable and trustworthy AI. Nonetheless, we want to explicitly encourage researchers and practitioners who use or build on our insights to actively consider and evaluate our proposed uncertainty estimate, especially in high-risk domains.

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

# A. Proofs

**Theorem 4.1** (Data Processing Equality for MPNNs). *Let $Z^{(i)}$ be random variables corresponding to the hidden representations of a node $v$ in an MPNN according to Equation (2) after the $i$-th layer. Let $G_v = \sqcup_i G_v^{(i)}$ and $Y$ random variables representing the corresponding ego-graph and label. The information the subsequent representation $Z^{(i+1)}$ holds about $Y$ decomposes additively into the information in $Z^{(i)}$, their relative information $\Delta_{(-)}^{(0:i)}$ w.r.t. $G_v^{(0:i)}$, and the information gain $\Delta_{(+)}^{(i+1)}$ w.r.t. $G_v^{(i+1)} \mid G_v^{(0:i)}$.*

$$I(Y; Z^{(i+1)}) = I(Y; Z^{(i)}) - \Delta_{(-)}^{(0:i)} + \Delta_{(+)}^{(i+1)}. \tag{5}$$

*Proof.* First, we notice that that for each layer $i$, $Y \to G_v^{(0:i)} \to Z^{(i)}$ form a Markov Chain. Therefore, the Data Processing Inequality (Equation (4)) applies and we obtain:

$$I(Y; Z^{(i)}) = I(Y; G_v^{(0:i)}) - I(Y; G_v^{(0:i)} \mid Z^{(i)}) \tag{8}$$

$$I(Y; G_v^{(0:i)}) = I(Y; Z^{(i)}) + I(Y; G_v^{(0:i)} \mid Z^{(i)}). \tag{9}$$

Similarly, when applied to $Z^{(i+1)}$:

$$
\begin{aligned}
I(Y; Z^{(i+1)}) &= I(Y; G_v^{(0:i+1)}) - I(Y; G_v^{(0:i+1)} \mid Z^{(i+1)}) \\
&= I(Y; G_v^{(0:i)}) + I(Y; G_v^{(i+1)} \mid G_v^{(0:i)}) - I(Y; G_v^{(0:i+1)} \mid Z^{(i+1)}) \\
&= I(Y; Z^{(i)}) + I(Y; G_v^{(0:i)} \mid Z^{(i)}) + I(Y; G_v^{(i+1)} \mid G_v^{(0:i)}) - I(Y; G_v^{(0:i+1)} \mid Z^{(i+1)}) \\
&= I(Y; Z^{(i)}) + I(Y; G_v^{(0:i)} \mid Z^{(i)}) + I(Y; G_v^{(i+1)} \mid G_v^{(0:i)}) \\
&\quad - I(Y; G_v^{(0:i)} \mid Z^{(i+1)}) - I(Y; G_v^{(i+1)} \mid G_v^{(0:i)}, Z^{(i+1)}) \\
&= I(Y; Z^{(i)}) - \left( I(Y; G_v^{(0:i)} \mid Z^{(i+1)}) - I(Y; G_v^{(0:i)} \mid Z^{(i)}) \right) \\
&\quad + \left( I(Y; G_v^{(i+1)} \mid G_v^{(0:i)}) - I(Y; G_v^{(i+1)} \mid G_v^{(0:i)}, Z^{(i+1)}) \right).
\end{aligned}
$$

$\square$

First, we apply the DPI to $I(Y; Z^{(i+1)})$ similar to what is outlined in the proof for $I(Y; Z^{(i)})$. Then, we apply the chain rule of MI. Going from the second to the third line, we use the DPI applied to $I(Y; G_v^{(0:i)})$. We then apply the chain rule again by splitting $G_v^{(0:i+1)}$ into $G_v^{(0:i)}$ and $G_v^{(i+1)}$. Lastly, we regroup terms to obtain the definitions for $\Delta_{(+)}^{(i+1)}$ and $\Delta_{(-)}^{(0:i)}$.

**Proposition 4.1.** *Let $Z^{(i)}$, $Z^{(i+1)}$, $G_v^{(i)}$, and $Y$ be as in Theorem 4.1. The relative information about $Y$ that $Z^{(i+1)}$ and $Z^{(i)}$ hold in terms of the information in $G_v^{(0:i)}$ is bounded by $-I(Y; G_{0:i})$.*

$$-\Delta_{(-)}^{(0:i)} \le I(Y; G_v^{(0:i)}).$$

*Proof.*

$$
\begin{aligned}
-\Delta_{(-)}^{(0:i)} &= I(Y; G_v^{(0:i)} \mid Z^{(i)}) - I(Y; G_v^{(0:i)} \mid Z^{(i+1)}) \\
&\le I(Y; G_v^{(0:i)} \mid Z^{(i)}) \\
&= I(Y; G_v^{(0:i)}) - I(Y; Z^{(i)}) \\
&\le I(Y; G_v^{(0:i)}).
\end{aligned}
$$

$\square$

First, we insert the definition of $\Delta_{(-)}^{(0:i)}$. Then, we use that MI is non-negative. We then apply the chain rule of MI and, lastly, the non-negativeness of the chain rule once more to obtain the desired bound.

**Proposition 4.2.** *Let* $G_v^{(i)}$ *and* $Y$ *be as in Theorem 4.1. The attainable information gain* $\Delta_{(+)}^{(i+1)}$ *decreases with homophily* $h_v^{(i+1)}$:

$$\Delta_{(+)}^{(i+1)} \leq I(Y; G_v^{(i+1)} \mid G_v^{(0:i)}) = I(Y; G_v^{(i+1)}) - h_v^{(i+1)}.$$

*Proof.*

$$
\begin{aligned}
\Delta_{(+)}^{(i+1)} &\leq I(Y; G_v^{(i+1)} \mid G_v^{(0:i)}) \\
&= I(G_v^{(i+1)}; Y, G_v^{(0:i)}) - I(G_v^{(i+1)}; G_v^{(0:i)}) \\
&= I(G_v^{(i+1)}; Y) + I(G_v^{(i+1)}; G_v^{(0:i)} \mid Y) - I(G_v^{(i+1)}; G_v^{(0:i)}) \\
&= I(G_v^{(i+1)}; Y) - \left[ I(G_v^{(i+1)}; G_v^{(0:i)}) - I(G_v^{(i+1)}; G_v^{(0:i)} \mid Y) \right] \\
&= I(Y; G_v^{(i+1)}) - h_v^{(i+1)}
\end{aligned}
$$

$\square$

The inequality follows directly from the definition of $\Delta_{(+)}^{(i+1)}$ and MI being non-negative. Then, we apply the chain rule of mutual information twice and regroup the terms to obtain the definition of homophily proposed in Definition 4.3.

**Proposition A.1.** *Let* $Z^{(i)}$, $Z^{(i+1)}$, $G_v^{(i)}$, *and* $Y$ *be as in Theorem 4.1. The gain in information about* $Y$ *by additionally considering* $G_v^{(i+1)}$ *is non-negative.*

$$\Delta_{(+)}^{(i+1)} = I(Y; Z^{(i+1)} \mid G_v^{(0:i)}) \geq 0.$$

Proposition A.1 also reveals that the information gain can be expressed as the information $Z^{(i+1)}$ that is not already contained in $G_v^{(0:i)}$ and, thus, must have originated in $G_v^{(i+1)}$.

*Proof.*

$$
\begin{aligned}
\Delta_{(+)}^{(i+1)} &= I(Y; G_v^{(i+1)} \mid G_v^{(0:i)}) - I(Y; G_v^{(i+1)} \mid G_v^{(0:i)}, Z^{(i+1)}) \\
&= I(Y; G_v^{(i+1)} \mid G_v^{(0:i)}) + I(Y; Z^{(i+1)} \mid G_v^{(0:i+1)}) - I(Y; G_v^{(i+1)} \mid G_v^{(0:i)}, Z^{(i+1)}) \\
&= I(Y; G_v^{(i+1)} \mid G_v^{(0:i)}) + I(Y; Z^{(i+1)} \mid G_v^{(i+1)}, G_v^{(0:i)}) - I(Y; G_v^{(i+1)} \mid G_v^{(0:i)}, Z^{(i+1)}) \\
&= I(Y; G_v^{(i+1)}, Z^{(i+1)} \mid G_v^{(0:i)}) - I(Y; G_v^{(i+1)} \mid G_v^{(0:i)}, Z^{(i+1)}) \\
&= I(Y; Z^{(i+1)} \mid G_v^{(0:i)}) + I(Y; G_v^{(i+1)} \mid G_v^{(0:i)}, Z^{(i+1)}) - I(Y; G_v^{(i+1)} \mid G_v^{(0:i)}, Z^{(i+1)}) \\
&= I(Y; Z^{(i+1)} \mid G_v^{(0:i)}) \\
&\geq 0.
\end{aligned}
$$

$\square$

First, we insert the definition of $\Delta_{(+)}^{(i+1)}$. Then we use that $I(Y; Z^{(i+1)} \mid G_v^{(0:i+1)}) = 0$ from the DPI. We then split $G_v^{(0:i+1)}$ into $G_v^{(i+1)}$ and $G_v^{(0:i)}$. The chain rule of MI yields the next line. Then, we use the chain rule again but condition on $Z^{(i+1)}$. Next, the last two terms cancel and we are left with a MI which is non-negative.

## B. Related Work: GNNs for Heterophilic Graphs

**Paradigms for Heterophilic Node Classification.** Compared to the more frequently studied homophilic graphs, heterophily poses difficulties because there are no community structures that can be exploited with smoothing. Methods that extend GNNs towards heterophilic graphs can be grouped into either adjusting the graph structure or changing the model to adapt to the high-frequency information. The former class of models alters the structure of the original graph to recover a homophilic problem. Pei et al. (2020b) facilitate a virtual neighborhood to aggregate information from nodes that are geometrically

similar. Other work constructs a surrogate adjacency to adjust the propagation of information (Jin et al., 2021; Zheng et al., 2024). Also, resorting to a fully connected graph has been proposed (Li et al., 2022). The latter paradigm is to model the high-frequency patterns in the graph. While for homophilic problems, graph diffusion has proven effective (Gasteiger et al., 2018; 2022), smoothing neglects the high-frequency patterns in heterophilic graphs. Generalizing Page-Rank-based diffusion to allow for learnable negative weights assigned to each $k$-hop neighborhood accommodates non-smooth features and improves performance (Chien et al., 2021). Other work like FAGCN (Bo et al., 2021) instead uses gating mechanisms to simultaneously account for low and high-frequency information. It can also be explicitly modeled using the graph Laplacian (Luan et al., 2022). Other work shows that separating the own features of a node from adjacent information can improve performance as well (Zhu et al., 2020). FSGNN (Maurya et al., 2021) aggregates representations from all $k$-hop neighborhoods.

**Modeling Graph Inductive Biases for Performance.** Recent work has focused on better understanding and refining the inductive biases embedded in GNNs, particularly regarding the role of graph structure, neighborhood similarity, and information aggregation across layers. Luan et al. (2022) revisit GNN performance under heterophily, challenging traditional homophily-based assumptions and proposing adaptive channel mixing to enhance representation learning across diverse neighborhoods. Similarly, Luan et al. (2024) analyze when GNNs are beneficial for node classification by introducing new metrics based on intra- and inter-class node distinguishability, showing that homophily alone does not fully explain GNN effectiveness. On the architectural side, Zhang & Li (2021) propose Nested GNNs, which extend beyond rooted subtrees to capture richer local substructures, emphasizing the importance of localized graph context. Complementarily, Yang et al. (2021) investigate feature propagation in GCNs and propose using the difference between a layer's input and output to mitigate over-smoothing, but do not aggregate information across multiple layers.

In contrast, our work examines the inductive biases of GNNs from an information-theoretic perspective. We formally prove the benefits of leveraging information from *all* layers rather than relying solely on the final one or the difference of two. Building on this insight, we propose a novel, theoretically-grounded uncertainty estimator that achieves state-of-the-art performance. We further demonstrate that our approach is especially effective in graphs with strong heterophily, where neighbor information is less reliable and global feature integration becomes more important.

## C. Experimental Details

### C.1. Datasets

We use four heterophilic datasets and two homophilic datasets. Our primary focus lies on the novel heterophilic graphs provided by Platonov et al. (2023) as they are devised specifically for benchmarking heterophilic models. Out of the five datasets the authors develop, only two correspond to multi-class classification problems and permit a LoC distribution shift: Amazon Ratings and Roman Empire. We additionally consider the Chameleon and Squirrel dataset (Pei et al., 2020a) but use the filtered versions that Platonov et al. (2023) provide to counteract data leakage and incorrect data pre-processing. For comparison, we also include two popular homophilic graphs: CoraML (Bandyopadhyay et al., 2005) and PubMed (Namata et al., 2012). The statistics for all six datasets are reported in Table 3.

*Table 3.* Statistics of the datasets: Features are *bag of words* (BoW), *TF/IDF weighted word frequencies* (TF/IDF-WF), *word embeddings* (WE) and mean of word embeddings (M-WE).

| | #Nodes | #Edges | #Features | Feature Type | #Clases | $h_{\mathcal{E}}$ | $h_{\mathcal{V}}$ | $h_C$ | $h_A$ | %LoC | #LoC | LoC Type |
|---|---|---|---|---|---|---|---|---|---|---|---|---|
| Amazon Ratings | 24,492 | 186,100 | 300 | M-WE | 5 | 0.38 | 0.38 | 0.13 | 0.14 | 0.13 | 2 | last |
| Roman Empire | 22,662 | 65,854 | 300 | WE | 18 | 0.05 | 0.05 | 0.02 | -0.05 | 0.19 | 7 | last |
| Chameleon | 890 | 17,708 | 2,325 | BoW | 5 | 0.24 | 0.24 | 0.04 | 0.03 | 0.42 | 2 | first |
| Squirrel | 2,223 | 93,996 | 2,089 | BoW | 5 | 0.21 | 0.19 | 0.04 | 0.01 | 0.57 | 2 | first |
| CoraML | 2,995 | 16,316 | 2,879 | BoW | 7 | 0.79 | 0.81 | 0.74 | 0.75 | 0.45 | 3 | last |
| PubMed | 19,717 | 88,648 | 500 | TF/IDF-WF | 3 | 0.80 | 0.79 | 0.66 | 0.69 | 0.40 | 1 | last |

**Homophily.** We report different measures of homophily for these datasets that are frequently used in the literature (Lim et al., 2021; Luan et al., 2022; Platonov et al., 2024):

1. Edge Homophily $h_{\mathcal{E}}$ quantifies the fraction of edges with endpoints of the same class:

$$h_{\mathcal{E}} = \frac{|\{(u,v) \in \mathcal{E} \mid \boldsymbol{y}_u = \boldsymbol{y}_v\}|}{|\mathcal{E}|}. \tag{10}$$

2. Node Homophily $h_{\mathcal{V}}$ first computes how many neighbors of each node $v$ match $v$'s label and averages this quantity over all nodes:

$$h_{\mathcal{V}} = \frac{1}{|\mathcal{V}|} \sum_{v \in \mathcal{V}} \frac{|\{u \in \mathcal{N}(v) \mid \boldsymbol{y}_u = \boldsymbol{y}_v\}|}{|\mathcal{N}(v)|}. \tag{11}$$

3. Class Homophily measures the excess homophily of the graph compared to a randomly wired graph with the same number of nodes and edges and is less sensitive to the number of classes and their respective sizes (Lim et al., 2021):

$$h_C = \frac{1}{C-1} \sum_{c=1}^{C} \max \left\{ 0, \left[ \frac{\sum_{v \in \mathcal{V}, \boldsymbol{y}_v = c} |\{u \in \mathcal{N}(v) \mid \boldsymbol{y}_u = \boldsymbol{y}_v\}|}{\sum_{v \in \mathcal{V}, \boldsymbol{y}_v = c} |\mathcal{N}(v)|} - \frac{\{v \in \mathcal{V} \mid \boldsymbol{y}_v = c\}}{n} \right] \right\}. \tag{12}$$

4. Adjusted Homophily $h_{\boldsymbol{A}}$ as proposed by Platonov et al. (2024).

For all measures, high values indicate a high degree of homophily in the graph and low values represent heterophily. Additionally, we visualize the class compatibility matrices (Zhu et al., 2021) for each graph in Figure 5. These allow us to analyze homophily for each class. Each entry represents the fraction of edges that are incoming to nodes of a certain class, grouped by the class of the corresponding neighbor it originates from.

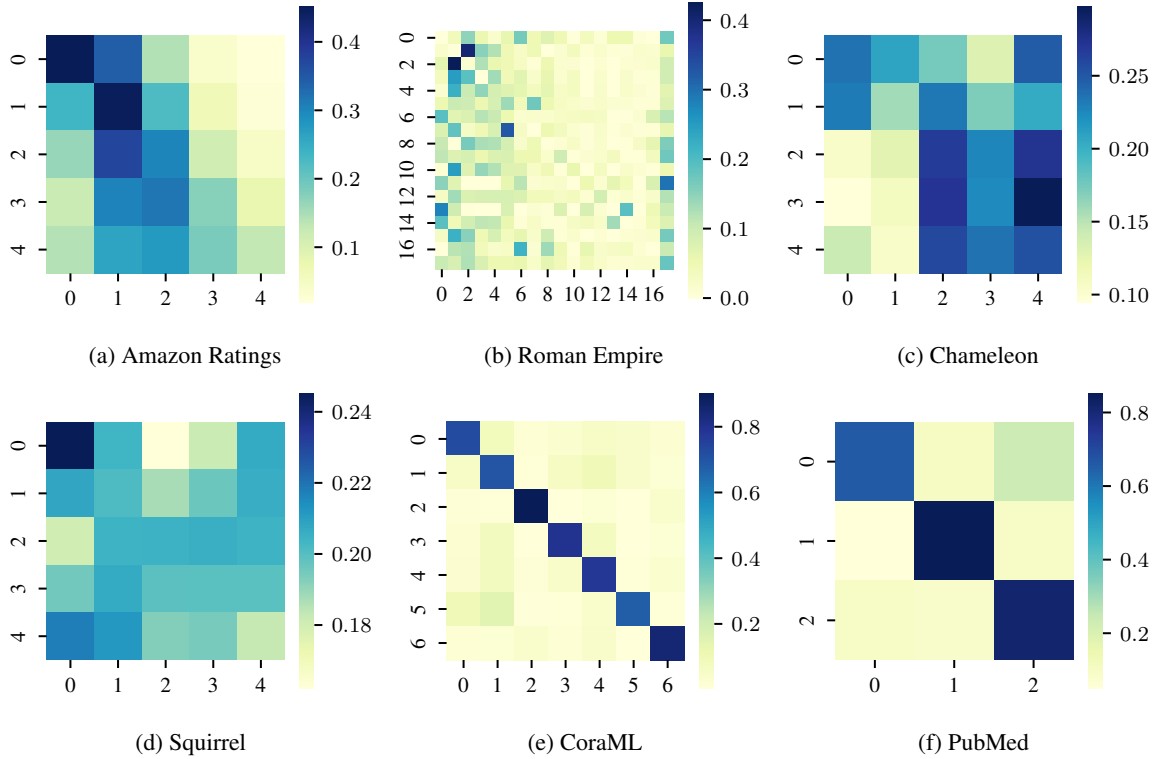

*Figure 5.* Compatibility matrices for each dataset.

**Distribution Shifts.** We study three distribution shifts to assess the quality of an uncertainty estimator.

1. Leave-out-Classes (LoC): We designate a subset of all classes as o.o.d. and remove all corresponding nodes from the training and validation set. At inference, these nodes are treated as o.o.d. To ensure that the left-out nodes are

semantically different from the i.d. nodes, we use the compatibility matrix in Figure 5. In particular, we select classes such that the rows of o.o.d. classes are dissimilar from rows of i.d. classes. For most datasets, selecting the $l$-last classes according to the enumeration of the data source meets this requirement. For the Chameleon and Squirrel datasets, we designate the $l$-first classes as o.o.d.

2. Near Feature Perturbations: We randomly sample nodes designated as o.o.d. and replace their features with noise at inference. For a distribution shift that is semantically similar to the training distribution, we match the data modality of the training set. That is, for datasets with Bag-of-Words (BoW) features, we use Bernoulli noise. For datasets with continuous features, we use Gaussian noise. Its mean and variance are the maximum likelihood estimates obtained on the clean data.

3. Far Feature Perturbations: Similar to the near feature perturbation shift, we replace the features of o.o.d. nodes with noise. Here, we use standard normal noise to increase dissimilarity from the training distribution.

### C.2. Models

While there is a plethora of advances in the development of models for heterophilic graphs, we base most of our evaluation on the recent study of Platonov et al. (2023). They find that under fair conditions regarding hyperparameter tuning, many architectures devised for heterophilic problems can not outperform GNNs for homophilic graphs. It is important to remark, however, that these homophilic baselines are only effective for heterophilic data when coupled with certain augmentations: (i) Both the input as well as the final node representations after $L$ layers are processed with additional 1-layer MLPs. (ii) The feature transformation after each aggregation step is realized with 2-layer MLPs instead of the commonly used affine transformations. (iii) Residual connections are implemented to enable information flow irrespective of the aggregation steps. (iv) Layer norm is applied before each aggregation step. Therefore, we realize our backbones with the same architectural choices that – importantly – deviate from what the corresponding works that introduce them suggest. We ablate models with and without residual connections. The architecture of models without residual connections can be described as follows:

$$\boldsymbol{H}_0' = \boldsymbol{X}\boldsymbol{W}_0$$
$$\boldsymbol{H}_0 = \sigma(\boldsymbol{H}_0')$$
$$\dots$$
$$\mathrm{AG}_l = \mathrm{AGGREGATE}^{(l)}(\boldsymbol{H}_{l-1}, \boldsymbol{A})$$
$$\boldsymbol{H}_l' = \mathrm{COMBINE}^{(l)}(\boldsymbol{H}_{l-1}, \mathrm{AG}_l)$$
$$\boldsymbol{H}_l = \sigma(\boldsymbol{H}_l')$$
$$\dots$$
$$\hat{P} = softmax(\boldsymbol{H}_L'\boldsymbol{W}_P)$$

The architecture of models with residual connections is described by:

$$\boldsymbol{H}_0' = \boldsymbol{X}\boldsymbol{W}_0$$
$$\boldsymbol{H}_0 = \sigma(\boldsymbol{H}_0')$$
$$\dots$$
$$\boldsymbol{H}_l'' = \mathrm{LayerNorm}(\boldsymbol{H}_{l-1})$$
$$\mathrm{AG}_l = \mathrm{AGGREGATE}^{(l)}(\boldsymbol{H}_l'', \boldsymbol{A})$$
$$\boldsymbol{H}_l' = \mathrm{COMBINE}^{(l)}(\boldsymbol{H}_l'', \mathrm{AG}_l)$$
$$\boldsymbol{H}_l = \boldsymbol{H}_{l-1} + \boldsymbol{H}_l'$$
$$\dots$$
$$\boldsymbol{H}_F = \mathrm{LayerNorm}(\boldsymbol{H}_L)$$
$$\hat{P} = softmax(\boldsymbol{H}_F'\boldsymbol{W}_P)$$

We study the following models as baselines/backbones in our work: A *GCN* (Kipf & Welling, 2017) and a *GATv2* (Veličković et al., 2018; Brody et al., 2022) with the aforementioned augmentations and no residual connections. Graph Posterior

Network (*GPN*) (Stadler et al., 2021) as an evidential method that uses a homophilic smoothing kernel to diffuse uncertainty. *SGCN* that uses the GCN architecture (without augmentations) for evidential learning coupled with student-teacher learning (Zhao et al., 2020). A Bayesian GCN (without augmentations) with Gaussian weights (Blundell et al., 2015). We apply the augmentations regarding residual connections to the GCN architecture (*Res-GCN*), an attention network that separates the own representation of each node from the aggregate of its neighbors as used in Platonov et al. (2023) (*Res-GAT-Sep*), and the recently proposed attention-based *Res-GATE* (Mustafa & Burkholz, 2024). For completeness, we also study three best-performing GNNs for heterophilic data in the study of Platonov et al. (2023): *GPRGNN* that generalizes Page-Rank diffusion to enabling high-frequency filters (Chien et al., 2021), *FAGCN* (Bo et al., 2021) that relies on a gating mechanism to account for low-frequency and high-frequency data, and *FSGNN* that aggregates information from all $k$-hop neighborhoods (Maurya et al., 2021).

### C.3. Experimental Setup

**Hyperparamter Tuning.** For a fair comparison, we tune all models over the same hyperparameter grid (Table 4) and select the best configuration in terms of validation accuracy for each model to conduct further experiments which we report in this work. If not stated otherwise, we use two layers in each model. The optimized configurations are supplied in our code.

| Hidden Dimension | Dropout | Learning Rate | Weight Decay |
|:---:|:---:|:---:|:---:|
| $\{64, 512\}$ | $\{0.2, 0.5\}$ | $\{0.01, 0.001\}$ | $\{0.0, 0.0001\}$ |

*Table 4.* Hyperparameter search space for all models if not stated otherwise.

**Training** for both the hyperparameter optimization as well as in the experiments we report in this work is done as follows: We average all results over 10 different dataset splits and 10 model initializations each. We use early stopping on the validation accuracy with a patience of 200 epochs. We optimize model weights using the ADAM optimizer (Kingma & Ba, 2014). Models are implemented in PyTorch (Paszke et al., 2017) and PyTorch Geometric (Fey & Lenssen, 2019) and trained on these types of machines: (i) Xeon E5-2630 v4 CPU @ 2.20GHz with a NVIDA GTX 1080TI GPU and 128 GB of RAM. (ii) AMD EPYC 7543 CPU @ 2.80GHz with a NVIDA A100 GPU and 128 GB of RAM .

**Uncertainty Estimation.** We use the following proxies for uncertainty estimation that closely follow Stadler et al. (2021):

- *Aleatoric uncertainty* is, in general, estimated from the maximal softmax score $\boldsymbol{p}_c$ each model predicts: $u^{\text{alea}} = 1 - \max_c \boldsymbol{p}_c$.

- For evidential models (GPN and SGCN), we estimate *epistemic uncertainty* from the overall evidence $\boldsymbol{\alpha}$: $u^{\text{epi}} = -\sum_c \boldsymbol{\alpha}_c$.

- For EBMs, we compute *epistemic uncertainty* from the logits $\boldsymbol{l}$ as: $u^{\text{epi}} = -\log \sum_c \exp(\boldsymbol{l}_c)$. For models that offer no explicit estimate of epistemic uncertainty (GCN, GATv2, FAGCN, FSGNN, GPRGNN), we report results regarding this estimator as a simple and highly applicable post-hoc proxy.

- For sampling-based methods (BGCN, Ensemble, MCD), we compute epistemic uncertainty from the variability of softmax scores $\boldsymbol{p}^{(i)}$: $u^{\text{epi}} = c^{-1} \sum_c \text{Var}_i \boldsymbol{p}_c^{(i)}$. For ensembles, we use 10 members. For BGCNs and MCD, we draw 50 samples during inference.

For JLDE, we use a KNN-based density estimate (Loftsgaarden & Quesenberry, 1965). For stability, we first use PCA to downproject each latent embedding $\boldsymbol{z}^{(i)}$ such that at least $95\%$ of the variance is preserved. We refer to these as $\tilde{\boldsymbol{z}}^{(i)}$. We then approximate the density from the $k$-nearest neighbors $\text{NN}^{(k)}(\boldsymbol{z}) \subset \mathcal{V}_{\text{train}}$ in the training set for each node $v$. We select $k = 5$ and compute:

$$u^{\text{epi}}(v) = -\log p(\mathcal{G}_v) \approx -\log p_\theta \left( \overset{L}{\underset{i=1}{\|}} \boldsymbol{z}_v^{(i)} \right) = \left( \frac{1}{k} \sum_{u \in \text{k-NN}(\|_{i=1}^L \tilde{\boldsymbol{z}}_v^{(i)})} \left| \left( \overset{L}{\underset{i=1}{\|}} \tilde{\boldsymbol{z}}_v^{(i)} \right) - \left( \overset{L}{\underset{i=1}{\|}} \tilde{\boldsymbol{z}}_u^{(i)} \right) \right|_2 \right)^2 . \quad (13)$$

Algorithm 1 algorithmically implements this KNN-based realization of JLDE.

---

**Algorithm 1** KNN-based JLDE

---

**Input:** Trained MPNN $f_\theta$, graph $\mathcal{G} = (\mathcal{V}, \mathcal{E})$, node features $\mathbf{X}$, set of training nodes, number of neighbors $k$
**Output:** Epistemic uncertainty $u(v) = -\log p_\theta(v) + C$ for a node $v \in \mathcal{V} \setminus \mathcal{V}_{\text{train}}$

1: $\mathbf{H}^{(1)}, \ldots, \mathbf{H}^{(L)} = f_\theta(\mathcal{G}, \mathbf{X})$ {Obtain latent node representations}
2: $\mathbf{H}^{(0)} \leftarrow \mathbf{X}$
3: **for all** $i \in \{0, \ldots, L\}$ **do**
4:    $\hat{\mathbf{H}}^{(i)} \leftarrow \text{PCA}_{0.95}(\mathbf{H}^{(i)})$ {Optional Dimensionality Reduction}
5: **end for**
6: $\mathbf{Z} \leftarrow \|_{i=0}^{L} \hat{\mathbf{H}}^{(i)}$
7: $u(v) \leftarrow 0$
8: **for all** $t \in \text{NN}^{(k)}(v, \mathcal{V}_{\text{train}})$ **do**
9:    $u(v) \leftarrow u(v) + \|\mathbf{Z}_v - \mathbf{Z}_t\|_2^2$ {Omit normalizer}
10: **end for**
11: **return** $u(v)$

---

The non-concatenated version we ablate is given by:

$$u^{\text{epi}}(v) = -\log p(\mathcal{G}_v) \approx -\log p_\theta\left(\mathbf{z}_v^{(1:L)}\right) = \sum_{i=1}^{L} \left( \frac{1}{k} \sum_{u \in \text{NN}^{(k)}(\tilde{\mathbf{z}}_v^{(i)})} \left| \left(\tilde{\mathbf{z}}_v^{(i)}\right) - \left(\tilde{\mathbf{z}}_u^{(i)}\right) \right|_2 \right)^2. \quad (14)$$

## D. Benchmarking Heterophilic GNNs

We benchmark the models described in Appendix C.2 in terms of accuracy on all six datasets (without distribution shifts) and report results in Table 5. We confirm the trends observed by Platonov et al. (2023): Many of the GNNs designated for heterophilic graphs fall short of the augmented homophilic models. In particular, all residual GNN variants (Res-GCN, Res-GATE, Res-GAT-Sep) consistently perform well across different datasets. GPRGNN is the most consistent model explicitly designed to accommodate heterophilic data. Consequently, we utilize these as backbones in our experiments.

*Table 5.* Accuracy of different models on clean data (best and runner-up).

| Model | Roman Empire | Amazon Ratings | Chameleon | Squirrel | CoraML | PubMed |
|---|---|---|---|---|---|---|
| GCN | $62.7_{\pm 0.8}$ | $50.1_{\pm 0.5}$ | $41.6_{\pm 3.2}$ | $41.6_{\pm 2.1}$ | $78.7_{\pm 2.1}$ | $85.8_{\pm 0.3}$ |
| GATv2 | $87.1_{\pm 0.5}$ | $51.0_{\pm 0.6}$ | $41.8_{\pm 3.2}$ | $34.7_{\pm 1.9}$ | $78.7_{\pm 2.2}$ | $85.2_{\pm 0.4}$ |
| GPN | $40.8_{\pm 1.5}$ | $47.4_{\pm 0.7}$ | $37.2_{\pm 4.0}$ | $34.6_{\pm 1.6}$ | $81.3_{\pm 1.9}$ | $84.8_{\pm 0.5}$ |
| FAGCN | $74.9_{\pm 0.8}$ | $52.3_{\pm 0.6}$ | $39.5_{\pm 3.6}$ | $40.6_{\pm 2.9}$ | $82.5_{\pm 1.9}$ | $86.4_{\pm 0.3}$ |
| FSGNN | $79.7_{\pm 0.5}$ | $51.4_{\pm 0.6}$ | $36.9_{\pm 2.9}$ | $36.0_{\pm 2.1}$ | $75.3_{\pm 1.2}$ | $85.7_{\pm 0.4}$ |
| GPRGNN | $71.0_{\pm 0.5}$ | $52.0_{\pm 0.7}$ | $37.4_{\pm 3.3}$ | $35.3_{\pm 1.3}$ | $84.6_{\pm 1.4}$ | $86.6_{\pm 0.3}$ |
| SGCN | $50.9_{\pm 1.8}$ | $48.9_{\pm 0.4}$ | $39.0_{\pm 3.5}$ | $36.9_{\pm 1.6}$ | $82.6_{\pm 1.4}$ | $86.3_{\pm 0.3}$ |
| BGCN | $46.6_{\pm 0.6}$ | $44.9_{\pm 1.2}$ | $39.0_{\pm 3.4}$ | $34.1_{\pm 2.6}$ | $82.0_{\pm 1.4}$ | $86.0_{\pm 0.3}$ |
| Res-GCN | $80.7_{\pm 0.7}$ | $49.8_{\pm 0.5}$ | $41.8_{\pm 3.8}$ | $41.9_{\pm 2.1}$ | $80.4_{\pm 1.5}$ | $85.7_{\pm 0.3}$ |
| Res-GATE | $87.2_{\pm 0.6}$ | $52.4_{\pm 0.7}$ | $37.7_{\pm 3.5}$ | $34.2_{\pm 1.4}$ | $81.3_{\pm 1.5}$ | $86.3_{\pm 0.3}$ |
| Res-GAT-Sep | $88.0_{\pm 0.6}$ | $53.6_{\pm 0.5}$ | $34.9_{\pm 4.0}$ | $34.9_{\pm 1.6}$ | $79.3_{\pm 1.7}$ | $85.5_{\pm 0.3}$ |

## E. Additional Results

### E.1. Runtime of JLDE

The computational cost of KNN-based JLDE is governed by KNN, which can be implemented in $\mathcal{O}(n_{\text{train}} * d_{\text{hidden}} + n_{\text{train}} * k)$. For smaller training sets that are typical in transductive node classification, this is negligible but can become prohibitive for large training sets. In this case, JLDE can be realized with more efficient density estimators. We opt for KNN due to

its simplicity and applicability to estimating a high dimensional density from little d. However, in general, any density estimator can be used. We measure the runtime of KNN-based JLDE in Table 6. For all datasets with small training sets (all but Amazon Ratings), the cost of JLDE is comparable to competitors. Using multiple layers effectively increases $d_{\text{hidden}}$ by a factor of $L$ in terms of runtime complexity. This only incurs a small additional cost (see comparison to 1-layer KNN-based JLDE in Table 6.

The memory complexity is also determined by the density estimator (KNN) and amounts to keeping the embeddings of the training set in the memory, $\mathcal{O}(n_{\text{train}} * d_{\text{hidden}})$. Since by default, these activations are kept in memory for a forward pass unless explicitly deallocated, there is no overhead in practice.

*Table 6.* Runtime (s) for different uncertainty estimators averaged over 25 runs, including a variation of JLDE (ours) that estimates uncertainty from one layer only (JLDE-1L).

| **Uncertainty** | Roman Empire | Amazon Ratings | Chame-leon | Squirrel | CoraML | PubMed |
|---|---|---|---|---|---|---|
| JLDE | $1.50_{\pm 0.0}$ | $5.32_{\pm 0.0}$ | $0.17_{\pm 0.1}$ | $0.28_{\pm 0.1}$ | $0.27_{\pm 0.1}$ | $0.71_{\pm 0.1}$ |
| JLDE-1L | $1.36_{\pm 0.1}$ | $4.04_{\pm 0.1}$ | $0.13_{\pm 0.0}$ | $0.28_{\pm 0.1}$ | $0.23_{\pm 0.0}$ | $0.71_{\pm 0.0}$ |
| GEBM | $0.73_{\pm 0.1}$ | $5.96_{\pm 0.1}$ | $0.70_{\pm 0.2}$ | $1.08_{\pm 0.2}$ | $0.66_{\pm 0.1}$ | $1.38_{\pm 0.2}$ |
| MCD | $5.91_{\pm 1.1}$ | $163.83_{\pm 42.8}$ | $4.95_{\pm 0.6}$ | $11.84_{\pm 1.3}$ | $5.89_{\pm 0.2}$ | $15.61_{\pm 0.6}$ |
| Ensemble | $1.21_{\pm 0.0}$ | $47.54_{\pm 1.1}$ | $1.53_{\pm 0.1}$ | $1.89_{\pm 0.1}$ | $1.53_{\pm 0.2}$ | $3.04_{\pm 0.3}$ |
| Energy | $0.08_{\pm 0.0}$ | $5.13_{\pm 0.3}$ | $0.21_{\pm 0.0}$ | $0.34_{\pm 0.0}$ | $0.15_{\pm 0.0}$ | $0.41_{\pm 0.1}$ |
| GPN | $0.32_{\pm 0.0}$ | $1.50_{\pm 0.2}$ | $0.46_{\pm 0.0}$ | $0.75_{\pm 0.1}$ | $0.64_{\pm 0.1}$ | $1.12_{\pm 0.2}$ |
| BGCN | $5.64_{\pm 0.3}$ | $20.06_{\pm 0.6}$ | $3.30_{\pm 0.2}$ | $8.56_{\pm 0.1}$ | $4.16_{\pm 0.1}$ | $10.58_{\pm 0.1}$ |
| SGCN | $0.07_{\pm 0.0}$ | $0.54_{\pm 0.0}$ | $0.10_{\pm 0.0}$ | $0.27_{\pm 0.0}$ | $0.14_{\pm 0.0}$ | $0.32_{\pm 0.0}$ |

## E.2. Out-of-Distribution Detection

We report o.o.d. detection performance of all baselines and JLDE with a Res-GCN backbone for all datasets and distribution shifts in Tables 7 and 8. JLDE performs the best on the high-quality novel Roman Empire and Amazon Ratings datasets, and is the most consistent post-hoc method for Squirrel and Chameleon, even though sometimes other methods outperform it on occasions. On homophilic data, JLDE performs comparable to other post-hoc estimators but is outperformed by methods dedicated to homophily (e.g. GPN, SGCN) for the LoC and near feature perturbations shifts. On far feature perturbations, it benefits from a density-based estimate and performs best.

**Backbones.** We compare JLDE to other model-agnostic estimators on four different backbones that perform the best in terms of accuracy according to Appendix D: Res-GCN, Res-GAT-Sep, Res-GATE, and GPRGNN. Tables 9 to 11 show that JLDE is the only method that consistently performs well on all distribution shifts. This is also reflected by its average rank among these estimators: In Tables 2 and 12 ranks all estimators per dataset and distribution shift and averages them over the three distribution shifts such that LoC and both feature perturbations are weighted equally (i.e. weights of $50\%/25\%/25\%$). Table 12 shows the average rank ($\downarrow$) only among epistemic estimators and also the rank among both epistemic and aleatoric estimators. JLDE consistently achieves the best or second-best rank on heterophilic data and performs competitively on homophilic datasets.

## E.3. Misclassification Detection

We report the misclassification AUC-ROC and AUC-PR metrics in Tables 13 and 14. Consistent with prior work (Stadler et al., 2021; Fuchsgruber et al., 2024b), we find that aleatoric uncertainty often is the better proxy to detect erroneous predictions. As remarked in Section 5.2, JLDE is a model-agnostic post-hoc estimator that leaves the softmax predictions of the backbone model unaltered. This permits improvements regarding aleatoric uncertainty which is, however, beyond the scope of this work. JLDE performs similarly to other epistemic estimators in terms of misclassification detection.

## E.4. Calibration

We report the Expected Calibration Error (ECE) ($\downarrow$) (Naeini et al., 2015) and Brier Score ($\downarrow$) (Brier, 1950) to assess the calibration of the backbones on each dataset without any distribution shifts in Tables 15 and 16. Again, the calibration

depends on the softmax prediction of the model which is unaltered by post-hoc methods like JLDE. Improving calibration is therefore an orthogonal avenue for future work on heterophilic graphs.

Like other epistemic uncertainty estimators (e.g. GPN (Stadler et al., 2021) or EBMs (Fuchsgruber et al., 2024b; Wu et al., 2023)), JLDE's epistemic uncertainty estimate is not normalized to $[0, 1]$ which prohibits measures like ECE. Instead, we visualize how the confidence (i.e. the inverse uncertainty) of JLDE correlates with the model accuracy. To that end, we normalize the confidence of different runs to $[0, 1]$ and compute the average model accuracy on the test set over 20 bins.

Figure 6 shows that except the Chameleon and Squirrel datasets, JLDE outputs confidence that is well aligned with predictive performance. We suspect that for the two datasets where this correlation is less pronounced, the poor model accuracy overall inhibits uncertainty calibration.

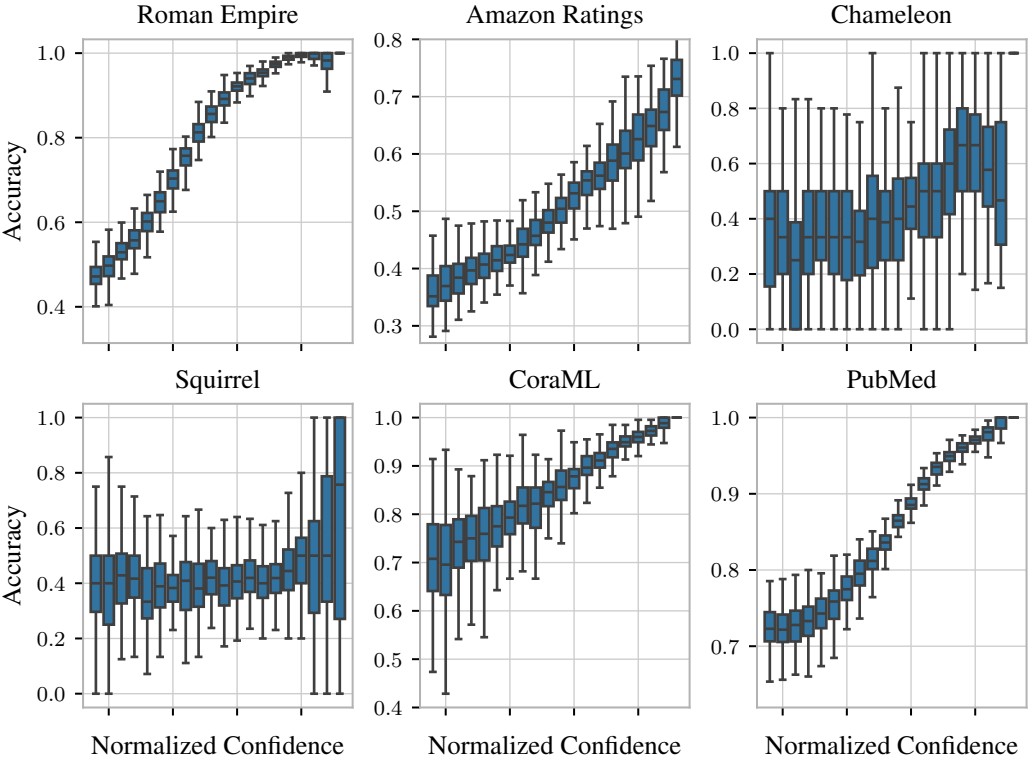

*Figure 6.* Average accuracy of the Res-GCN backbone binned by JLDE's (ours) epistemic confidence normalized to $[0, 1]$.

### E.5. Varying Degree of Homophily

We also study how the degree of homophily affects JLDE's performance compared to GEBM (Fuchsgruber et al., 2024b), an uncertainty estimator devised for homophilic data. To that end, we create synthetic data similar to Das et al. (2025). In addition to the two classes from the moons dataset, we create an additional anomalous class where node features are sampled independently from a 2-dimensional normal at $[1, 1]^T$ with an isotropic diagonal variance of $0.1$. The connectivity pattern follows the same procedure as in Das et al. (2025). We do not train the model on nodes from this anomalous class and report the o.o.d. detection AUC-ROC for JLDE and GEBM on settings with varying node homophily $h$ that is measured as the fraction of intra-class edges for each node.

Figure 7 shows that while the state-of-the-art estimator for homophilic graph GEBM deteriorates under increasing heterophily, JLDE maintains its performance even for highly heterophilic graphs.

### E.6. Ablations

We supply the ablation regarding the informativeness of individual latent representations $Z^{(i)}$ in Section 5.3 on the other backbones (Res-GAT-Sep, Res-GATE, GPRGNN) in Figures 8 to 10. Similar to Figure 4, we find that on heterophilic graphs, different latent representations contain different information while on the homophilic CoraML, deeper layers amplify the information of more shallow representations. Consequently, JLDE provides better uncertainty on heterophilic graphs

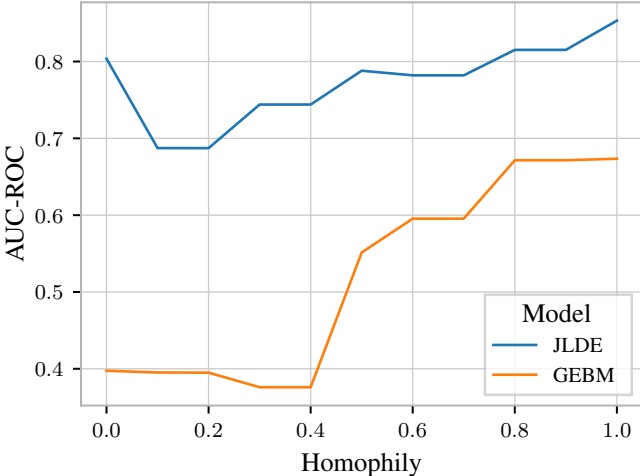

Figure 7. Average o.o.d. detection AUC-ROC of JLDE (ours) and the best-performing uncertainty estimator for homophilic graphs (GEBM) at different degrees of label homophily on synthetic graphs and a leave-out-classes distribution shift.

Figure 8. O.o.d. detection AUC-ROC (↑) under a leave-out-classes distribution shift when estimating the density from different layers of a Res-GAT-Sep backbone.

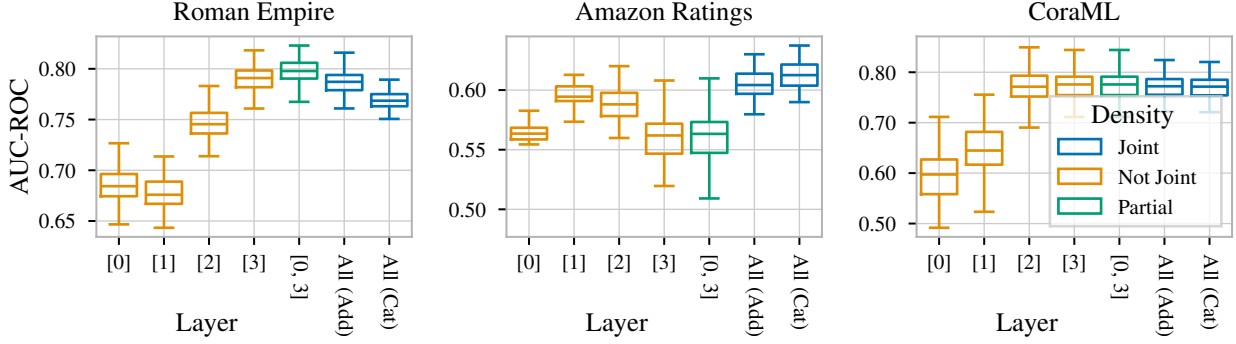

through jointly considering all $Z^{(i)}$ instead of relying on the final or penultimate representations. This holds over different backbone models, confirming the generality of our analysis.

### E.6.1. DENSITY ESTIMATORS

We also ablate JLDE using different density estimators in Table 17. In addition to KNN, we also ablate two other simple estimators: (i) A mixture of Gaussians (MoG) where the means and diagonal variances correspond to the maximum likelihood estimates for all concatenated node embeddings of each class in the training dataset separately. (ii) A RBF-based kernel density estimator (KDE) that is computed as the average RBF-distance to each concatenated training node embedding. KDE performs similarly to JLDE but MoG falls short in some settings because of its limited expressiveness.

*Figure 9.* O.o.d. detection AUC-ROC (↑) under a leave-out-classes distribution shift when estimating the density from different layers of a Res-GATE backbone.

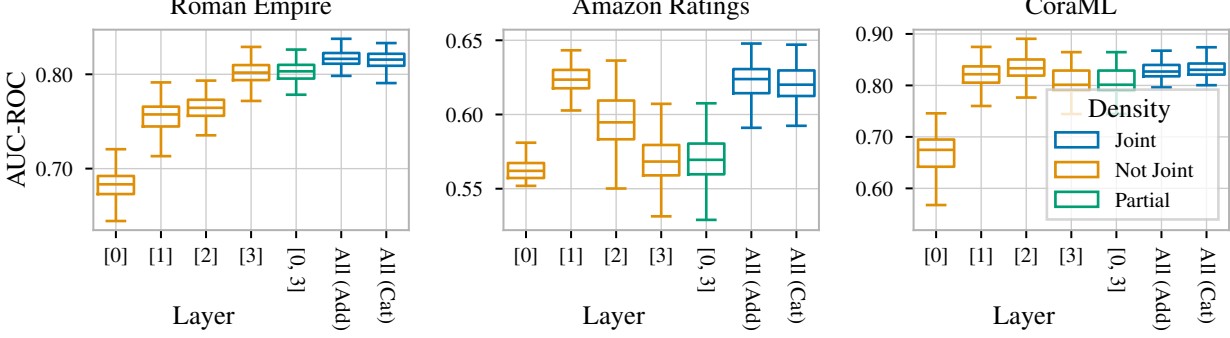

*Figure 10.* O.o.d. detection AUC-ROC (↑) under a leave-out-classes distribution shift when estimating the density from different layers of a GPRGNN backbone.

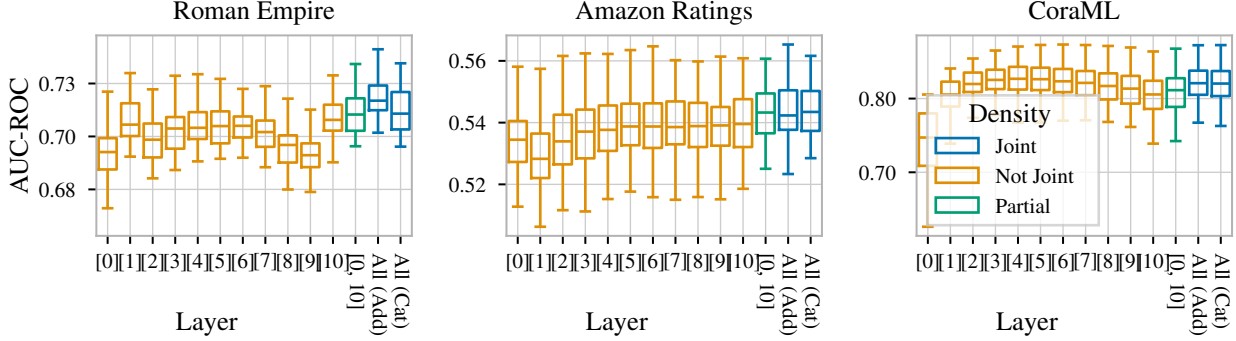

*Table 17.* O.o.d. detection AUC for JLDE (ours) using different density estimators and a GATE backbone for different distribution shifts (**bold** numbers indicate best performance and **OOM** out-of-memory).

|  | **Density** | LoC | Near-Features | Far-Features |
|---|---|---|---|---|
| **Roman Empire** | KNN | $81.5_{\pm 0.9}$ | $88.8_{\pm 0.9}$ | $\mathbf{100.0}_{\pm 0.0}$ |
|  | MoG | $74.6_{\pm 0.9}$ | $87.8_{\pm 1.0}$ | $\mathbf{100.0}_{\pm 0.0}$ |
|  | KDE | $\mathbf{81.6}_{\pm 0.9}$ | $\mathbf{89.5}_{\pm 0.9}$ | $\mathbf{100.0}_{\pm 0.0}$ |
| **Amazon Ratings** | KNN | $62.1_{\pm 1.2}$ | $\mathbf{63.5}_{\pm 2.0}$ | $93.9_{\pm 0.9}$ |
|  | MoG | $57.0_{\pm 1.3}$ | $58.3_{\pm 1.4}$ | $\mathbf{100.0}_{\pm 0.0}$ |
|  | KDE | $\mathbf{63.5}_{\pm 1.1}$ | OOM | OOM |
| **Chameleon** | KNN | $62.8_{\pm 6.6}$ | $\mathbf{51.3}_{\pm 7.2}$ | $99.5_{\pm 1.2}$ |
|  | MoG | $\mathbf{67.1}_{\pm 6.2}$ | $34.1_{\pm 11.3}$ | $\mathbf{100.0}_{\pm 0.0}$ |
|  | KDE | $62.6_{\pm 6.6}$ | $51.2_{\pm 7.5}$ | $99.6_{\pm 1.2}$ |
| **Squirrel** | KNN | $62.5_{\pm 7.8}$ | $\mathbf{38.4}_{\pm 6.5}$ | $\mathbf{100.0}_{\pm 0.1}$ |
|  | MoG | $\mathbf{68.7}_{\pm 8.8}$ | $25.3_{\pm 6.6}$ | $\mathbf{100.0}_{\pm 0.0}$ |
|  | KDE | $62.1_{\pm 8.4}$ | $\mathbf{38.4}_{\pm 6.8}$ | $\mathbf{100.0}_{\pm 0.1}$ |
| **CoraML** | KNN | $83.2_{\pm 1.8}$ | $54.0_{\pm 2.3}$ | $95.1_{\pm 2.2}$ |
|  | MoG | $\mathbf{83.4}_{\pm 2.4}$ | $\mathbf{55.4}_{\pm 2.5}$ | $\mathbf{100.0}_{\pm 0.0}$ |
|  | KDE | $83.3_{\pm 2.0}$ | $54.1_{\pm 2.3}$ | $95.7_{\pm 2.2}$ |
| **PubMed** | KNN | $\mathbf{63.3}_{\pm 2.2}$ | $\mathbf{75.0}_{\pm 4.2}$ | $99.9_{\pm 0.0}$ |
|  | MoG | $61.3_{\pm 2.9}$ | $62.0_{\pm 2.9}$ | $\mathbf{100.0}_{\pm 0.0}$ |
|  | KDE | $62.8_{\pm 2.4}$ | $73.8_{\pm 4.2}$ | $99.9_{\pm 0.0}$ |

*Table 7.* O.o.d. detection AUC-ROC using different uncertainty estimators and our method using a Res-GCN backbone.

| | Model | LoC AUC-ROC (Alea. / Epi.) | Acc.↑ | Near-Features AUC-ROC (Alea. / Epi.) | Acc.↑ | Far-Features AUC-ROC (Alea. / Epi.) | Acc.↑ |
|---|---|---|---|---|---|---|---|
| Roman Empire | GPN | $60.3_{\pm3.1}$/$64.0_{\pm2.8}$ | $48.6_{\pm1.9}$ | $59.9_{\pm2.0}$/$66.4_{\pm2.1}$ | $38.1_{\pm1.3}$ | $54.8_{\pm1.5}$/$77.3_{\pm1.6}$ | $38.0_{\pm1.3}$ |
| | SGCN | $60.0_{\pm0.7}$/$56.5_{\pm1.2}$ | $58.3_{\pm1.6}$ | $55.7_{\pm1.7}$/$54.1_{\pm2.4}$ | $45.5_{\pm1.6}$ | $12.5_{\pm0.5}$/$7.1_{\pm0.5}$ | $30.6_{\pm1.0}$ |
| | BGCN | $54.9_{\pm1.8}$/$60.7_{\pm3.2}$ | $53.0_{\pm0.7}$ | $58.1_{\pm1.2}$/$46.2_{\pm2.1}$ | $42.4_{\pm0.5}$ | $38.3_{\pm4.1}$/$73.6_{\pm2.4}$ | $28.8_{\pm0.6}$ |
| | GPRGNN | $66.5_{\pm0.9}$/$63.4_{\pm1.0}$ | $78.1_{\pm0.4}$ | $86.5_{\pm0.9}$/$87.9_{\pm2.6}$ | $64.8_{\pm0.6}$ | $17.6_{\pm1.3}$/$79.1_{\pm4.9}$ | $61.0_{\pm0.6}$ |
| | FSGNN | $73.0_{\pm1.1}$/$70.2_{\pm1.7}$ | $86.6_{\pm0.4}$ | $70.5_{\pm0.9}$/$76.3_{\pm1.6}$ | $71.6_{\pm0.5}$ | $21.9_{\pm1.4}$/$3.7_{\pm0.8}$ | $56.9_{\pm0.9}$ |
| | FAGCN | $72.7_{\pm1.3}$/$68.8_{\pm1.9}$ | $83.6_{\pm1.1}$ | $77.8_{\pm1.5}$/$85.6_{\pm1.7}$ | $68.0_{\pm0.9}$ | $23.9_{\pm5.5}$/$11.2_{\pm6.5}$ | $44.3_{\pm2.8}$ |
| | Ens. | $76.4_{\pm0.8}$/$76.2_{\pm0.7}$ | | $78.4_{\pm0.6}$/$78.2_{\pm0.7}$ | | $89.2_{\pm0.8}$/$86.9_{\pm0.7}$ | |
| | MCD | $76.3_{\pm1.0}$/$75.9_{\pm0.9}$ | | $75.8_{\pm1.0}$/$72.5_{\pm1.0}$ | | $82.7_{\pm0.8}$/$72.9_{\pm1.8}$ | |
| | EBM | $73.3_{\pm1.3}$/$72.5_{\pm1.6}$ | | $74.5_{\pm1.0}$/$81.3_{\pm0.9}$ | | $83.0_{\pm0.8}$/$92.0_{\pm1.1}$ | |
| | Safe | $73.3_{\pm1.3}$/$64.8_{\pm2.1}$ | $87.6_{\pm0.7}$ | $74.5_{\pm1.0}$/$73.4_{\pm1.3}$ | $72.3_{\pm0.6}$ | $83.0_{\pm0.8}$/$81.0_{\pm1.3}$ | $70.6_{\pm0.6}$ |
| | GEBM | $73.3_{\pm1.3}$/$53.6_{\pm3.8}$ | | $74.5_{\pm1.0}$/$66.4_{\pm3.0}$ | | $83.0_{\pm0.8}$/$80.9_{\pm3.3}$ | |
| | JLDE | $73.3_{\pm1.3}$/$76.9_{\pm1.4}$ | | $74.5_{\pm1.0}$/$93.8_{\pm0.9}$ | | $83.0_{\pm0.8}$/$100.0_{\pm0.0}$ | |
| Amazon Ratings | GPN | $48.7_{\pm1.5}$/$51.5_{\pm4.1}$ | $54.2_{\pm1.2}$ | $54.3_{\pm1.3}$/$52.4_{\pm2.2}$ | $46.8_{\pm0.7}$ | $52.6_{\pm1.0}$/$58.5_{\pm1.8}$ | $47.0_{\pm0.7}$ |
| | SGCN | $47.8_{\pm1.9}$/$46.6_{\pm1.8}$ | $55.5_{\pm0.7}$ | $54.0_{\pm1.0}$/$54.2_{\pm1.0}$ | $47.2_{\pm0.5}$ | $27.1_{\pm1.3}$/$22.8_{\pm2.1}$ | $33.5_{\pm1.3}$ |
| | BGCN | $52.7_{\pm1.4}$/$49.0_{\pm3.2}$ | $51.4_{\pm0.8}$ | $54.1_{\pm1.4}$/$46.6_{\pm1.7}$ | $44.2_{\pm1.3}$ | $44.5_{\pm4.4}$/$72.1_{\pm3.0}$ | $33.5_{\pm1.3}$ |
| | GPRGNN | $52.5_{\pm1.2}$/$52.1_{\pm1.4}$ | $57.9_{\pm0.6}$ | $62.1_{\pm1.2}$/$70.0_{\pm1.3}$ | $50.6_{\pm0.7}$ | $10.9_{\pm3.1}$/$48.6_{\pm10.9}$ | $34.1_{\pm2.7}$ |
| | FSGNN | $54.4_{\pm1.2}$/$53.6_{\pm1.3}$ | $56.8_{\pm0.6}$ | $45.1_{\pm1.3}$/$38.6_{\pm1.2}$ | $49.1_{\pm0.6}$ | $12.7_{\pm2.5}$/$2.2_{\pm1.2}$ | $38.6_{\pm0.8}$ |
| | FAGCN | $53.1_{\pm1.4}$/$53.3_{\pm1.4}$ | $58.1_{\pm0.8}$ | $59.8_{\pm1.3}$/$62.9_{\pm1.7}$ | $51.2_{\pm0.7}$ | $17.9_{\pm2.2}$/$12.0_{\pm3.7}$ | $31.7_{\pm2.4}$ |
| | Ens. | $51.4_{\pm0.8}$/$57.9_{\pm0.8}$ | | $60.3_{\pm1.6}$/$63.6_{\pm1.4}$ | | $74.3_{\pm0.9}$/$50.4_{\pm2.1}$ | |
| | MCD | $49.2_{\pm1.1}$/$53.5_{\pm1.3}$ | | $56.6_{\pm2.1}$/$58.2_{\pm1.9}$ | | $72.9_{\pm1.0}$/$33.2_{\pm4.8}$ | |
| | EBM | $48.8_{\pm1.2}$/$48.9_{\pm3.5}$ | | $54.8_{\pm1.9}$/$56.1_{\pm2.3}$ | | $78.2_{\pm2.1}$/$94.4_{\pm1.2}$ | |
| | Safe | $48.8_{\pm1.2}$/$48.0_{\pm4.2}$ | $56.0_{\pm0.6}$ | $54.8_{\pm1.9}$/$55.3_{\pm2.4}$ | $48.2_{\pm0.5}$ | $78.2_{\pm2.1}$/$67.2_{\pm2.5}$ | $47.3_{\pm0.6}$ |
| | GEBM | $48.8_{\pm1.2}$/$49.9_{\pm3.5}$ | | $54.8_{\pm1.9}$/$47.1_{\pm1.5}$ | | $78.2_{\pm2.1}$/$58.9_{\pm7.4}$ | |
| | JLDE | $48.8_{\pm1.2}$/$61.2_{\pm0.8}$ | | $54.8_{\pm1.9}$/$65.4_{\pm1.3}$ | | $78.2_{\pm2.1}$/$99.2_{\pm0.2}$ | |
| Chameleon | GPN | $48.6_{\pm8.9}$/$71.6_{\pm9.4}$ | $39.6_{\pm3.8}$ | $57.8_{\pm7.6}$/$54.0_{\pm6.2}$ | $37.0_{\pm3.8}$ | $53.3_{\pm6.7}$/$62.1_{\pm5.0}$ | $37.0_{\pm3.8}$ |
| | SGCN | $48.6_{\pm10.3}$/$61.6_{\pm7.0}$ | $43.3_{\pm4.2}$ | $55.6_{\pm6.4}$/$54.9_{\pm9.7}$ | $38.2_{\pm3.1}$ | $22.2_{\pm6.3}$/$17.2_{\pm4.9}$ | $25.2_{\pm4.8}$ |
| | BGCN | $45.4_{\pm8.5}$/$36.6_{\pm4.8}$ | $40.5_{\pm4.0}$ | $56.2_{\pm5.4}$/$52.9_{\pm4.8}$ | $38.3_{\pm3.3}$ | $27.4_{\pm6.3}$/$36.0_{\pm10.5}$ | $26.9_{\pm4.8}$ |
| | GPRGNN | $51.6_{\pm7.1}$/$48.8_{\pm7.7}$ | $36.8_{\pm4.6}$ | $62.9_{\pm7.2}$/$60.8_{\pm6.5}$ | $37.1_{\pm3.6}$ | $15.9_{\pm7.6}$/$7.9_{\pm7.2}$ | $30.5_{\pm4.1}$ |
| | FSGNN | $34.0_{\pm6.2}$/$46.2_{\pm7.9}$ | $43.9_{\pm4.2}$ | $61.8_{\pm7.7}$/$62.9_{\pm5.8}$ | $36.9_{\pm3.6}$ | $7.5_{\pm3.3}$/$4.1_{\pm4.4}$ | $30.5_{\pm3.7}$ |
| | FAGCN | $42.1_{\pm6.4}$/$43.9_{\pm6.2}$ | $43.2_{\pm6.4}$ | $66.9_{\pm6.6}$/$70.1_{\pm8.7}$ | $39.7_{\pm3.6}$ | $13.8_{\pm5.5}$/$3.6_{\pm2.3}$ | $30.2_{\pm4.1}$ |
| | Ens. | $34.4_{\pm6.3}$/$49.9_{\pm11.4}$ | | $55.2_{\pm5.4}$/$53.8_{\pm5.8}$ | | $81.2_{\pm6.9}$/$51.0_{\pm10.9}$ | |
| | MCD | $35.6_{\pm7.3}$/$40.1_{\pm5.4}$ | | $56.0_{\pm5.9}$/$55.2_{\pm7.0}$ | | $82.2_{\pm9.1}$/$41.1_{\pm10.8}$ | |
| | EBM | $35.3_{\pm8.2}$/$35.7_{\pm8.8}$ | | $54.6_{\pm5.8}$/$55.9_{\pm6.3}$ | | $78.4_{\pm8.9}$/$92.5_{\pm7.1}$ | |
| | Safe | $35.3_{\pm8.2}$/$36.6_{\pm8.5}$ | $47.8_{\pm4.9}$ | $54.6_{\pm5.8}$/$55.7_{\pm7.6}$ | $40.7_{\pm3.9}$ | $78.4_{\pm8.9}$/$68.6_{\pm7.8}$ | $40.1_{\pm3.3}$ |
| | GEBM | $35.3_{\pm8.2}$/$66.4_{\pm9.1}$ | | $54.6_{\pm5.8}$/$37.3_{\pm14.4}$ | | $78.4_{\pm8.9}$/$85.9_{\pm7.3}$ | |
| | JLDE | $35.3_{\pm8.2}$/$61.5_{\pm5.8}$ | | $54.6_{\pm5.8}$/$51.1_{\pm7.6}$ | | $78.4_{\pm8.9}$/$99.9_{\pm0.2}$ | |
| Squirrel | GPN | $51.5_{\pm8.7}$/$73.5_{\pm2.4}$ | $41.1_{\pm2.9}$ | $54.7_{\pm8.7}$/$50.8_{\pm4.9}$ | $34.5_{\pm1.6}$ | $54.9_{\pm11.5}$/$62.0_{\pm6.3}$ | $34.6_{\pm1.6}$ |
| | SGCN | $40.5_{\pm4.0}$/$67.2_{\pm4.8}$ | $47.1_{\pm2.9}$ | $56.5_{\pm4.7}$/$56.3_{\pm5.4}$ | $37.7_{\pm1.7}$ | $29.9_{\pm5.3}$/$24.9_{\pm6.4}$ | $28.2_{\pm3.8}$ |
| | BGCN | $50.4_{\pm8.2}$/$24.3_{\pm2.1}$ | $43.2_{\pm4.4}$ | $53.8_{\pm4.8}$/$49.1_{\pm3.8}$ | $33.9_{\pm3.1}$ | $32.2_{\pm6.8}$/$65.5_{\pm7.9}$ | $30.2_{\pm3.0}$ |
| | GPRGNN | $48.3_{\pm16.0}$/$48.1_{\pm14.6}$ | $40.9_{\pm3.6}$ | $53.1_{\pm4.9}$/$57.0_{\pm6.9}$ | $35.4_{\pm1.4}$ | $14.7_{\pm11.3}$/$18.4_{\pm17.6}$ | $34.4_{\pm1.9}$ |
| | FSGNN | $31.3_{\pm4.7}$/$32.8_{\pm8.3}$ | $45.1_{\pm3.2}$ | $56.7_{\pm5.8}$/$55.9_{\pm5.2}$ | $36.1_{\pm1.7}$ | $8.1_{\pm2.4}$/$6.2_{\pm2.5}$ | $33.9_{\pm2.4}$ |
| | FAGCN | $39.6_{\pm8.6}$/$44.6_{\pm8.0}$ | $47.0_{\pm6.5}$ | $54.5_{\pm6.1}$/$54.3_{\pm5.4}$ | $40.5_{\pm2.8}$ | $12.1_{\pm7.3}$/$10.2_{\pm8.3}$ | $35.3_{\pm2.7}$ |
| | Ens. | $29.5_{\pm6.5}$/$44.9_{\pm4.8}$ | | $53.0_{\pm3.9}$/$50.6_{\pm5.4}$ | | $71.0_{\pm2.2}$/$70.1_{\pm13.6}$ | |
| | MCD | $32.0_{\pm7.4}$/$37.2_{\pm5.3}$ | | $52.7_{\pm4.1}$/$51.2_{\pm4.8}$ | | $70.1_{\pm4.5}$/$40.0_{\pm6.7}$ | |
| | EBM | $29.8_{\pm7.7}$/$29.3_{\pm7.9}$ | | $52.6_{\pm3.8}$/$52.7_{\pm4.1}$ | | $69.9_{\pm4.7}$/$86.7_{\pm9.4}$ | |
| | Safe | $29.8_{\pm7.7}$/$38.9_{\pm7.2}$ | $49.0_{\pm3.0}$ | $52.6_{\pm3.8}$/$51.6_{\pm4.1}$ | $41.8_{\pm2.2}$ | $69.9_{\pm4.7}$/$62.4_{\pm6.8}$ | $40.5_{\pm1.9}$ |
| | GEBM | $29.8_{\pm7.7}$/$61.5_{\pm8.0}$ | | $52.6_{\pm3.8}$/$28.6_{\pm5.8}$ | | $69.9_{\pm4.7}$/$96.6_{\pm4.5}$ | |
| | JLDE | $29.8_{\pm7.7}$/$54.7_{\pm4.4}$ | | $52.6_{\pm3.8}$/$43.1_{\pm4.7}$ | | $69.9_{\pm4.7}$/$100.0_{\pm0.0}$ | |
| CoraML | GPN | $82.8_{\pm2.5}$/$81.9_{\pm8.2}$ | $88.9_{\pm3.9}$ | $55.4_{\pm2.8}$/$53.8_{\pm2.4}$ | $80.9_{\pm2.0}$ | $49.1_{\pm2.0}$/$61.4_{\pm5.0}$ | $81.1_{\pm1.9}$ |
| | SGCN | $86.3_{\pm1.4}$/$87.2_{\pm1.8}$ | $89.4_{\pm1.5}$ | $63.7_{\pm2.2}$/$66.9_{\pm1.9}$ | $82.0_{\pm1.5}$ | $16.8_{\pm1.4}$/$12.2_{\pm2.7}$ | $34.4_{\pm2.7}$ |
| | BGCN | $85.3_{\pm2.9}$/$75.1_{\pm7.7}$ | $88.3_{\pm2.0}$ | $59.8_{\pm2.3}$/$57.7_{\pm2.3}$ | $81.3_{\pm1.6}$ | $28.5_{\pm6.1}$/$34.3_{\pm6.4}$ | $34.1_{\pm2.9}$ |
| | GPRGNN | $85.1_{\pm2.9}$/$87.4_{\pm2.7}$ | $90.2_{\pm1.8}$ | $61.0_{\pm1.9}$/$65.9_{\pm1.9}$ | $84.1_{\pm1.2}$ | $17.5_{\pm2.9}$/$4.8_{\pm0.8}$ | $37.3_{\pm4.2}$ |
| | FSGNN | $82.7_{\pm1.5}$/$81.8_{\pm1.8}$ | $87.5_{\pm1.5}$ | $63.5_{\pm2.4}$/$69.8_{\pm3.2}$ | $71.7_{\pm1.3}$ | $6.8_{\pm0.9}$/$1.6_{\pm0.4}$ | $40.2_{\pm3.0}$ |
| | FAGCN | $85.0_{\pm1.7}$/$86.4_{\pm1.9}$ | $88.9_{\pm1.6}$ | $68.7_{\pm3.0}$/$76.0_{\pm3.4}$ | $81.4_{\pm2.3}$ | $11.2_{\pm1.3}$/$1.2_{\pm0.2}$ | $40.4_{\pm3.4}$ |
| | Ens. | $81.3_{\pm1.3}$/$81.4_{\pm1.9}$ | | $52.7_{\pm2.3}$/$52.8_{\pm2.4}$ | | $84.7_{\pm2.1}$/$78.4_{\pm1.8}$ | |
| | MCD | $79.6_{\pm1.9}$/$78.1_{\pm2.9}$ | | $54.6_{\pm2.5}$/$55.1_{\pm2.4}$ | | $88.0_{\pm1.9}$/$64.0_{\pm9.3}$ | |
| | EBM | $76.9_{\pm2.7}$/$76.0_{\pm4.4}$ | | $52.1_{\pm2.5}$/$53.3_{\pm2.7}$ | | $85.0_{\pm2.2}$/$93.5_{\pm1.5}$ | |
| | Safe | $76.9_{\pm2.7}$/$80.7_{\pm5.3}$ | $88.3_{\pm1.4}$ | $52.1_{\pm2.5}$/$51.0_{\pm1.9}$ | $79.6_{\pm1.5}$ | $85.0_{\pm2.2}$/$72.0_{\pm2.4}$ | $77.1_{\pm1.8}$ |
| | GEBM | $76.9_{\pm2.7}$/$81.4_{\pm3.7}$ | | $52.1_{\pm2.5}$/$53.3_{\pm10.0}$ | | $85.0_{\pm2.2}$/$80.5_{\pm3.2}$ | |
| | JLDE | $76.9_{\pm2.7}$/$80.8_{\pm1.5}$ | | $52.1_{\pm2.5}$/$54.9_{\pm2.1}$ | | $85.0_{\pm2.2}$/$95.7_{\pm0.8}$ | |
| PubMed | GPN | $69.1_{\pm4.4}$/$68.2_{\pm9.3}$ | $91.2_{\pm8.1}$ | $61.4_{\pm2.3}$/$60.6_{\pm2.3}$ | $84.5_{\pm0.5}$ | $54.1_{\pm0.9}$/$66.6_{\pm2.6}$ | $84.5_{\pm0.5}$ |
| | SGCN | $68.7_{\pm2.1}$/$68.7_{\pm2.1}$ | $94.6_{\pm0.2}$ | $64.8_{\pm0.4}$/$67.3_{\pm1.3}$ | $85.7_{\pm0.3}$ | $21.3_{\pm3.4}$/$20.6_{\pm3.8}$ | $50.2_{\pm1.6}$ |
| | BGCN | $68.5_{\pm4.0}$/$68.1_{\pm4.3}$ | $94.3_{\pm0.5}$ | $65.6_{\pm0.6}$/$62.8_{\pm1.3}$ | $85.7_{\pm0.3}$ | $22.7_{\pm1.3}$/$30.0_{\pm1.6}$ | $53.4_{\pm1.4}$ |
| | GPRGNN | $69.9_{\pm2.7}$/$70.0_{\pm2.6}$ | $95.2_{\pm0.3}$ | $66.5_{\pm1.2}$/$72.9_{\pm1.5}$ | $86.0_{\pm0.3}$ | $18.6_{\pm7.5}$/$8.5_{\pm1.3}$ | $51.5_{\pm4.5}$ |
| | FSGNN | $63.9_{\pm2.2}$/$63.9_{\pm2.1}$ | $95.4_{\pm0.2}$ | $87.6_{\pm2.9}$/$88.3_{\pm2.3}$ | $82.8_{\pm0.5}$ | $3.4_{\pm0.5}$/$0.6_{\pm0.2}$ | $69.9_{\pm1.7}$ |
| | FAGCN | $68.5_{\pm2.2}$/$68.4_{\pm2.6}$ | $94.7_{\pm0.2}$ | $69.5_{\pm1.0}$/$73.6_{\pm1.8}$ | $85.8_{\pm0.4}$ | $12.8_{\pm1.1}$/$4.3_{\pm0.6}$ | $54.2_{\pm1.8}$ |
| | Ens. | $59.5_{\pm1.9}$/$59.6_{\pm2.5}$ | | $64.0_{\pm2.1}$/$64.7_{\pm2.6}$ | | $82.0_{\pm0.5}$/$81.4_{\pm0.6}$ | |
| | MCD | $56.3_{\pm3.7}$/$56.9_{\pm3.5}$ | | $63.3_{\pm2.4}$/$63.4_{\pm2.3}$ | | $85.3_{\pm1.0}$/$74.9_{\pm1.1}$ | |
| | EBM | $57.5_{\pm3.8}$/$58.0_{\pm5.4}$ | | $61.9_{\pm2.9}$/$61.0_{\pm5.4}$ | | $81.0_{\pm1.3}$/$88.5_{\pm2.3}$ | |
| | Safe | $57.5_{\pm3.8}$/$60.8_{\pm4.0}$ | $94.9_{\pm0.3}$ | $61.9_{\pm2.9}$/$54.0_{\pm1.8}$ | $83.9_{\pm0.5}$ | $81.0_{\pm1.3}$/$66.3_{\pm3.3}$ | $82.9_{\pm0.4}$ |
| | GEBM | $57.5_{\pm3.8}$/$65.0_{\pm4.2}$ | | $61.9_{\pm2.9}$/$62.1_{\pm5.0}$ | | $81.0_{\pm1.3}$/$74.2_{\pm3.9}$ | |
| | JLDE | $57.5_{\pm3.8}$/$59.4_{\pm2.1}$ | | $61.9_{\pm2.9}$/$64.3_{\pm3.0}$ | | $81.0_{\pm1.3}$/$99.4_{\pm0.1}$ | |

*Table 8.* O.o.d. detection AUC-PR using different uncertainty estimators and our method using a Res-GCN backbone.

| | Model | LoC AUC-PR (Alea. / Epi.) | LoC Acc.↑ | Near-Features AUC-PR (Alea. / Epi.) | Near-Features Acc.↑ | Far-Features AUC-PR (Alea. / Epi.) | Far-Features Acc.↑ |
|---|---|---|---|---|---|---|---|
| **Roman Empire** | GPN | $26.4_{\pm2.1}/30.5_{\pm2.2}$ | $48.6_{\pm1.9}$ | $13.1_{\pm0.8}/17.6_{\pm1.7}$ | $38.1_{\pm1.5}$ | $11.4_{\pm0.5}/30.0_{\pm3.1}$ | $38.0_{\pm1.3}$ |
| | SGCN | $26.1_{\pm0.6}/22.4_{\pm0.7}$ | $58.3_{\pm1.6}$ | $12.1_{\pm0.7}/11.9_{\pm0.9}$ | $45.5_{\pm1.6}$ | $5.6_{\pm0.0}/5.4_{\pm0.0}$ | $30.6_{\pm1.0}$ |
| | BGCN | $23.3_{\pm1.1}/27.3_{\pm2.3}$ | $53.0_{\pm0.7}$ | $14.3_{\pm0.9}/8.8_{\pm0.4}$ | $42.4_{\pm0.5}$ | $7.6_{\pm0.7}/29.4_{\pm1.7}$ | $28.8_{\pm0.6}$ |
| | GPRGNN | $30.6_{\pm0.8}/24.9_{\pm0.9}$ | $78.1_{\pm0.4}$ | $41.9_{\pm2.7}/43.9_{\pm6.6}$ | $64.8_{\pm0.6}$ | $5.8_{\pm0.1}/72.9_{\pm6.0}$ | $61.0_{\pm0.6}$ |
| | FSGNN | $35.2_{\pm1.1}/37.5_{\pm1.5}$ | $86.6_{\pm0.4}$ | $17.9_{\pm1.0}/25.7_{\pm2.3}$ | $71.6_{\pm0.5}$ | $6.8_{\pm0.3}/5.2_{\pm0.1}$ | $56.9_{\pm0.9}$ |
| | FAGCN | $35.6_{\pm1.6}/34.7_{\pm1.8}$ | $83.6_{\pm1.1}$ | $26.6_{\pm2.7}/46.8_{\pm5.1}$ | $68.0_{\pm0.9}$ | $6.8_{\pm0.3}/5.5_{\pm0.3}$ | $44.3_{\pm2.8}$ |
| | Ens. | $41.4_{\pm1.4}/40.1_{\pm1.1}$ | | $27.6_{\pm0.8}/24.7_{\pm1.3}$ | | $52.6_{\pm1.9}/35.7_{\pm2.0}$ | |
| | MCD | $40.4_{\pm1.4}/39.8_{\pm1.5}$ | | $23.2_{\pm1.4}/16.9_{\pm0.7}$ | | $33.4_{\pm2.0}/15.6_{\pm1.0}$ | |
| | EBM | $36.9_{\pm1.5}/40.8_{\pm2.3}$ | $87.6_{\pm0.7}$ | $21.8_{\pm1.1}/35.6_{\pm2.3}$ | $72.3_{\pm0.6}$ | $34.6_{\pm2.1}/66.3_{\pm3.2}$ | $70.6_{\pm0.6}$ |
| | Safe | $36.9_{\pm1.5}/30.2_{\pm1.6}$ | | $21.8_{\pm1.1}/22.6_{\pm1.6}$ | | $34.6_{\pm2.1}/31.5_{\pm2.2}$ | |
| | GEBM | $36.9_{\pm1.5}/21.9_{\pm2.8}$ | | $21.8_{\pm1.1}/17.1_{\pm2.6}$ | | $34.6_{\pm2.1}/37.1_{\pm7.2}$ | |
| | JLDE | $36.9_{\pm1.5}/42.6_{\pm2.0}$ | | $21.8_{\pm1.1}/63.9_{\pm4.0}$ | | $34.6_{\pm2.1}/100.0_{\pm0.0}$ | |
| **Amazon Ratings** | GPN | $12.9_{\pm0.0}/14.2_{\pm1.6}$ | $54.2_{\pm1.2}$ | $11.2_{\pm0.5}/10.3_{\pm0.7}$ | $46.8_{\pm0.7}$ | $10.7_{\pm0.4}/12.7_{\pm0.8}$ | $47.0_{\pm0.7}$ |
| | SGCN | $12.9_{\pm0.7}/12.5_{\pm0.6}$ | $55.5_{\pm0.7}$ | $11.3_{\pm0.4}/11.2_{\pm0.6}$ | $47.2_{\pm0.5}$ | $7.3_{\pm0.6}/6.8_{\pm0.6}$ | $33.5_{\pm1.3}$ |
| | BGCN | $14.4_{\pm0.9}/12.6_{\pm1.3}$ | $51.4_{\pm0.8}$ | $11.2_{\pm0.6}/9.0_{\pm0.4}$ | $44.2_{\pm1.3}$ | $8.8_{\pm0.9}/22.4_{\pm1.3}$ | $33.5_{\pm1.3}$ |
| | GPRGNN | $14.2_{\pm0.9}/14.0_{\pm0.6}$ | $57.9_{\pm0.8}$ | $12.7_{\pm0.5}/16.8_{\pm1.0}$ | $50.6_{\pm0.7}$ | $9.3_{\pm0.2}/48.1_{\pm10.4}$ | $34.1_{\pm2.7}$ |
| | FSGNN | $14.9_{\pm0.5}/14.8_{\pm0.6}$ | $56.8_{\pm0.6}$ | $8.7_{\pm0.2}/7.8_{\pm0.3}$ | $49.1_{\pm0.6}$ | $9.6_{\pm0.1}/6.5_{\pm0.8}$ | $38.6_{\pm0.8}$ |
| | FAGCN | $14.5_{\pm0.7}/14.7_{\pm0.7}$ | $58.1_{\pm0.8}$ | $12.5_{\pm0.6}/13.7_{\pm1.0}$ | $51.2_{\pm0.7}$ | $7.9_{\pm0.4}/7.4_{\pm1.6}$ | $31.7_{\pm2.4}$ |
| | Ens. | $13.7_{\pm0.4}/18.9_{\pm0.9}$ | | $13.3_{\pm0.8}/15.7_{\pm0.9}$ | | $26.3_{\pm1.8}/9.1_{\pm0.3}$ | |
| | MCD | $13.0_{\pm0.5}/15.1_{\pm0.7}$ | | $12.0_{\pm0.8}/13.1_{\pm0.8}$ | | $25.4_{\pm2.4}/7.1_{\pm0.4}$ | |
| | EBM | $13.0_{\pm0.5}/13.2_{\pm1.1}$ | $56.0_{\pm0.6}$ | $11.3_{\pm0.7}/12.0_{\pm1.0}$ | $48.2_{\pm0.5}$ | $30.8_{\pm3.3}/71.1_{\pm5.5}$ | $47.3_{\pm0.6}$ |
| | Safe | $13.0_{\pm0.5}/13.1_{\pm1.3}$ | | $11.3_{\pm0.7}/11.8_{\pm1.0}$ | | $30.8_{\pm3.3}/16.9_{\pm1.9}$ | |
| | GEBM | $13.0_{\pm0.5}/13.4_{\pm1.1}$ | | $11.3_{\pm0.7}/9.1_{\pm0.4}$ | | $30.8_{\pm3.3}/14.3_{\pm3.6}$ | |
| | JLDE | $13.0_{\pm0.5}/17.2_{\pm0.5}$ | | $11.3_{\pm0.7}/15.3_{\pm0.7}$ | | $30.8_{\pm3.3}/89.0_{\pm1.8}$ | |
| **Chameleon** | GPN | $41.5_{\pm7.9}/64.4_{\pm12.0}$ | $39.6_{\pm3.8}$ | $14.8_{\pm4.4}/12.1_{\pm2.8}$ | $37.0_{\pm3.8}$ | $13.0_{\pm3.8}/18.5_{\pm6.8}$ | $37.0_{\pm3.8}$ |
| | SGCN | $42.1_{\pm8.6}/53.4_{\pm6.9}$ | $43.3_{\pm4.2}$ | $14.2_{\pm3.1}/12.1_{\pm3.1}$ | $38.2_{\pm3.1}$ | $6.8_{\pm1.0}/6.2_{\pm1.0}$ | $25.2_{\pm4.8}$ |
| | BGCN | $39.9_{\pm6.7}/34.3_{\pm3.7}$ | $40.5_{\pm4.0}$ | $15.4_{\pm5.3}/11.4_{\pm1.9}$ | $38.3_{\pm3.3}$ | $7.7_{\pm1.3}/14.0_{\pm6.0}$ | $26.9_{\pm4.8}$ |
| | GPRGNN | $45.3_{\pm7.1}/41.8_{\pm6.8}$ | $36.8_{\pm4.6}$ | $16.6_{\pm4.2}/16.5_{\pm5.1}$ | $37.1_{\pm3.6}$ | $6.3_{\pm0.5}/6.0_{\pm2.6}$ | $30.5_{\pm4.1}$ |
| | FSGNN | $34.2_{\pm5.8}/44.9_{\pm8.3}$ | $43.9_{\pm4.2}$ | $16.6_{\pm7.3}/16.2_{\pm3.8}$ | $36.9_{\pm3.6}$ | $5.9_{\pm0.4}/6.1_{\pm2.2}$ | $30.5_{\pm3.7}$ |
| | FAGCN | $38.5_{\pm5.0}/39.7_{\pm4.9}$ | $43.2_{\pm6.4}$ | $23.1_{\pm7.4}/25.0_{\pm8.6}$ | $39.7_{\pm3.6}$ | $7.5_{\pm0.7}/5.4_{\pm0.1}$ | $30.2_{\pm4.1}$ |
| | Ens. | $35.6_{\pm4.5}/43.8_{\pm6.4}$ | | $13.8_{\pm4.9}/12.1_{\pm2.9}$ | | $46.2_{\pm10.6}/10.0_{\pm2.4}$ | |
| | MCD | $35.8_{\pm5.2}/37.8_{\pm4.3}$ | | $13.7_{\pm3.5}/13.1_{\pm3.8}$ | | $47.6_{\pm15.0}/8.4_{\pm1.6}$ | |
| | EBM | $35.8_{\pm5.0}/36.1_{\pm6.4}$ | $47.8_{\pm4.9}$ | $13.6_{\pm4.5}/13.9_{\pm4.5}$ | $40.7_{\pm3.9}$ | $39.8_{\pm12.4}/74.3_{\pm18.8}$ | $40.1_{\pm3.3}$ |
| | Safe | $35.8_{\pm5.0}/36.1_{\pm5.3}$ | | $13.6_{\pm4.5}/14.1_{\pm5.0}$ | | $39.8_{\pm12.4}/23.1_{\pm8.1}$ | |
| | GEBM | $35.8_{\pm5.0}/59.3_{\pm10.1}$ | | $13.6_{\pm4.5}/8.7_{\pm4.5}$ | | $39.8_{\pm12.4}/56.0_{\pm16.9}$ | |
| | JLDE | $35.8_{\pm5.0}/50.6_{\pm5.1}$ | | $13.6_{\pm4.5}/11.4_{\pm2.7}$ | | $39.8_{\pm12.4}/98.9_{\pm2.9}$ | |
| **Squirrel** | GPN | $59.8_{\pm8.5}/79.4_{\pm2.9}$ | $41.1_{\pm2.9}$ | $12.3_{\pm3.2}/12.0_{\pm2.7}$ | $34.5_{\pm1.6}$ | $20.2_{\pm14.0}/30.4_{\pm10.2}$ | $34.6_{\pm1.6}$ |
| | SGCN | $51.3_{\pm3.1}/72.7_{\pm5.3}$ | $47.1_{\pm2.9}$ | $12.9_{\pm1.9}/16.7_{\pm5.2}$ | $37.7_{\pm1.7}$ | $7.1_{\pm1.4}/6.9_{\pm2.1}$ | $28.2_{\pm3.8}$ |
| | BGCN | $58.3_{\pm6.6}/42.6_{\pm1.1}$ | $43.2_{\pm4.4}$ | $12.6_{\pm2.3}/10.6_{\pm1.3}$ | $33.9_{\pm3.1}$ | $7.3_{\pm1.2}/25.7_{\pm6.3}$ | $30.2_{\pm3.0}$ |
| | GPRGNN | $59.1_{\pm13.1}/59.9_{\pm11.3}$ | $40.9_{\pm3.6}$ | $12.7_{\pm2.9}/13.0_{\pm3.0}$ | $35.4_{\pm1.4}$ | $6.3_{\pm2.8}/11.8_{\pm3.6}$ | $34.4_{\pm1.9}$ |
| | FSGNN | $46.5_{\pm3.4}/48.6_{\pm6.2}$ | $45.1_{\pm3.2}$ | $15.1_{\pm3.5}/11.2_{\pm1.7}$ | $36.1_{\pm1.7}$ | $5.8_{\pm0.4}/5.9_{\pm1.0}$ | $33.9_{\pm2.4}$ |
| | FAGCN | $52.6_{\pm7.0}/56.0_{\pm6.2}$ | $47.0_{\pm6.5}$ | $12.9_{\pm2.7}/11.6_{\pm1.8}$ | $40.5_{\pm2.8}$ | $5.8_{\pm0.4}/7.4_{\pm7.4}$ | $35.3_{\pm2.7}$ |
| | Ens. | $46.4_{\pm3.6}/56.3_{\pm1.8}$ | | $11.7_{\pm1.9}/10.6_{\pm1.4}$ | | $27.9_{\pm4.6}/18.2_{\pm7.5}$ | |
| | MCD | $48.2_{\pm4.7}/51.4_{\pm3.5}$ | | $11.5_{\pm1.8}/10.7_{\pm1.9}$ | | $23.9_{\pm8.8}/8.0_{\pm1.2}$ | |
| | EBM | $46.4_{\pm4.4}/46.0_{\pm4.8}$ | $49.0_{\pm3.0}$ | $11.5_{\pm1.9}/11.4_{\pm1.7}$ | $41.8_{\pm2.2}$ | $26.5_{\pm8.2}/64.0_{\pm20.0}$ | $40.5_{\pm1.9}$ |
| | Safe | $46.4_{\pm4.4}/53.7_{\pm4.3}$ | | $11.5_{\pm1.9}/11.4_{\pm1.7}$ | | $26.5_{\pm8.2}/18.3_{\pm5.7}$ | |
| | GEBM | $46.4_{\pm4.4}/69.2_{\pm7.3}$ | | $11.5_{\pm1.9}/6.7_{\pm0.5}$ | | $26.5_{\pm8.2}/79.0_{\pm17.6}$ | |
| | JLDE | $46.4_{\pm4.4}/61.5_{\pm3.0}$ | | $11.5_{\pm1.9}/8.9_{\pm0.9}$ | | $26.5_{\pm8.2}/99.8_{\pm0.7}$ | |
| **CoraML** | GPN | $75.5_{\pm4.4}/75.6_{\pm9.2}$ | $88.9_{\pm3.9}$ | $11.7_{\pm1.0}/11.3_{\pm0.9}$ | $80.9_{\pm2.0}$ | $10.3_{\pm0.9}/23.5_{\pm7.0}$ | $81.1_{\pm1.9}$ |
| | SGCN | $82.1_{\pm1.7}/84.5_{\pm2.2}$ | $89.4_{\pm1.5}$ | $16.3_{\pm0.9}/17.3_{\pm1.3}$ | $82.0_{\pm1.5}$ | $7.0_{\pm0.6}/6.0_{\pm0.5}$ | $34.4_{\pm2.7}$ |
| | BGCN | $80.4_{\pm4.6}/65.7_{\pm7.3}$ | $88.3_{\pm2.0}$ | $14.2_{\pm1.3}/11.9_{\pm0.8}$ | $81.3_{\pm1.6}$ | $7.7_{\pm0.7}/14.2_{\pm1.4}$ | $34.1_{\pm2.9}$ |
| | GPRGNN | $79.4_{\pm4.0}/82.9_{\pm4.6}$ | $90.2_{\pm1.8}$ | $15.1_{\pm1.5}/15.4_{\pm1.0}$ | $84.1_{\pm1.2}$ | $6.9_{\pm0.4}/5.3_{\pm0.1}$ | $37.3_{\pm4.2}$ |
| | FSGNN | $75.2_{\pm2.3}/74.9_{\pm4.3}$ | $87.5_{\pm1.5}$ | $13.0_{\pm1.1}/15.8_{\pm1.6}$ | $71.7_{\pm1.3}$ | $8.1_{\pm0.2}/5.3_{\pm0.1}$ | $40.2_{\pm3.0}$ |
| | FAGCN | $79.9_{\pm2.9}/82.1_{\pm3.3}$ | $88.9_{\pm1.6}$ | $19.2_{\pm2.6}/24.2_{\pm3.0}$ | $81.4_{\pm2.3}$ | $7.4_{\pm0.2}/5.2_{\pm0.0}$ | $40.4_{\pm3.4}$ |
| | Ens. | $74.8_{\pm1.9}/76.1_{\pm2.6}$ | | $11.3_{\pm0.8}/11.7_{\pm1.0}$ | | $42.3_{\pm6.0}/19.8_{\pm1.6}$ | |
| | MCD | $72.3_{\pm2.7}/70.1_{\pm3.5}$ | | $12.0_{\pm1.2}/12.6_{\pm1.3}$ | | $53.1_{\pm5.9}/12.7_{\pm2.1}$ | |
| | EBM | $69.4_{\pm2.8}/70.5_{\pm5.3}$ | $88.3_{\pm1.4}$ | $10.9_{\pm0.9}/11.7_{\pm1.2}$ | $79.6_{\pm1.5}$ | $42.4_{\pm7.6}/76.2_{\pm5.5}$ | $77.1_{\pm1.8}$ |
| | Safe | $69.4_{\pm2.8}/75.9_{\pm7.2}$ | | $10.9_{\pm0.9}/10.6_{\pm0.6}$ | | $42.4_{\pm7.6}/22.0_{\pm2.2}$ | |
| | GEBM | $69.4_{\pm2.8}/75.7_{\pm5.0}$ | | $10.9_{\pm0.9}/11.1_{\pm4.2}$ | | $42.4_{\pm7.6}/37.3_{\pm7.1}$ | |
| | JLDE | $69.4_{\pm2.8}/72.6_{\pm2.4}$ | | $10.9_{\pm0.9}/11.1_{\pm0.7}$ | | $42.4_{\pm7.6}/70.1_{\pm3.5}$ | |
| **PubMed** | GPN | $55.7_{\pm4.4}/61.3_{\pm12.0}$ | $91.2_{\pm8.1}$ | $13.3_{\pm0.9}/13.7_{\pm1.5}$ | $84.5_{\pm0.5}$ | $11.0_{\pm0.2}/22.2_{\pm4.8}$ | $84.5_{\pm0.5}$ |
| | SGCN | $55.1_{\pm2.3}/54.4_{\pm2.1}$ | $94.6_{\pm0.2}$ | $16.1_{\pm0.4}/16.9_{\pm0.8}$ | $85.7_{\pm0.3}$ | $8.0_{\pm0.7}/7.2_{\pm0.8}$ | $50.2_{\pm1.6}$ |
| | BGCN | $54.3_{\pm3.0}/53.1_{\pm4.3}$ | $94.3_{\pm0.5}$ | $16.6_{\pm0.3}/13.3_{\pm0.5}$ | $85.7_{\pm0.3}$ | $6.8_{\pm0.1}/17.2_{\pm1.1}$ | $53.4_{\pm1.4}$ |
| | GPRGNN | $57.2_{\pm2.9}/57.3_{\pm2.9}$ | $95.2_{\pm0.3}$ | $16.5_{\pm0.8}/20.0_{\pm1.9}$ | $86.0_{\pm0.3}$ | $8.4_{\pm0.6}/6.1_{\pm0.8}$ | $51.5_{\pm4.5}$ |
| | FSGNN | $50.2_{\pm2.1}/50.1_{\pm2.1}$ | $95.4_{\pm0.2}$ | $37.2_{\pm6.2}/32.1_{\pm4.8}$ | $82.8_{\pm0.5}$ | $7.6_{\pm0.4}/5.2_{\pm0.1}$ | $69.9_{\pm1.7}$ |
| | FAGCN | $55.5_{\pm2.7}/55.6_{\pm2.7}$ | $94.7_{\pm0.2}$ | $18.5_{\pm1.6}/22.1_{\pm1.8}$ | $85.8_{\pm0.4}$ | $8.0_{\pm0.2}/5.3_{\pm0.1}$ | $54.2_{\pm1.8}$ |
| | Ens. | $48.2_{\pm1.8}/48.9_{\pm2.0}$ | | $15.5_{\pm1.1}/17.5_{\pm2.5}$ | | $29.6_{\pm0.8}/26.4_{\pm1.6}$ | |
| | MCD | $46.4_{\pm2.7}/47.1_{\pm2.6}$ | | $16.1_{\pm1.6}/17.6_{\pm2.5}$ | | $41.2_{\pm2.9}/17.2_{\pm0.6}$ | |
| | EBM | $46.2_{\pm2.3}/47.0_{\pm3.8}$ | $94.9_{\pm0.3}$ | $14.1_{\pm1.1}/14.5_{\pm2.3}$ | $83.9_{\pm0.5}$ | $28.2_{\pm1.8}/61.1_{\pm5.7}$ | $82.9_{\pm0.4}$ |
| | Safe | $46.2_{\pm2.3}/47.7_{\pm3.6}$ | | $14.1_{\pm1.1}/11.0_{\pm0.6}$ | | $28.2_{\pm1.8}/17.0_{\pm1.7}$ | |
| | GEBM | $46.2_{\pm2.3}/51.9_{\pm4.0}$ | | $14.1_{\pm1.1}/13.8_{\pm2.3}$ | | $28.2_{\pm1.8}/30.4_{\pm4.0}$ | |
| | JLDE | $46.2_{\pm2.3}/47.4_{\pm2.0}$ | | $14.1_{\pm1.1}/13.5_{\pm1.0}$ | | $28.2_{\pm1.8}/90.0_{\pm1.6}$ | |

*Table 9.* O.o.d. detection AUC-ROC using different uncertainty estimators for different GNN backbones for a leave-out-classes distribution shift.

| Backbone | Model | Roman Empire AUC-ROC (Alea./Epi.) | Roman Empire Acc.↑ | Amazon Ratings AUC-ROC (Alea./Epi.) | Amazon Ratings Acc.↑ | Chameleon AUC-ROC (Alea./Epi.) | Chameleon Acc.↑ | Squirrel AUC-ROC (Alea./Epi.) | Squirrel Acc.↑ | CoraML AUC-ROC (Alea./Epi.) | CoraML Acc.↑ | PubMed AUC-ROC (Alea./Epi.) | PubMed Acc.↑ |
|---|---|---|---|---|---|---|---|---|---|---|---|---|---|
| Res-GCN | Ens. | **76.4**±0.8/**76.2**±0.7 | 87.6±0.7 | 51.4±0.8/**57.9**±0.8 | 56.0±0.6 | 34.4±6.3/49.9±11.4 | 47.8±4.9 | 29.5±6.5/44.9±4.8 | 49.0±3.0 | **81.3**±1.3/**81.4**±1.9 | 88.3±1.4 | 59.5±1.9/59.6±2.5 | 94.9±0.3 |
| | MCD | 76.3±1.0/75.9±0.9 | | 49.2±1.1/53.5±1.3 | | 35.6±7.3/40.1±5.4 | | 32.0±7.3/37.2±5.3 | | 79.6±1.1/78.1±2.9 | | 56.3±3.7/56.9±3.5 | |
| | EBM | 73.3±1.3/72.5±1.6 | | 48.8±1.2/48.9±3.5 | | 35.3±8.2/35.7±8.8 | | 29.8±7.7/29.3±7.9 | | 76.9±2.7/76.0±4.4 | | 57.5±3.8/58.0±5.4 | |
| | Safe | 73.3±1.3/64.8±2.1 | | 48.8±1.2/48.0±4.2 | | 35.3±8.2/36.6±8.5 | | 29.8±7.7/38.9±7.2 | | 76.9±2.7/80.7±5.3 | | 57.5±3.8/**60.8**±4.0 | |
| | GEBM | 73.3±1.3/53.6±3.8 | | 48.8±1.2/49.9±3.5 | | 35.3±8.2/**66.4**±9.1 | | 29.8±7.7/**61.5**±8.0 | | 76.9±2.7/**81.4**±3.7 | | 57.5±3.8/**65.0**±4.2 | |
| | JLDE | 73.3±1.3/**76.9**±1.4 | | 48.8±1.2/**61.2**±0.8 | | 35.3±8.2/**61.5**±5.8 | | 29.8±7.7/**54.7**±4.4 | | 76.9±2.7/80.8±1.5 | | 57.5±3.8/59.4±2.1 | |
| Res-GAT-Sep | Ens. | **79.2**±0.7/**79.1**±0.8 | 91.8±0.4 | **60.4**±0.6/**60.4**±0.7 | 58.6±0.8 | 37.4±6.0/42.4±5.0 | 37.5±5.2 | 41.4±4.3/47.5±6.1 | 40.3±2.8 | **81.4**±1.3/**81.4**±1.6 | 87.6±1.5 | **64.4**±3.5/63.9±4.1 | 95.1±0.3 |
| | MCD | 77.4±1.1/77.0±1.0 | | 57.2±1.2/57.7±1.1 | | 42.5±8.0/46.9±6.2 | | 45.0±5.5/46.9±4.6 | | 79.7±1.8/79.9±1.8 | | 62.3±5.3/62.2±5.2 | |
| | EBM | 75.5±1.6/72.0±3.5 | | 56.1±1.4/54.3±2.0 | | 41.7±8.7/45.1±9.9 | | 44.4±4.6/45.7±8.0 | | 75.9±2.8/74.3±4.4 | | 61.7±5.7/61.7±5.7 | |
| | Safe | 75.5±1.6/68.4±4.6 | | 56.1±1.4/53.4±3.6 | | 41.7±8.7/45.2±12.1 | | 44.4±4.6/45.4±8.0 | | 75.9±2.8/**80.5**±5.3 | | 61.7±5.7/63.3±5.6 | |
| | GEBM | 75.5±1.6/65.9±2.6 | | 56.1±1.4/50.1±1.9 | | 41.7±8.7/**58.1**±13.7 | | 44.4±4.6/**57.4**±7.1 | | 75.9±2.8/79.0±5.1 | | 61.7±5.7/**64.5**±5.0 | |
| | JLDE | 75.5±1.6/76.9±1.0 | | 56.1±1.4/**61.3**±1.1 | | 41.7±8.7/**55.0**±9.7 | | 44.4±4.6/**60.6**±7.3 | | 75.9±2.8/77.0±2.7 | | 61.7±5.7/56.5±4.5 | |
| Res-GATE | Ens. | **78.6**±1.0/**78.7**±1.0 | 92.3±0.4 | **61.8**±1.0/61.6±1.0 | 57.4±0.6 | 40.8±6.3/48.5±5.7 | 39.9±4.0 | 40.5±3.1/48.3±6.4 | 40.9±2.6 | 82.3±1.9/**82.7**±2.2 | 88.9±1.3 | 59.6±2.2/59.1±4.1 | 95.2±0.2 |
| | MCD | 77.8±1.1/77.1±1.1 | | 57.5±1.2/57.5±1.1 | | 41.6±7.0/49.1±5.9 | | 43.1±5.4/49.7±4.3 | | 80.7±2.7/80.3±3.7 | | 56.1±3.9/56.9±3.6 | |
| | EBM | 75.8±1.5/73.7±2.0 | | 57.0±1.4/56.3±1.8 | | 40.3±7.1/42.7±8.2 | | 41.5±5.0/43.2±5.0 | | 77.9±3.1/76.3±5.1 | | 57.8±3.9/58.5±5.7 | |
| | Safe | 75.8±1.5/71.4±2.4 | | 57.0±1.4/57.8±3.3 | | 40.3±7.1/47.4±9.9 | | 41.5±5.0/47.2±6.2 | | 77.9±3.1/80.6±6.0 | | 57.8±3.9/61.1±4.8 | |
| | GEBM | 75.8±1.5/58.4±4.7 | | 57.0±1.4/50.2±2.8 | | 40.3±7.1/**59.9**±10.8 | | 41.5±5.0/**59.8**±8.8 | | 77.9±3.1/81.1±5.3 | | 57.8±3.9/**64.7**±4.5 | |
| | JLDE | 75.8±1.5/**81.5**±0.9 | | 57.0±1.4/**62.1**±1.2 | | 40.3±7.1/**62.8**±6.6 | | 41.5±5.0/**62.5**±7.8 | | 77.9±3.1/**83.2**±1.8 | | 57.8±3.9/**63.3**±2.2 | |
| GPRGNN | Ens. | 66.8±0.8/69.2±1.2 | 78.1±0.4 | 52.9±0.9/53.9±1.3 | 57.9±0.8 | 51.2±5.4/46.7±6.7 | 36.8±4.6 | 44.1±14.6/51.7±12.9 | 40.9±3.6 | 86.5±1.5/74.9±19.9 | 90.2±1.8 | 71.5±3.3/72.0±2.3 | 95.2±0.3 |
| | MCD | 68.2±0.9/57.2±1.9 | | 52.1±1.3/52.3±1.1 | | 53.3±5.7/53.9±6.4 | | 49.0±15.5/**56.8**±6.8 | | 85.2±3.1/77.2±9.0 | | 68.5±3.1/67.3±2.5 | |
| | EBM | 66.5±0.9/63.4±1.0 | | 52.5±1.2/52.1±1.4 | | 51.6±7.1/48.8±7.7 | | 48.3±16.0/48.1±14.6 | | 85.1±2.9/87.4±2.7 | | 69.9±2.9/70.0±2.6 | |
| | Safe | 66.5±0.9/**70.3**±1.7 | | 52.5±1.2/51.1±1.7 | | 51.6±7.1/53.2±14.5 | | 48.3±16.0/41.2±3.9 | | 85.1±2.9/**89.7**±2.0 | | 69.9±2.7/**72.1**±2.9 | |
| | GEBM | 66.5±0.9/38.4±3.9 | | 52.5±1.2/**54.3**±1.1 | | 51.6±7.1/**64.0**±6.5 | | 48.3±16.0/**62.0**±5.6 | | 85.1±2.9/**88.6**±2.5 | | 69.9±2.7/**72.5**±2.6 | |
| | JLDE | 66.5±0.9/**71.2**±1.0 | | 52.5±1.2/**54.4**±0.8 | | 51.6±7.1/**55.9**±5.3 | | 48.3±16.0/56.1±6.4 | | 85.1±2.9/81.9±2.4 | | 69.9±2.7/56.8±2.2 | |

Table 10. O.o.d. detection AUC-ROC using different uncertainty estimators for different GNN backbones for a near features shift.

| Backbone | Model | Roman Empire AUC-ROC (Alea. / Epi.) | Roman Empire Acc. | Amazon Ratings AUC-ROC (Alea. / Epi.) | Amazon Ratings Acc. | Chameleon AUC-ROC (Alea. / Epi.) | Chameleon Acc. | Squirrel AUC-ROC (Alea. / Epi.) | Squirrel Acc. | CoraML AUC-ROC (Alea. / Epi.) | CoraML Acc. | PubMed AUC-ROC (Alea. / Epi.) | PubMed Acc. |
|---|---|---|---|---|---|---|---|---|---|---|---|---|---|
| Res-GCN | Ens. | $78.4_{\pm0.6}$/$78.2_{\pm0.7}$ | | $60.3_{\pm1.6}$/$63.6_{\pm1.4}$ | | $55.2_{\pm5.4}$/$53.8_{\pm5.8}$ | | $53.0_{\pm3.9}$/$50.6_{\pm5.4}$ | | $52.7_{\pm2.3}$/$52.8_{\pm2.4}$ | | $64.0_{\pm2.1}$/$64.7_{\pm2.6}$ | |
| | MCD | $75.8_{\pm1.0}$/$72.5_{\pm1.0}$ | | $56.6_{\pm2.1}$/$58.2_{\pm1.9}$ | | $56.0_{\pm5.5}$/$55.2_{\pm7.0}$ | | $52.7_{\pm4.1}$/$51.2_{\pm4.8}$ | | $54.6_{\pm2.5}$/$55.1_{\pm2.4}$ | | $63.3_{\pm2.4}$/$63.4_{\pm2.3}$ | |
| | EBM | $74.5_{\pm1.0}$/$81.3_{\pm0.9}$ | $72.3_{\pm0.6}$ | $54.8_{\pm1.9}$/$56.1_{\pm2.3}$ | $48.2_{\pm0.5}$ | $54.6_{\pm5.8}$/$55.9_{\pm6.3}$ | $40.7_{\pm3.9}$ | $52.6_{\pm3.8}$/$52.7_{\pm4.1}$ | $41.8_{\pm2.2}$ | $52.1_{\pm2.5}$/$53.3_{\pm2.7}$ | $79.6_{\pm1.5}$ | $61.9_{\pm2.9}$/$61.0_{\pm5.4}$ | $83.9_{\pm0.5}$ |
| | Safe | $74.5_{\pm1.0}$/$73.4_{\pm1.3}$ | | $54.8_{\pm1.9}$/$55.3_{\pm2.4}$ | | $54.6_{\pm5.8}$/$55.7_{\pm7.6}$ | | $52.6_{\pm3.8}$/$51.6_{\pm4.1}$ | | $52.1_{\pm2.5}$/$51.0_{\pm1.9}$ | | $61.9_{\pm2.9}$/$54.0_{\pm1.8}$ | |
| | GEBM | $74.5_{\pm1.0}$/$66.4_{\pm3.0}$ | | $54.8_{\pm1.9}$/$47.1_{\pm1.5}$ | | $54.6_{\pm5.8}$/$37.3_{\pm14.4}$ | | $52.6_{\pm3.8}$/$28.6_{\pm5.8}$ | | $52.1_{\pm2.5}$/$53.3_{\pm10.0}$ | | $61.9_{\pm2.9}$/$62.1_{\pm5.0}$ | |
| | JLDE | $74.5_{\pm1.0}$/$93.8_{\pm0.9}$ | | $54.8_{\pm1.9}$/$65.4_{\pm1.3}$ | | $54.6_{\pm5.8}$/$51.1_{\pm7.6}$ | | $52.6_{\pm3.8}$/$43.1_{\pm4.7}$ | | $52.1_{\pm2.5}$/$54.9_{\pm2.1}$ | | $61.9_{\pm2.9}$/$64.3_{\pm3.0}$ | |
| Res-GAT-Sep | Ens. | $81.6_{\pm0.7}$/$81.5_{\pm0.8}$ | | $62.0_{\pm0.9}$/$63.0_{\pm0.8}$ | | $58.0_{\pm11.6}$/$38.1_{\pm8.1}$ | | $72.8_{\pm10.2}$/$69.5_{\pm7.1}$ | | $53.6_{\pm3.0}$/$53.4_{\pm3.2}$ | | $66.6_{\pm2.3}$/$65.5_{\pm2.3}$ | |
| | MCD | $81.2_{\pm0.8}$/$81.1_{\pm0.8}$ | | $57.8_{\pm1.4}$/$59.8_{\pm1.1}$ | | $58.6_{\pm9.6}$/$50.3_{\pm8.5}$ | | $59.3_{\pm13.1}$/$52.5_{\pm6.8}$ | | $56.8_{\pm3.2}$/$56.5_{\pm3.1}$ | | $63.2_{\pm2.9}$/$63.4_{\pm2.5}$ | |
| | EBM | $77.7_{\pm1.0}$/$81.3_{\pm1.1}$ | $78.9_{\pm0.6}$ | $56.3_{\pm1.3}$/$56.5_{\pm1.5}$ | $50.8_{\pm0.6}$ | $58.7_{\pm10.3}$/$58.1_{\pm9.4}$ | $34.5_{\pm4.1}$ | $65.9_{\pm20.4}$/$70.2_{\pm20.8}$ | $34.8_{\pm1.6}$ | $53.7_{\pm3.0}$/$55.9_{\pm3.4}$ | $78.1_{\pm1.8}$ | $65.0_{\pm3.4}$/$64.2_{\pm6.8}$ | $83.2_{\pm0.6}$ |
| | Safe | $77.7_{\pm1.0}$/$77.8_{\pm1.0}$ | | $56.3_{\pm1.3}$/$58.2_{\pm1.5}$ | | $58.7_{\pm10.3}$/$55.0_{\pm8.7}$ | | $65.9_{\pm20.4}$/$56.6_{\pm7.9}$ | | $53.7_{\pm3.0}$/$51.7_{\pm2.3}$ | | $65.0_{\pm3.4}$/$55.4_{\pm2.4}$ | |
| | GEBM | $77.7_{\pm1.0}$/$70.8_{\pm1.1}$ | | $56.3_{\pm1.3}$/$48.6_{\pm1.5}$ | | $58.7_{\pm10.3}$/$53.4_{\pm17.9}$ | | $65.9_{\pm20.4}$/$45.4_{\pm20.2}$ | | $53.7_{\pm3.0}$/$55.7_{\pm13.2}$ | | $65.0_{\pm3.4}$/$66.8_{\pm5.4}$ | |
| | JLDE | $77.7_{\pm1.0}$/$92.7_{\pm0.5}$ | | $56.3_{\pm1.3}$/$79.1_{\pm1.1}$ | | $58.7_{\pm10.3}$/$33.2_{\pm13.8}$ | | $65.9_{\pm20.4}$/$40.7_{\pm20.7}$ | | $53.7_{\pm3.0}$/$60.9_{\pm3.6}$ | | $65.0_{\pm3.4}$/$76.1_{\pm5.2}$ | |
| Res-GATE | Ens. | $84.3_{\pm0.9}$/$84.2_{\pm0.8}$ | | $61.7_{\pm1.2}$/$63.0_{\pm1.1}$ | | $55.8_{\pm5.1}$/$51.6_{\pm5.5}$ | | $56.5_{\pm7.2}$/$46.4_{\pm4.8}$ | | $51.1_{\pm2.4}$/$51.1_{\pm2.5}$ | | $66.5_{\pm2.0}$/$65.9_{\pm2.6}$ | |
| | MCD | $80.7_{\pm1.1}$/$79.7_{\pm1.0}$ | | $57.2_{\pm1.6}$/$59.4_{\pm1.4}$ | | $56.2_{\pm7.9}$/$54.1_{\pm7.2}$ | | $53.5_{\pm6.2}$/$50.6_{\pm5.0}$ | | $52.7_{\pm2.4}$/$53.1_{\pm2.4}$ | | $65.2_{\pm2.5}$/$65.8_{\pm2.5}$ | |
| | EBM | $77.8_{\pm1.2}$/$83.5_{\pm1.2}$ | $77.5_{\pm0.5}$ | $54.5_{\pm1.5}$/$54.3_{\pm1.5}$ | $49.5_{\pm0.7}$ | $55.2_{\pm7.5}$/$56.3_{\pm8.2}$ | $37.3_{\pm3.6}$ | $53.8_{\pm6.4}$/$54.2_{\pm6.9}$ | $34.0_{\pm1.6}$ | $50.9_{\pm2.5}$/$52.0_{\pm2.5}$ | $80.6_{\pm1.5}$ | $64.6_{\pm1.3}$/$63.9_{\pm6.5}$ | $84.3_{\pm0.5}$ |
| | Safe | $77.8_{\pm1.2}$/$78.5_{\pm0.9}$ | | $54.5_{\pm1.5}$/$56.4_{\pm1.5}$ | | $55.2_{\pm7.5}$/$54.6_{\pm8.3}$ | | $53.8_{\pm6.4}$/$52.1_{\pm5.5}$ | | $50.9_{\pm2.5}$/$50.7_{\pm2.0}$ | | $64.6_{\pm1.3}$/$54.9_{\pm2.0}$ | |
| | GEBM | $77.8_{\pm1.2}$/$64.2_{\pm2.8}$ | | $54.5_{\pm1.5}$/$47.2_{\pm1.6}$ | | $55.2_{\pm7.5}$/$55.9_{\pm12.1}$ | | $53.8_{\pm6.4}$/$34.6_{\pm11.4}$ | | $50.9_{\pm2.5}$/$52.8_{\pm10.2}$ | | $64.6_{\pm1.3}$/$63.2_{\pm4.7}$ | |
| | JLDE | $77.8_{\pm1.2}$/$88.8_{\pm0.9}$ | | $54.5_{\pm1.5}$/$63.5_{\pm2.0}$ | | $55.2_{\pm7.5}$/$51.3_{\pm7.2}$ | | $53.8_{\pm6.4}$/$38.4_{\pm6.5}$ | | $50.9_{\pm2.5}$/$54.0_{\pm2.3}$ | | $64.6_{\pm1.3}$/$75.0_{\pm4.2}$ | |
| GPRGNN | Ens. | $87.4_{\pm0.6}$/$84.2_{\pm1.0}$ | | $68.3_{\pm1.5}$/$77.4_{\pm1.0}$ | | $63.4_{\pm7.2}$/$33.5_{\pm4.8}$ | | $54.4_{\pm4.4}$/$45.6_{\pm2.9}$ | | $60.9_{\pm2.0}$/$51.5_{\pm3.0}$ | | $66.4_{\pm1.1}$/$61.0_{\pm3.1}$ | |
| | MCD | $89.3_{\pm0.9}$/$58.4_{\pm2.4}$ | | $62.3_{\pm1.8}$/$66.8_{\pm2.5}$ | | $62.5_{\pm7.1}$/$27.2_{\pm6.3}$ | | $53.5_{\pm5.1}$/$30.4_{\pm4.9}$ | | $59.7_{\pm2.1}$/$45.3_{\pm4.9}$ | | $61.8_{\pm1.1}$/$47.8_{\pm2.9}$ | |
| | EBM | $86.5_{\pm0.9}$/$87.9_{\pm2.6}$ | $64.8_{\pm0.6}$ | $62.1_{\pm1.2}$/$70.0_{\pm1.3}$ | $50.6_{\pm0.7}$ | $62.9_{\pm7.2}$/$60.8_{\pm6.5}$ | $37.1_{\pm3.6}$ | $53.1_{\pm4.9}$/$57.0_{\pm6.9}$ | $35.4_{\pm1.4}$ | $61.0_{\pm1.9}$/$65.9_{\pm1.9}$ | $84.1_{\pm1.2}$ | $66.5_{\pm1.2}$/$72.9_{\pm1.5}$ | $86.0_{\pm0.3}$ |
| | Safe | $86.5_{\pm0.9}$/$72.9_{\pm1.4}$ | | $62.1_{\pm1.2}$/$56.0_{\pm1.3}$ | | $62.9_{\pm7.2}$/$53.6_{\pm8.1}$ | | $53.1_{\pm4.9}$/$51.4_{\pm4.8}$ | | $61.0_{\pm1.9}$/$54.9_{\pm1.8}$ | | $66.5_{\pm1.2}$/$55.2_{\pm0.7}$ | |
| | GEBM | $86.5_{\pm0.9}$/$43.6_{\pm1.4}$ | | $62.1_{\pm1.2}$/$49.7_{\pm1.2}$ | | $62.9_{\pm7.2}$/$41.4_{\pm5.2}$ | | $53.1_{\pm4.9}$/$43.4_{\pm11.3}$ | | $61.0_{\pm1.9}$/$76.2_{\pm2.9}$ | | $66.5_{\pm1.2}$/$62.2_{\pm3.4}$ | |
| | JLDE | $86.5_{\pm0.9}$/$85.6_{\pm1.7}$ | | $62.1_{\pm1.2}$/$47.8_{\pm2.4}$ | | $62.9_{\pm7.2}$/$55.2_{\pm9.5}$ | | $53.1_{\pm4.9}$/$38.8_{\pm11.0}$ | | $61.0_{\pm1.9}$/$63.3_{\pm2.3}$ | | $66.5_{\pm1.2}$/$65.8_{\pm3.7}$ | |

Table 11. O.o.d. detection AUC-ROC using different uncertainty estimators for different GNN backbones for a far features shift.

| Backbone | Model | Roman Empire AUC-ROC (Alea./Epi.) | Roman Empire Acc.↑ | Amazon Ratings AUC-ROC (Alea./Epi.) | Amazon Ratings Acc.↑ | Chameleon AUC-ROC (Alea./Epi.) | Chameleon Acc.↑ | Squirrel AUC-ROC (Alea./Epi.) | Squirrel Acc.↑ | CoraML AUC-ROC (Alea./Epi.) | CoraML Acc.↑ | PubMed AUC-ROC (Alea./Epi.) | PubMed Acc.↑ |
|---|---|---|---|---|---|---|---|---|---|---|---|---|---|
| Res-GCN | Ens. | $89.2_{\pm0.5}$/$86.9_{\pm0.7}$ | $70.6_{\pm0.6}$ | $74.3_{\pm0.9}$/$50.4_{\pm2.1}$ | $47.3_{\pm0.6}$ | $81.2_{\pm6.9}$/$51.0_{\pm10.9}$ | $40.1_{\pm3.3}$ | $71.0_{\pm2.2}$/$70.1_{\pm13.6}$ | $40.5_{\pm1.9}$ | $84.7_{\pm2.1}$/$78.4_{\pm1.8}$ | $77.1_{\pm1.8}$ | $82.0_{\pm0.5}$/$81.4_{\pm0.6}$ | $82.9_{\pm0.4}$ |
| | MCD | $82.7_{\pm0.8}$/$72.9_{\pm1.8}$ | | $72.9_{\pm1.6}$/$33.2_{\pm4.8}$ | | $82.2_{\pm9.1}$/$41.1_{\pm10.8}$ | | $70.1_{\pm4.5}$/$40.0_{\pm6.7}$ | | $88.0_{\pm1.9}$/$64.0_{\pm9.3}$ | | $85.3_{\pm1.1}$/$74.9_{\pm1.1}$ | |
| | EBM | $83.0_{\pm0.8}$/**$92.0_{\pm1.1}$** | | $78.2_{\pm2.1}$/**$94.4_{\pm1.2}$** | | $78.4_{\pm8.9}$/**$92.5_{\pm7.1}$** | | $69.9_{\pm4.1}$/$86.7_{\pm9.4}$ | | $85.0_{\pm2.2}$/**$93.5_{\pm1.5}$** | | $81.0_{\pm1.3}$/**$88.5_{\pm2.3}$** | |
| | Safe | $83.0_{\pm0.8}$/$81.0_{\pm1.3}$ | | $78.2_{\pm2.1}$/$67.2_{\pm2.5}$ | | $78.4_{\pm8.9}$/$68.6_{\pm7.8}$ | | $69.9_{\pm4.7}$/$62.4_{\pm6.4}$ | | $85.0_{\pm2.2}$/$72.0_{\pm2.4}$ | | $81.0_{\pm1.3}$/$66.3_{\pm3.3}$ | |
| | GEBM | $83.0_{\pm0.8}$/$80.9_{\pm3.3}$ | | $78.2_{\pm2.1}$/$58.9_{\pm7.4}$ | | $78.4_{\pm8.9}$/$85.9_{\pm7.3}$ | | $69.9_{\pm4.7}$/**$96.6_{\pm4.5}$** | | $85.0_{\pm2.2}$/$80.5_{\pm3.2}$ | | $81.0_{\pm1.3}$/$74.2_{\pm3.9}$ | |
| | JLDE | $83.0_{\pm0.8}$/**$100.0_{\pm0.0}$** | | $78.2_{\pm2.1}$/**$99.2_{\pm0.2}$** | | $78.4_{\pm8.9}$/**$99.9_{\pm0.2}$** | | $69.9_{\pm4.7}$/**$100.0_{\pm0.0}$** | | $85.0_{\pm2.2}$/**$95.7_{\pm0.8}$** | | $81.0_{\pm1.3}$/**$99.4_{\pm0.1}$** | |
| Res-GAT-Sep | Ens. | $84.1_{\pm0.6}$/$83.3_{\pm0.6}$ | $79.3_{\pm0.6}$ | $62.9_{\pm1.2}$/$52.5_{\pm1.3}$ | $50.8_{\pm0.6}$ | $64.7_{\pm8.2}$/$38.7_{\pm4.8}$ | $33.9_{\pm4.0}$ | $14.6_{\pm14.4}$/$51.9_{\pm35.5}$ | $34.5_{\pm2.0}$ | $83.1_{\pm1.6}$/$76.7_{\pm1.5}$ | $75.0_{\pm1.8}$ | $81.6_{\pm0.8}$/$81.1_{\pm0.8}$ | $81.4_{\pm0.5}$ |
| | MCD | $85.0_{\pm0.7}$/$79.4_{\pm1.2}$ | | $65.8_{\pm2.0}$/$34.3_{\pm1.8}$ | | $60.6_{\pm13.8}$/$32.8_{\pm9.2}$ | | $26.4_{\pm31.0}$/$32.3_{\pm25.8}$ | | $87.2_{\pm2.1}$/$66.7_{\pm4.4}$ | | **$84.3_{\pm1.1}$**/$78.0_{\pm0.9}$ | |
| | EBM | $82.0_{\pm1.0}$/**$88.8_{\pm1.1}$** | | $70.1_{\pm2.2}$/**$80.0_{\pm2.4}$** | | $64.8_{\pm10.6}$/$78.1_{\pm16.2}$ | | $26.6_{\pm32.2}$/$21.3_{\pm27.8}$ | | $83.9_{\pm1.9}$/**$92.6_{\pm1.8}$** | | $79.4_{\pm1.7}$/**$84.3_{\pm3.3}$** | |
| | Safe | $82.0_{\pm1.0}$/$81.0_{\pm1.0}$ | | $70.1_{\pm2.2}$/$66.2_{\pm1.7}$ | | $64.8_{\pm10.6}$/$66.1_{\pm12.6}$ | | $26.6_{\pm32.2}$/$27.5_{\pm22.8}$ | | $83.9_{\pm1.9}$/$70.9_{\pm2.4}$ | | $79.4_{\pm1.7}$/$64.7_{\pm3.1}$ | |
| | GEBM | $82.0_{\pm1.0}$/$76.5_{\pm1.9}$ | | $70.1_{\pm2.2}$/$63.4_{\pm2.1}$ | | $64.8_{\pm10.6}$/**$93.9_{\pm6.5}$** | | $26.6_{\pm32.2}$/**$95.8_{\pm6.6}$** | | $83.9_{\pm1.9}$/$85.3_{\pm3.2}$ | | $79.4_{\pm1.7}$/$79.5_{\pm4.7}$ | |
| | JLDE | $82.0_{\pm1.0}$/**$96.8_{\pm0.3}$** | | $70.1_{\pm2.2}$/**$98.5_{\pm0.3}$** | | $64.8_{\pm10.6}$/**$98.2_{\pm3.8}$** | | $26.6_{\pm32.2}$/**$99.9_{\pm0.3}$** | | $83.9_{\pm1.9}$/**$96.8_{\pm0.9}$** | | $79.4_{\pm1.7}$/**$99.8_{\pm0.1}$** | |
| Res-GATE | Ens. | $96.6_{\pm0.3}$/$83.3_{\pm0.6}$ | $76.4_{\pm0.5}$ | $63.4_{\pm1.0}$/$54.5_{\pm1.0}$ | $49.3_{\pm0.8}$ | $71.3_{\pm7.3}$/$49.9_{\pm9.7}$ | $36.7_{\pm4.0}$ | $82.4_{\pm16.4}$/$86.1_{\pm23.0}$ | $33.7_{\pm1.5}$ | $84.5_{\pm1.9}$/$78.7_{\pm1.5}$ | $78.2_{\pm1.7}$ | $80.6_{\pm0.6}$/$79.4_{\pm0.4}$ | $83.6_{\pm0.5}$ |
| | MCD | $93.0_{\pm0.5}$/$73.5_{\pm1.0}$ | | $68.7_{\pm1.6}$/$39.3_{\pm2.0}$ | | $70.8_{\pm11.5}$/$44.4_{\pm7.1}$ | | $60.4_{\pm14.5}$/$51.0_{\pm21.3}$ | | $89.3_{\pm2.3}$/$69.5_{\pm7.2}$ | | $84.0_{\pm1.2}$/$75.7_{\pm1.4}$ | |
| | EBM | $94.9_{\pm0.5}$/**$99.7_{\pm0.1}$** | | $69.8_{\pm1.9}$/**$80.7_{\pm2.3}$** | | $67.7_{\pm11.0}$/$83.7_{\pm12.1}$ | | $65.0_{\pm15.0}$/$56.4_{\pm19.1}$ | | $85.2_{\pm2.5}$/**$94.2_{\pm2.1}$** | | $79.5_{\pm1.6}$/**$86.3_{\pm2.9}$** | |
| | Safe | $94.9_{\pm0.5}$/$90.7_{\pm0.4}$ | | $69.8_{\pm1.9}$/$67.4_{\pm2.1}$ | | $67.7_{\pm11.0}$/$66.9_{\pm11.1}$ | | $65.0_{\pm15.0}$/$53.5_{\pm15.0}$ | | $85.2_{\pm2.5}$/$70.8_{\pm2.2}$ | | $79.5_{\pm1.6}$/$64.4_{\pm2.5}$ | |
| | GEBM | $94.9_{\pm0.5}$/$89.3_{\pm2.4}$ | | $69.8_{\pm1.9}$/$56.0_{\pm2.7}$ | | $67.7_{\pm11.0}$/**$88.3_{\pm3.3}$** | | $65.0_{\pm15.0}$/**$99.3_{\pm3.3}$** | | $85.2_{\pm2.5}$/$80.5_{\pm3.9}$ | | $79.5_{\pm1.6}$/$75.0_{\pm4.4}$ | |
| | JLDE | $94.9_{\pm0.5}$/**$100.0_{\pm0.0}$** | | $69.8_{\pm1.9}$/**$93.9_{\pm0.9}$** | | $67.7_{\pm11.0}$/**$99.5_{\pm1.2}$** | | $65.0_{\pm15.0}$/**$100.0_{\pm0.0}$** | | $85.2_{\pm2.5}$/**$95.1_{\pm2.2}$** | | $79.5_{\pm1.6}$/**$99.9_{\pm0.0}$** | |
| GPRGNN | Ens. | $44.4_{\pm1.8}$/$88.6_{\pm1.3}$ | $61.0_{\pm0.6}$ | $51.5_{\pm1.3}$/$68.4_{\pm3.2}$ | $34.1_{\pm2.7}$ | $24.2_{\pm12.6}$/$55.0_{\pm18.0}$ | $30.5_{\pm4.1}$ | $8.5_{\pm3.3}$/$90.6_{\pm14.2}$ | $34.4_{\pm1.9}$ | $32.9_{\pm5.6}$/$43.6_{\pm5.0}$ | $37.3_{\pm4.2}$ | $33.8_{\pm5.6}$/$38.8_{\pm5.6}$ | $51.5_{\pm4.5}$ |
| | MCD | $87.7_{\pm2.9}$/**$99.3_{\pm0.4}$** | | $60.7_{\pm1.9}$/$78.9_{\pm3.0}$ | | $56.0_{\pm14.4}$/$93.1_{\pm4.4}$ | | $10.5_{\pm11.4}$/**$99.8_{\pm0.3}$** | | $75.3_{\pm4.3}$/$90.3_{\pm1.3}$ | | $70.7_{\pm5.6}$/$82.6_{\pm2.0}$ | |
| | EBM | $17.6_{\pm1.3}$/$79.1_{\pm4.9}$ | | $10.9_{\pm3.1}$/$48.6_{\pm10.9}$ | | $15.9_{\pm7.6}$/$7.9_{\pm7.2}$ | | $14.7_{\pm11.3}$/$18.4_{\pm17.6}$ | | $17.5_{\pm2.9}$/$4.8_{\pm0.8}$ | | $18.6_{\pm7.5}$/$8.5_{\pm1.3}$ | |
| | Safe | $17.6_{\pm1.3}$/$80.4_{\pm2.9}$ | | $10.9_{\pm3.1}$/$50.9_{\pm9.0}$ | | $15.9_{\pm7.6}$/$26.8_{\pm7.7}$ | | $14.7_{\pm11.3}$/$28.8_{\pm12.8}$ | | $17.5_{\pm2.9}$/$18.7_{\pm1.4}$ | | $18.6_{\pm7.5}$/$24.0_{\pm1.3}$ | |
| | GEBM | $17.6_{\pm1.3}$/$98.3_{\pm0.3}$ | | $10.9_{\pm3.1}$/**$99.4_{\pm0.2}$** | | $15.9_{\pm7.6}$/**$94.3_{\pm3.3}$** | | $14.7_{\pm11.3}$/$89.6_{\pm6.2}$ | | $17.5_{\pm2.9}$/**$97.3_{\pm0.6}$** | | $18.6_{\pm7.5}$/**$92.0_{\pm1.0}$** | |
| | JLDE | $17.6_{\pm1.3}$/**$100.0_{\pm0.0}$** | | $10.9_{\pm3.1}$/**$100.0_{\pm0.0}$** | | $15.9_{\pm7.6}$/**$97.0_{\pm1.8}$** | | $14.7_{\pm11.3}$/**$96.7_{\pm2.3}$** | | $17.5_{\pm2.9}$/**$98.3_{\pm0.2}$** | | $18.6_{\pm7.5}$/**$97.7_{\pm0.5}$** | |

Uncertainty Estimation for Heterophilic Graphs Through the Lens of Information Theory

Table 12. O.o.d. detection AUC-ROC rank of model-agnostic epistemic and aleatoric estimators averaged over all distribution shifts (best and runner-up).

| | Model | Roman Empire | Amazon Ratings | Chameleon | Squirrel | CoraML | PubMed |
|---|---|---|---|---|---|---|---|
| Res-GCN | Ens. | 2.5 / 4.0 | 2.8 / 4.2 | 3.8 / 6.8 | 3.5 / 5.2 | 2.8 / 4.5 | 2.5 / 3.0 |
| | MCD | 4.2 / 8.2 | 3.8 / 5.5 | 4.2 / 6.0 | 4.8 / 7.8 | 4.2 / 6.8 | 4.8 / 9.0 |
| | EBM | 3.0 / 6.0 | 4.0 / 5.5 | 3.8 / 4.0 | 4.0 / 7.2 | 4.2 / 7.5 | 4.2 / 6.2 |
| | Safe | 4.5 / 10.5 | 5.0 / 10.0 | 4.0 / 5.8 | 3.8 / 6.8 | 4.8 / 8.2 | 4.0 / 7.0 |
| | GEBM | 5.8 / 11.8 | 4.5 / 8.0 | 2.8 / 4.2 | 2.5 / 4.0 | 2.8 / 4.5 | 2.8 / 4.8 |
| | JLDE | 1.0 / 1.0 | 1.0 / 1.0 | 2.5 / 4.0 | 2.5 / 4.0 | 2.2 / 2.8 | 2.8 / 3.2 |
| Res-GAT-Sep | Ens. | 1.8 / 3.0 | 2.8 / 4.8 | 5.5 / 9.0 | 2.8 / 3.0 | 2.8 / 5.8 | 2.5 / 3.8 |
| | MCD | 3.2 / 6.2 | 3.8 / 6.0 | 4.0 / 7.0 | 4.0 / 5.5 | 4.0 / 6.2 | 4.5 / 8.2 |
| | EBM | 3.2 / 6.5 | 3.8 / 7.2 | 3.5 / 4.8 | 4.2 / 5.8 | 4.2 / 7.5 | 4.0 / 6.2 |
| | Safe | 4.8 / 9.8 | 4.2 / 8.5 | 3.5 / 5.0 | 5.0 / 6.5 | 3.8 / 7.2 | 4.5 / 8.0 |
| | GEBM | 6.0 / 12.0 | 5.5 / 11.2 | 1.8 / 3.2 | 2.8 / 4.2 | 3.2 / 4.2 | 2.0 / 2.5 |
| | JLDE | 2.0 / 3.0 | 1.0 / 1.0 | 2.8 / 4.2 | 2.2 / 3.8 | 3.0 / 4.0 | 3.5 / 6.5 |
| Res-GATE | Ens. | 2.8 / 4.5 | 2.8 / 4.8 | 4.5 / 7.5 | 3.8 / 5.2 | 3.2 / 5.0 | 3.2 / 5.5 |
| | MCD | 4.0 / 7.0 | 4.2 / 6.5 | 4.0 / 7.0 | 3.8 / 6.8 | 4.5 / 7.0 | 4.8 / 9.0 |
| | EBM | 3.2 / 6.5 | 4.2 / 8.8 | 4.0 / 4.0 | 4.2 / 6.0 | 4.5 / 7.8 | 4.0 / 6.0 |
| | Safe | 4.5 / 9.5 | 3.2 / 5.5 | 4.2 / 7.2 | 4.2 / 7.2 | 4.8 / 8.8 | 4.5 / 7.5 |
| | GEBM | 5.5 / 11.5 | 5.5 / 11.5 | 2.0 / 2.2 | 3.0 / 4.5 | 3.0 / 5.0 | 3.0 / 6.0 |
| | JLDE | 1.0 / 1.0 | 1.0 / 1.0 | 2.2 / 3.8 | 2.0 / 3.5 | 1.0 / 1.0 | 1.5 / 1.5 |
| GPRGNN | Ens. | 3.2 / 4.8 | 2.8 / 2.8 | 5.2 / 10.0 | 3.5 / 5.0 | 5.2 / 10.0 | 3.5 / 5.2 |
| | MCD | 4.2 / 8.8 | 3.5 / 6.2 | 3.8 / 5.2 | 2.8 / 4.2 | 4.8 / 9.2 | 4.8 / 9.2 |
| | EBM | 3.8 / 7.2 | 4.5 / 8.0 | 4.2 / 10.2 | 4.2 / 6.8 | 3.5 / 5.0 | 3.8 / 5.8 |
| | Safe | 3.2 / 5.0 | 5.2 / 10.2 | 4.0 / 6.2 | 4.8 / 9.2 | 2.8 / 4.8 | 3.5 / 5.5 |
| | GEBM | 5.2 / 9.8 | 2.8 / 4.2 | 2.0 / 3.5 | 2.5 / 4.0 | 1.8 / 1.8 | 1.8 / 3.0 |
| | JLDE | 1.2 / 2.8 | 2.2 / 3.8 | 1.8 / 3.2 | 3.2 / 4.8 | 3.0 / 6.0 | 3.8 / 8.0 |

*Table 13.* Misclassification detection AUC-ROC using different uncertainty estimators and our method using a Res-GCN backbone.

| | Model | LoC AUC-ROC (Alea. / Epi.) | LoC Acc.↑ | Near-Features AUC-ROC (Alea. / Epi.) | Near Acc.↑ | Far-Features AUC-ROC (Alea. / Epi.) | Far Acc.↑ |
|---|---|---|---|---|---|---|---|
| **Roman Empire** | GPN | $72.4_{\pm2.6}/57.1_{\pm4.9}$ | $48.6_{\pm1.9}$ | $71.1_{\pm3.2}/57.1_{\pm3.2}$ | $38.1_{\pm1.3}$ | $70.4_{\pm2.9}/58.7_{\pm3.1}$ | $38.0_{\pm1.3}$ |
| | SGCN | $66.3_{\pm1.4}/58.8_{\pm0.9}$ | $58.3_{\pm1.6}$ | $66.7_{\pm1.3}/60.6_{\pm0.8}$ | $45.5_{\pm1.6}$ | $36.6_{\pm0.9}/32.7_{\pm1.3}$ | $30.6_{\pm1.0}$ |
| | BGCN | $65.6_{\pm1.0}/61.1_{\pm1.6}$ | $53.0_{\pm0.7}$ | $66.0_{\pm0.9}/59.1_{\pm1.9}$ | $42.4_{\pm0.5}$ | $50.6_{\pm2.4}/67.6_{\pm2.0}$ | $28.8_{\pm0.6}$ |
| | GPRGNN | $85.0_{\pm0.8}/78.3_{\pm0.6}$ | $78.1_{\pm0.4}$ | $84.8_{\pm0.4}/78.9_{\pm0.6}$ | $64.8_{\pm0.6}$ | $68.4_{\pm0.7}/77.3_{\pm1.2}$ | $61.0_{\pm0.6}$ |
| | FSGNN | $88.1_{\pm0.5}/77.2_{\pm1.1}$ | $86.6_{\pm0.4}$ | $83.0_{\pm0.6}/77.1_{\pm0.7}$ | $71.6_{\pm0.5}$ | $55.8_{\pm1.5}/34.2_{\pm1.3}$ | $56.9_{\pm0.9}$ |
| | FAGCN | $86.8_{\pm1.1}/77.6_{\pm2.8}$ | $83.6_{\pm1.1}$ | $84.2_{\pm0.6}/78.0_{\pm1.0}$ | $68.0_{\pm0.9}$ | $48.0_{\pm4.1}/27.1_{\pm4.7}$ | $44.3_{\pm2.8}$ |
| | Ens. | $88.8_{\pm0.4}/88.3_{\pm0.4}$ | | $85.5_{\pm0.6}/84.8_{\pm0.4}$ | | $88.1_{\pm0.4}/87.1_{\pm0.4}$ | |
| | MCD | $89.0_{\pm0.4}/88.3_{\pm0.7}$ | | $84.5_{\pm0.6}/82.9_{\pm0.8}$ | | $86.4_{\pm0.5}/83.5_{\pm0.8}$ | |
| | EBM | $87.0_{\pm0.8}/79.8_{\pm2.8}$ | $87.6_{\pm0.7}$ | $83.2_{\pm0.6}/80.7_{\pm1.2}$ | $72.3_{\pm0.6}$ | $85.3_{\pm0.5}/83.7_{\pm1.1}$ | $70.6_{\pm0.6}$ |
| | Safe | $87.0_{\pm0.8}/71.6_{\pm1.9}$ | | $83.2_{\pm0.6}/73.6_{\pm0.9}$ | | $85.3_{\pm0.5}/75.6_{\pm0.8}$ | |
| | GEBM | $87.0_{\pm0.8}/45.2_{\pm3.4}$ | | $83.2_{\pm0.6}/50.9_{\pm2.9}$ | | $85.3_{\pm0.5}/56.9_{\pm3.0}$ | |
| | JLDE | $87.0_{\pm0.8}/84.2_{\pm1.1}$ | | $83.2_{\pm0.6}/83.6_{\pm0.7}$ | | $85.3_{\pm0.5}/86.1_{\pm0.5}$ | |
| **Amazon Ratings** | GPN | $60.7_{\pm0.9}/57.2_{\pm1.7}$ | $54.2_{\pm1.2}$ | $60.7_{\pm0.7}/56.5_{\pm1.1}$ | $46.8_{\pm0.7}$ | $60.7_{\pm0.7}/56.5_{\pm1.2}$ | $47.0_{\pm0.7}$ |
| | SGCN | $60.7_{\pm0.9}/59.7_{\pm0.8}$ | $55.5_{\pm0.7}$ | $60.6_{\pm1.1}/57.5_{\pm1.0}$ | $47.2_{\pm0.5}$ | $48.8_{\pm1.5}/47.3_{\pm1.6}$ | $33.5_{\pm1.3}$ |
| | BGCN | $60.1_{\pm0.8}/53.3_{\pm1.2}$ | $51.4_{\pm0.8}$ | $61.0_{\pm1.1}/53.0_{\pm2.1}$ | $44.2_{\pm1.3}$ | $52.9_{\pm1.6}/55.5_{\pm1.5}$ | $33.5_{\pm1.3}$ |
| | GPRGNN | $63.5_{\pm0.9}/61.4_{\pm1.2}$ | $57.9_{\pm0.8}$ | $64.2_{\pm0.7}/63.4_{\pm1.1}$ | $50.6_{\pm0.7}$ | $48.7_{\pm1.5}/54.4_{\pm1.6}$ | $34.1_{\pm2.7}$ |
| | FSGNN | $63.2_{\pm0.6}/62.8_{\pm0.7}$ | $56.8_{\pm0.6}$ | $63.5_{\pm0.5}/60.8_{\pm0.6}$ | $49.1_{\pm0.6}$ | $48.6_{\pm0.8}/43.5_{\pm0.8}$ | $38.6_{\pm0.8}$ |
| | FAGCN | $64.2_{\pm0.7}/62.5_{\pm1.2}$ | $58.1_{\pm0.8}$ | $64.6_{\pm0.8}/61.8_{\pm1.1}$ | $51.2_{\pm0.7}$ | $47.9_{\pm1.4}/44.6_{\pm2.8}$ | $31.7_{\pm2.4}$ |
| | Ens. | $65.1_{\pm0.6}/62.5_{\pm0.5}$ | | $65.5_{\pm0.8}/63.9_{\pm0.8}$ | | $65.6_{\pm0.8}/63.9_{\pm0.7}$ | |
| | MCD | $63.9_{\pm0.8}/58.6_{\pm1.0}$ | | $64.1_{\pm0.8}/62.3_{\pm1.4}$ | | $64.5_{\pm0.9}/61.4_{\pm1.5}$ | |
| | EBM | $63.8_{\pm0.8}/62.6_{\pm1.0}$ | $56.0_{\pm0.6}$ | $63.9_{\pm0.9}/64.8_{\pm0.9}$ | $48.2_{\pm0.5}$ | $64.0_{\pm0.9}/64.4_{\pm0.9}$ | $47.3_{\pm0.6}$ |
| | Safe | $63.8_{\pm0.8}/60.7_{\pm1.2}$ | | $63.9_{\pm0.9}/62.5_{\pm0.8}$ | | $64.0_{\pm0.9}/62.6_{\pm0.8}$ | |
| | GEBM | $63.8_{\pm0.8}/49.7_{\pm2.0}$ | | $63.9_{\pm0.9}/52.1_{\pm1.7}$ | | $64.0_{\pm0.9}/52.5_{\pm1.5}$ | |
| | JLDE | $63.8_{\pm0.8}/58.6_{\pm0.9}$ | | $63.9_{\pm0.9}/61.7_{\pm2.1}$ | | $64.0_{\pm0.9}/61.5_{\pm1.9}$ | |
| **Chameleon** | GPN | $55.8_{\pm6.6}/44.3_{\pm5.1}$ | $39.6_{\pm3.8}$ | $66.7_{\pm3.9}/42.3_{\pm5.1}$ | $37.0_{\pm3.8}$ | $67.1_{\pm4.0}/42.5_{\pm4.9}$ | $37.0_{\pm3.8}$ |
| | SGCN | $53.5_{\pm4.2}/56.0_{\pm7.7}$ | $43.3_{\pm4.2}$ | $64.6_{\pm4.0}/59.8_{\pm8.2}$ | $38.2_{\pm3.1}$ | $49.5_{\pm5.2}/48.7_{\pm5.2}$ | $25.2_{\pm4.8}$ |
| | BGCN | $54.5_{\pm5.0}/52.9_{\pm5.6}$ | $40.5_{\pm4.0}$ | $62.9_{\pm4.3}/62.5_{\pm4.9}$ | $38.3_{\pm3.3}$ | $49.7_{\pm5.2}/50.9_{\pm5.6}$ | $26.9_{\pm4.8}$ |
| | GPRGNN | $52.4_{\pm6.6}/53.4_{\pm7.2}$ | $36.8_{\pm4.6}$ | $62.5_{\pm4.0}/46.9_{\pm6.2}$ | $37.1_{\pm3.6}$ | $54.1_{\pm1.8}/50.5_{\pm7.5}$ | $30.5_{\pm4.1}$ |
| | FSGNN | $55.3_{\pm7.4}/53.9_{\pm7.7}$ | $43.9_{\pm4.2}$ | $68.4_{\pm4.1}/58.0_{\pm4.2}$ | $36.9_{\pm3.6}$ | $53.8_{\pm4.8}/50.0_{\pm6.0}$ | $30.5_{\pm3.7}$ |
| | FAGCN | $57.9_{\pm5.4}/57.5_{\pm6.0}$ | $43.2_{\pm6.4}$ | $65.3_{\pm4.6}/63.2_{\pm5.3}$ | $39.7_{\pm3.6}$ | $55.6_{\pm6.6}/52.3_{\pm5.9}$ | $30.2_{\pm4.1}$ |
| | Ens. | $59.2_{\pm4.9}/54.4_{\pm5.5}$ | | $64.0_{\pm4.2}/64.1_{\pm4.7}$ | | $62.4_{\pm2.9}/65.1_{\pm3.0}$ | |
| | MCD | $58.4_{\pm6.6}/54.2_{\pm5.7}$ | | $64.6_{\pm4.4}/62.5_{\pm4.1}$ | | $64.1_{\pm4.4}/62.8_{\pm4.3}$ | |
| | EBM | $56.9_{\pm5.8}/56.4_{\pm6.3}$ | $47.8_{\pm4.9}$ | $62.8_{\pm5.1}/64.4_{\pm4.3}$ | $40.7_{\pm3.9}$ | $62.3_{\pm4.9}/63.8_{\pm4.4}$ | $40.1_{\pm3.3}$ |
| | Safe | $56.9_{\pm5.8}/52.6_{\pm6.0}$ | | $62.8_{\pm5.1}/60.4_{\pm6.3}$ | | $62.3_{\pm4.9}/60.8_{\pm5.5}$ | |
| | GEBM | $56.9_{\pm5.8}/49.3_{\pm5.6}$ | | $62.8_{\pm5.1}/48.8_{\pm7.4}$ | | $62.3_{\pm4.9}/51.3_{\pm8.6}$ | |
| | JLDE | $56.9_{\pm5.8}/49.9_{\pm8.0}$ | | $62.8_{\pm5.1}/58.9_{\pm6.7}$ | | $62.3_{\pm4.9}/60.2_{\pm5.9}$ | |
| **Squirrel** | GPN | $52.2_{\pm5.5}/39.9_{\pm5.3}$ | $41.1_{\pm2.9}$ | $47.1_{\pm16.3}/28.6_{\pm3.1}$ | $34.5_{\pm1.6}$ | $47.6_{\pm16.0}/28.7_{\pm2.8}$ | $34.6_{\pm1.6}$ |
| | SGCN | $58.3_{\pm3.9}/49.4_{\pm3.8}$ | $47.1_{\pm2.9}$ | $62.4_{\pm7.3}/42.0_{\pm5.8}$ | $37.7_{\pm1.7}$ | $44.5_{\pm4.1}/39.0_{\pm4.5}$ | $28.2_{\pm3.8}$ |
| | BGCN | $58.6_{\pm5.4}/56.2_{\pm7.4}$ | $43.2_{\pm4.4}$ | $58.7_{\pm7.7}/65.2_{\pm6.3}$ | $33.9_{\pm3.1}$ | $48.8_{\pm4.8}/61.4_{\pm4.2}$ | $30.2_{\pm3.0}$ |
| | GPRGNN | $50.3_{\pm10.3}/49.7_{\pm9.9}$ | $40.9_{\pm3.6}$ | $53.4_{\pm17.9}/54.3_{\pm16.0}$ | $35.4_{\pm1.4}$ | $52.2_{\pm15.8}/46.4_{\pm11.7}$ | $34.4_{\pm1.9}$ |
| | FSGNN | $59.5_{\pm6.6}/59.3_{\pm6.6}$ | $45.1_{\pm3.2}$ | $71.3_{\pm5.1}/72.4_{\pm5.3}$ | $36.1_{\pm1.7}$ | $53.9_{\pm2.9}/52.7_{\pm2.9}$ | $33.9_{\pm2.4}$ |
| | FAGCN | $58.9_{\pm7.2}/57.5_{\pm7.2}$ | $47.0_{\pm6.5}$ | $67.9_{\pm6.5}/63.9_{\pm5.5}$ | $40.5_{\pm2.8}$ | $53.4_{\pm4.9}/50.0_{\pm4.8}$ | $35.3_{\pm2.7}$ |
| | Ens. | $60.5_{\pm2.1}/53.7_{\pm5.8}$ | | $69.6_{\pm2.4}/61.0_{\pm4.5}$ | | $67.4_{\pm2.4}/62.4_{\pm2.8}$ | |
| | MCD | $60.6_{\pm4.2}/57.3_{\pm4.7}$ | | $68.8_{\pm2.5}/61.7_{\pm5.1}$ | | $67.8_{\pm2.5}/62.5_{\pm4.1}$ | |
| | EBM | $58.9_{\pm3.8}/59.4_{\pm4.1}$ | $49.0_{\pm3.0}$ | $68.4_{\pm2.9}/68.9_{\pm2.6}$ | $41.8_{\pm2.2}$ | $66.6_{\pm2.7}/66.6_{\pm2.3}$ | $40.5_{\pm1.9}$ |
| | Safe | $58.9_{\pm3.8}/55.4_{\pm4.5}$ | | $68.4_{\pm2.9}/64.0_{\pm3.1}$ | | $66.6_{\pm2.7}/62.7_{\pm2.8}$ | |
| | GEBM | $58.9_{\pm3.8}/43.4_{\pm3.8}$ | | $68.4_{\pm2.9}/41.1_{\pm3.1}$ | | $66.6_{\pm2.7}/43.2_{\pm2.9}$ | |
| | JLDE | $58.9_{\pm3.8}/47.8_{\pm6.9}$ | | $68.4_{\pm2.9}/50.3_{\pm4.4}$ | | $66.6_{\pm2.7}/52.9_{\pm3.8}$ | |
| **CoraML** | GPN | $87.3_{\pm2.2}/71.3_{\pm7.8}$ | $88.9_{\pm3.9}$ | $82.8_{\pm1.3}/68.6_{\pm3.5}$ | $80.9_{\pm2.0}$ | $82.7_{\pm1.3}/68.3_{\pm3.4}$ | $81.1_{\pm1.9}$ |
| | SGCN | $83.2_{\pm2.3}/78.4_{\pm3.8}$ | $89.4_{\pm1.5}$ | $79.2_{\pm1.5}/72.6_{\pm1.6}$ | $82.0_{\pm1.5}$ | $45.8_{\pm3.1}/44.6_{\pm3.5}$ | $34.4_{\pm2.7}$ |
| | BGCN | $84.7_{\pm2.6}/77.9_{\pm8.0}$ | $88.3_{\pm2.0}$ | $81.2_{\pm1.8}/78.1_{\pm2.7}$ | $81.3_{\pm1.6}$ | $45.8_{\pm4.4}/50.1_{\pm3.9}$ | $34.1_{\pm2.9}$ |
| | GPRGNN | $85.2_{\pm3.2}/82.5_{\pm3.2}$ | $90.2_{\pm1.8}$ | $82.4_{\pm1.6}/76.8_{\pm1.4}$ | $84.1_{\pm1.2}$ | $49.6_{\pm3.4}/47.6_{\pm3.1}$ | $37.3_{\pm4.2}$ |
| | FSGNN | $84.6_{\pm2.0}/78.1_{\pm3.7}$ | $87.5_{\pm1.5}$ | $79.3_{\pm1.6}/79.3_{\pm1.6}$ | $71.7_{\pm1.3}$ | $44.7_{\pm1.5}/39.5_{\pm1.6}$ | $40.2_{\pm3.0}$ |
| | FAGCN | $82.9_{\pm2.4}/79.7_{\pm3.1}$ | $88.9_{\pm1.6}$ | $81.1_{\pm1.3}/76.4_{\pm1.7}$ | $81.4_{\pm2.3}$ | $45.3_{\pm2.7}/37.5_{\pm2.3}$ | $40.4_{\pm3.4}$ |
| | Ens. | $85.5_{\pm1.8}/84.1_{\pm2.3}$ | | $80.3_{\pm1.6}/79.9_{\pm1.6}$ | | $80.0_{\pm1.4}/79.5_{\pm1.5}$ | |
| | MCD | $83.4_{\pm2.7}/80.3_{\pm4.6}$ | | $80.0_{\pm2.0}/79.7_{\pm1.9}$ | | $79.9_{\pm1.5}/78.4_{\pm2.7}$ | |
| | EBM | $82.8_{\pm3.5}/79.4_{\pm4.7}$ | $88.3_{\pm1.4}$ | $78.8_{\pm2.0}/77.8_{\pm2.4}$ | $79.6_{\pm1.5}$ | $79.2_{\pm1.5}/77.4_{\pm1.7}$ | $77.1_{\pm1.8}$ |
| | Safe | $82.8_{\pm3.5}/79.1_{\pm5.3}$ | | $78.8_{\pm2.0}/78.2_{\pm2.6}$ | | $79.2_{\pm1.5}/78.3_{\pm1.9}$ | |
| | GEBM | $82.8_{\pm3.5}/81.7_{\pm4.7}$ | | $78.8_{\pm2.0}/77.0_{\pm2.2}$ | | $79.2_{\pm1.5}/77.3_{\pm1.6}$ | |
| | JLDE | $82.8_{\pm3.5}/82.2_{\pm3.4}$ | | $78.8_{\pm2.0}/79.7_{\pm1.4}$ | | $79.2_{\pm1.5}/78.5_{\pm1.1}$ | |
| **PubMed** | GPN | $90.2_{\pm4.7}/70.4_{\pm11.3}$ | $91.2_{\pm8.1}$ | $80.1_{\pm0.7}/65.1_{\pm1.8}$ | $84.5_{\pm0.5}$ | $80.3_{\pm0.7}/65.0_{\pm1.9}$ | $84.5_{\pm0.5}$ |
| | SGCN | $90.8_{\pm0.8}/90.6_{\pm0.9}$ | $94.6_{\pm0.2}$ | $80.2_{\pm0.7}/71.6_{\pm2.0}$ | $85.7_{\pm0.3}$ | $53.0_{\pm1.4}/53.3_{\pm1.6}$ | $50.2_{\pm1.6}$ |
| | BGCN | $91.7_{\pm1.1}/90.8_{\pm2.6}$ | $94.3_{\pm0.5}$ | $80.5_{\pm0.7}/77.0_{\pm2.0}$ | $85.7_{\pm0.3}$ | $49.6_{\pm1.1}/51.6_{\pm1.3}$ | $53.4_{\pm1.4}$ |
| | GPRGNN | $90.2_{\pm0.9}/90.2_{\pm0.9}$ | $95.2_{\pm0.3}$ | $80.3_{\pm0.5}/75.7_{\pm1.0}$ | $86.0_{\pm0.3}$ | $51.6_{\pm1.9}/50.7_{\pm1.9}$ | $51.5_{\pm4.5}$ |
| | FSGNN | $92.3_{\pm0.8}/92.3_{\pm0.8}$ | $95.4_{\pm0.2}$ | $80.2_{\pm0.5}/72.0_{\pm1.7}$ | $82.8_{\pm0.5}$ | $49.2_{\pm2.2}/43.5_{\pm2.2}$ | $69.9_{\pm1.7}$ |
| | FAGCN | $91.1_{\pm0.7}/91.1_{\pm0.8}$ | $94.7_{\pm0.2}$ | $80.4_{\pm0.6}/73.4_{\pm0.9}$ | $85.8_{\pm0.4}$ | $47.1_{\pm1.1}/43.8_{\pm1.3}$ | $54.2_{\pm1.8}$ |
| | Ens. | $89.8_{\pm1.5}/89.7_{\pm1.3}$ | | $78.1_{\pm1.1}/78.6_{\pm1.1}$ | | $79.3_{\pm1.1}/79.5_{\pm0.9}$ | |
| | MCD | $87.7_{\pm3.3}/87.2_{\pm3.0}$ | | $79.1_{\pm0.8}/78.7_{\pm0.7}$ | | $79.9_{\pm0.8}/78.9_{\pm0.6}$ | |
| | EBM | $86.4_{\pm4.5}/83.8_{\pm7.4}$ | $94.9_{\pm0.3}$ | $76.0_{\pm1.4}/73.2_{\pm2.7}$ | $83.9_{\pm0.5}$ | $77.6_{\pm1.3}/75.1_{\pm2.0}$ | $82.9_{\pm0.4}$ |
| | Safe | $86.4_{\pm4.5}/82.1_{\pm6.6}$ | | $76.0_{\pm1.4}/69.1_{\pm3.9}$ | | $77.6_{\pm1.3}/70.3_{\pm3.8}$ | |
| | GEBM | $86.4_{\pm4.5}/85.5_{\pm4.8}$ | | $76.0_{\pm1.4}/70.4_{\pm3.4}$ | | $77.6_{\pm1.3}/70.2_{\pm3.7}$ | |
| | JLDE | $86.4_{\pm4.5}/84.5_{\pm1.3}$ | | $76.0_{\pm1.4}/68.2_{\pm0.9}$ | | $77.6_{\pm1.3}/70.6_{\pm0.9}$ | |

*Table 14.* Misclassification detection AUC-PR using different uncertainty estimators and our method using a Res-GCN backbone.

| | Model | LoC AUC-PR (Alea. / Epi.) | Acc.↑ | Near-Features AUC-PR (Alea. / Epi.) | Acc.↑ | Far-Features AUC-PR (Alea. / Epi.) | Acc.↑ |
|---|---|---|---|---|---|---|---|
| Roman Empire | GPN | $70.1_{\pm 3.4}$/$51.7_{\pm 5.1}$ | $48.6_{\pm 1.9}$ | $56.4_{\pm 4.9}$/$41.1_{\pm 3.4}$ | $38.1_{\pm 1.3}$ | $55.5_{\pm 4.3}$/$42.1_{\pm 3.4}$ | $38.0_{\pm 1.3}$ |
| | SGCN | $72.3_{\pm 2.7}$/$65.2_{\pm 1.7}$ | $58.3_{\pm 1.6}$ | $60.9_{\pm 3.2}$/$53.2_{\pm 1.7}$ | $45.5_{\pm 1.6}$ | $23.3_{\pm 0.6}$/$22.2_{\pm 0.7}$ | $30.6_{\pm 1.0}$ |
| | BGCN | $66.7_{\pm 1.2}$/$64.0_{\pm 1.5}$ | $53.0_{\pm 0.7}$ | $56.9_{\pm 1.3}$/$52.4_{\pm 2.0}$ | $42.4_{\pm 0.5}$ | $27.6_{\pm 1.4}$/$38.5_{\pm 2.2}$ | $28.8_{\pm 0.6}$ |
| | GPRGNN | $95.4_{\pm 0.3}$/$93.7_{\pm 0.2}$ | $78.1_{\pm 0.4}$ | $91.8_{\pm 0.3}$/$89.3_{\pm 0.4}$ | $64.8_{\pm 0.6}$ | $69.8_{\pm 0.6}$/$81.2_{\pm 1.4}$ | $61.0_{\pm 0.6}$ |
| | FSGNN | $98.0_{\pm 0.1}$/$95.3_{\pm 0.4}$ | $86.6_{\pm 0.4}$ | $92.8_{\pm 0.4}$/$89.4_{\pm 0.6}$ | $71.6_{\pm 0.5}$ | $56.0_{\pm 1.4}$/$44.9_{\pm 1.2}$ | $56.9_{\pm 0.9}$ |
| | FAGCN | $97.1_{\pm 0.5}$/$94.6_{\pm 1.0}$ | $83.6_{\pm 1.1}$ | $92.4_{\pm 0.6}$/$88.4_{\pm 0.9}$ | $68.0_{\pm 0.9}$ | $39.8_{\pm 4.7}$/$31.6_{\pm 3.5}$ | $44.3_{\pm 2.8}$ |
| | Ens. | **98.4**$_{\pm 0.1}$/**98.4**$_{\pm 0.1}$ | | **94.7**$_{\pm 0.3}$/**94.4**$_{\pm 0.3}$ | | **95.3**$_{\pm 0.2}$/**95.0**$_{\pm 0.2}$ | |
| | MCD | **98.3**$_{\pm 0.1}$/$98.2_{\pm 0.2}$ | | $93.8_{\pm 0.4}$/$93.3_{\pm 0.4}$ | | $94.4_{\pm 0.3}$/$93.4_{\pm 0.4}$ | |
| | EBM | $97.7_{\pm 0.2}$/$95.9_{\pm 0.8}$ | $87.6_{\pm 0.7}$ | $93.0_{\pm 0.4}$/$91.3_{\pm 0.9}$ | $72.3_{\pm 0.6}$ | $93.5_{\pm 0.4}$/$92.1_{\pm 0.8}$ | $70.6_{\pm 0.6}$ |
| | Safe | $97.7_{\pm 0.2}$/$94.2_{\pm 0.6}$ | | $93.0_{\pm 0.4}$/$87.9_{\pm 0.7}$ | | $93.5_{\pm 0.4}$/$88.1_{\pm 0.6}$ | |
| | GEBM | $97.7_{\pm 0.2}$/$86.1_{\pm 1.3}$ | | $93.0_{\pm 0.4}$/$74.1_{\pm 1.4}$ | | $93.5_{\pm 0.4}$/$75.3_{\pm 1.6}$ | |
| | JLDE | $97.7_{\pm 0.2}$/$97.3_{\pm 0.3}$ | | $93.0_{\pm 0.4}$/$93.0_{\pm 0.4}$ | | $93.5_{\pm 0.4}$/$93.4_{\pm 0.4}$ | |
| Amazon Ratings | GPN | $63.6_{\pm 1.6}$/$60.9_{\pm 2.3}$ | $54.2_{\pm 1.2}$ | $56.2_{\pm 1.1}$/$52.7_{\pm 1.2}$ | $46.8_{\pm 0.7}$ | $56.4_{\pm 1.1}$/$52.9_{\pm 1.2}$ | $47.0_{\pm 0.7}$ |
| | SGCN | $64.1_{\pm 1.2}$/$63.1_{\pm 1.1}$ | $55.5_{\pm 0.7}$ | $56.1_{\pm 1.4}$/$52.7_{\pm 1.1}$ | $47.2_{\pm 0.5}$ | $32.6_{\pm 2.2}$/$31.6_{\pm 2.1}$ | $33.5_{\pm 1.3}$ |
| | BGCN | $60.7_{\pm 1.5}$/$55.6_{\pm 1.8}$ | $51.4_{\pm 0.8}$ | $54.9_{\pm 1.9}$/$48.7_{\pm 2.4}$ | $44.2_{\pm 1.3}$ | $35.4_{\pm 2.1}$/$37.6_{\pm 2.0}$ | $33.5_{\pm 1.3}$ |
| | GPRGNN | $69.0_{\pm 1.1}$/$67.9_{\pm 1.2}$ | $57.9_{\pm 0.8}$ | $63.6_{\pm 0.9}$/$63.1_{\pm 1.2}$ | $50.6_{\pm 0.7}$ | $33.0_{\pm 3.1}$/$39.5_{\pm 3.6}$ | $34.1_{\pm 2.7}$ |
| | FSGNN | $66.7_{\pm 0.6}$/$66.4_{\pm 0.7}$ | $56.8_{\pm 0.6}$ | $60.8_{\pm 0.7}$/$58.1_{\pm 0.7}$ | $49.1_{\pm 0.6}$ | $36.6_{\pm 0.9}$/$33.9_{\pm 0.9}$ | $38.6_{\pm 0.8}$ |
| | FAGCN | $69.3_{\pm 1.0}$/$68.2_{\pm 1.1}$ | **58.1**$_{\pm 0.8}$ | $63.6_{\pm 0.9}$/$60.9_{\pm 1.1}$ | **51.2**$_{\pm 0.7}$ | $30.0_{\pm 2.2}$/$28.2_{\pm 3.1}$ | $31.7_{\pm 2.4}$ |
| | Ens. | **72.5**$_{\pm 0.6}$/**71.0**$_{\pm 0.7}$ | | **66.8**$_{\pm 0.8}$/**65.8**$_{\pm 0.8}$ | | **66.0**$_{\pm 0.8}$/**65.0**$_{\pm 0.8}$ | |
| | MCD | $69.8_{\pm 0.9}$/$66.3_{\pm 1.1}$ | | $63.0_{\pm 0.8}$/$61.6_{\pm 1.1}$ | | $62.5_{\pm 0.8}$/$60.3_{\pm 1.3}$ | |
| | EBM | $69.5_{\pm 0.8}$/$68.3_{\pm 1.4}$ | $56.0_{\pm 0.6}$ | $63.0_{\pm 0.9}$/$64.0_{\pm 1.0}$ | $48.2_{\pm 0.5}$ | $62.3_{\pm 0.9}$/$63.1_{\pm 1.0}$ | $47.3_{\pm 0.6}$ |
| | Safe | $69.5_{\pm 0.8}$/$64.9_{\pm 1.4}$ | | $63.0_{\pm 0.9}$/$59.8_{\pm 1.2}$ | | $62.3_{\pm 0.9}$/$59.0_{\pm 1.0}$ | |
| | GEBM | $69.5_{\pm 0.9}$/$56.6_{\pm 1.6}$ | | $63.0_{\pm 0.9}$/$50.4_{\pm 1.4}$ | | $62.3_{\pm 0.9}$/$49.9_{\pm 1.4}$ | |
| | JLDE | $69.5_{\pm 0.9}$/$65.1_{\pm 1.1}$ | | $63.0_{\pm 0.9}$/$60.0_{\pm 2.6}$ | | $62.3_{\pm 0.9}$/$59.3_{\pm 2.5}$ | |
| Chameleon | GPN | $45.6_{\pm 7.3}$/$37.3_{\pm 4.5}$ | $39.6_{\pm 3.8}$ | $52.5_{\pm 5.6}$/$33.1_{\pm 4.3}$ | $37.0_{\pm 3.8}$ | $53.0_{\pm 6.3}$/$33.2_{\pm 4.2}$ | $37.0_{\pm 3.8}$ |
| | SGCN | $48.0_{\pm 4.8}$/$50.6_{\pm 8.7}$ | $43.3_{\pm 4.2}$ | $52.1_{\pm 4.0}$/$46.5_{\pm 7.4}$ | $38.2_{\pm 3.1}$ | $26.4_{\pm 4.5}$/$26.4_{\pm 4.7}$ | $25.2_{\pm 4.8}$ |
| | BGCN | $45.4_{\pm 5.5}$/$44.9_{\pm 5.7}$ | $40.5_{\pm 4.0}$ | $48.7_{\pm 3.9}$/$51.4_{\pm 5.7}$ | $38.3_{\pm 3.3}$ | $27.8_{\pm 6.5}$/$28.5_{\pm 6.7}$ | $26.9_{\pm 4.8}$ |
| | GPRGNN | $40.6_{\pm 5.3}$/$41.7_{\pm 6.0}$ | $36.8_{\pm 4.6}$ | $46.0_{\pm 4.5}$/$37.2_{\pm 5.5}$ | $37.1_{\pm 3.6}$ | $33.9_{\pm 7.9}$/$31.8_{\pm 6.4}$ | $30.5_{\pm 4.1}$ |
| | FSGNN | $50.9_{\pm 7.8}$/$50.1_{\pm 9.3}$ | $43.9_{\pm 4.2}$ | $60.3_{\pm 4.3}$/$46.2_{\pm 6.1}$ | $36.9_{\pm 3.6}$ | $33.2_{\pm 4.4}$/$31.5_{\pm 4.5}$ | $30.5_{\pm 3.7}$ |
| | FAGCN | $53.9_{\pm 9.1}$/$54.2_{\pm 9.9}$ | $43.2_{\pm 6.4}$ | $57.8_{\pm 5.5}$/$55.1_{\pm 6.4}$ | $39.7_{\pm 3.6}$ | $33.7_{\pm 6.4}$/$32.3_{\pm 5.9}$ | $30.2_{\pm 4.1}$ |
| | Ens. | **59.1**$_{\pm 6.1}$/$55.6_{\pm 6.4}$ | | $62.6_{\pm 3.0}$/**62.8**$_{\pm 4.0}$ | | **61.0**$_{\pm 2.9}$/$63.1_{\pm 3.5}$ | |
| | MCD | **55.8**$_{\pm 7.3}$/$52.0_{\pm 6.6}$ | | $61.6_{\pm 5.1}$/$59.6_{\pm 4.9}$ | | $60.4_{\pm 5.0}$/$59.0_{\pm 4.9}$ | |
| | EBM | $55.5_{\pm 6.9}$/$55.6_{\pm 6.9}$ | $47.8_{\pm 4.9}$ | $57.9_{\pm 6.6}$/$60.4_{\pm 5.7}$ | $40.7_{\pm 3.9}$ | $56.6_{\pm 6.4}$/$59.6_{\pm 5.4}$ | $40.1_{\pm 3.3}$ |
| | Safe | $55.5_{\pm 6.9}$/$53.6_{\pm 6.3}$ | | $57.9_{\pm 6.6}$/$52.5_{\pm 7.4}$ | | $56.6_{\pm 6.4}$/$51.9_{\pm 6.3}$ | |
| | GEBM | $55.5_{\pm 6.9}$/$52.0_{\pm 7.9}$ | | $57.9_{\pm 6.6}$/$43.2_{\pm 6.6}$ | | $56.6_{\pm 6.4}$/$44.7_{\pm 7.5}$ | |
| | JLDE | $55.5_{\pm 6.9}$/$51.4_{\pm 8.9}$ | | $57.9_{\pm 6.6}$/$50.7_{\pm 9.1}$ | | $56.6_{\pm 6.4}$/$51.8_{\pm 7.6}$ | |
| Squirrel | GPN | $45.2_{\pm 5.6}$/$35.3_{\pm 3.6}$ | $41.1_{\pm 2.9}$ | $37.8_{\pm 11.4}$/$24.8_{\pm 1.2}$ | $34.5_{\pm 1.6}$ | $38.1_{\pm 11.4}$/$24.9_{\pm 1.3}$ | $34.6_{\pm 1.6}$ |
| | SGCN | $55.1_{\pm 4.3}$/$48.5_{\pm 5.1}$ | $47.1_{\pm 2.9}$ | $47.1_{\pm 5.4}$/$33.5_{\pm 4.0}$ | $37.7_{\pm 1.7}$ | $25.7_{\pm 3.5}$/$24.0_{\pm 2.4}$ | $28.2_{\pm 3.8}$ |
| | BGCN | $51.9_{\pm 6.3}$/$52.7_{\pm 7.8}$ | $43.2_{\pm 4.4}$ | $42.8_{\pm 6.6}$/$52.2_{\pm 8.2}$ | $33.9_{\pm 3.1}$ | $30.1_{\pm 3.1}$/$39.5_{\pm 6.5}$ | $30.2_{\pm 3.0}$ |
| | GPRGNN | $44.3_{\pm 9.9}$/$43.5_{\pm 8.9}$ | $40.9_{\pm 3.6}$ | $45.1_{\pm 14.6}$/$43.3_{\pm 12.9}$ | $35.4_{\pm 1.4}$ | $38.2_{\pm 9.4}$/$34.7_{\pm 8.2}$ | $34.4_{\pm 1.9}$ |
| | FSGNN | $56.4_{\pm 6.6}$/$55.8_{\pm 7.0}$ | $45.1_{\pm 3.2}$ | $59.1_{\pm 6.8}$/$60.8_{\pm 7.6}$ | $36.1_{\pm 1.7}$ | $35.1_{\pm 2.7}$/$34.8_{\pm 2.9}$ | $33.9_{\pm 2.4}$ |
| | FAGCN | $58.1_{\pm 10.9}$/$57.4_{\pm 11.4}$ | $47.0_{\pm 6.5}$ | $60.0_{\pm 7.7}$/$54.7_{\pm 6.6}$ | $40.5_{\pm 2.8}$ | $36.4_{\pm 4.3}$/$35.0_{\pm 4.5}$ | $35.3_{\pm 2.7}$ |
| | Ens. | **61.3**$_{\pm 2.6}$/$56.0_{\pm 5.5}$ | | **65.5**$_{\pm 2.3}$/$57.5_{\pm 5.6}$ | | **61.5**$_{\pm 2.3}$/$56.0_{\pm 3.5}$ | |
| | MCD | **60.0**$_{\pm 4.8}$/$57.3_{\pm 4.9}$ | | $63.1_{\pm 3.4}$/$57.0_{\pm 4.5}$ | | **59.9**$_{\pm 2.7}$/$55.5_{\pm 3.8}$ | |
| | EBM | $59.5_{\pm 4.4}$/**60.0**$_{\pm 3.9}$ | $49.0_{\pm 3.0}$ | $63.6_{\pm 3.6}$/**64.0**$_{\pm 3.7}$ | $41.8_{\pm 2.2}$ | $59.6_{\pm 3.0}$/**59.9**$_{\pm 3.1}$ | $40.5_{\pm 1.9}$ |
| | Safe | $59.5_{\pm 4.4}$/$56.0_{\pm 5.5}$ | | $63.6_{\pm 3.6}$/$57.7_{\pm 4.2}$ | | $59.6_{\pm 3.0}$/$54.4_{\pm 3.6}$ | |
| | GEBM | $59.5_{\pm 4.4}$/$46.1_{\pm 4.5}$ | | $63.6_{\pm 3.6}$/$36.4_{\pm 3.1}$ | | $59.6_{\pm 3.0}$/$36.2_{\pm 2.9}$ | |
| | JLDE | $59.5_{\pm 4.4}$/$49.4_{\pm 5.9}$ | | $63.6_{\pm 3.6}$/$44.2_{\pm 4.4}$ | | $59.6_{\pm 3.0}$/$44.2_{\pm 3.6}$ | |
| CoraML | GPN | **98.0**$_{\pm 0.7}$/$94.9_{\pm 1.9}$ | $88.9_{\pm 3.9}$ | **94.7**$_{\pm 0.9}$/$89.7_{\pm 1.6}$ | $80.9_{\pm 2.0}$ | **94.8**$_{\pm 0.8}$/$89.8_{\pm 1.5}$ | **81.1**$_{\pm 1.9}$ |
| | SGCN | $97.0_{\pm 1.0}$/$96.1_{\pm 1.3}$ | $89.4_{\pm 1.5}$ | $93.7_{\pm 0.9}$/$91.2_{\pm 1.0}$ | $82.0_{\pm 1.5}$ | $30.4_{\pm 3.3}$/$29.8_{\pm 3.3}$ | $34.4_{\pm 2.7}$ |
| | BGCN | $97.1_{\pm 1.0}$/$96.1_{\pm 1.9}$ | $88.3_{\pm 2.0}$ | $94.3_{\pm 1.0}$/$93.7_{\pm 1.2}$ | $81.3_{\pm 1.6}$ | $30.6_{\pm 3.7}$/$32.1_{\pm 3.8}$ | $34.1_{\pm 2.9}$ |
| | GPRGNN | $97.6_{\pm 1.2}$/$97.3_{\pm 1.1}$ | **90.2**$_{\pm 1.8}$ | **95.6**$_{\pm 0.7}$/$94.2_{\pm 0.8}$ | **84.1**$_{\pm 1.2}$ | $35.8_{\pm 5.7}$/$34.5_{\pm 4.9}$ | $37.3_{\pm 4.2}$ |
| | FSGNN | $97.0_{\pm 0.8}$/$95.6_{\pm 1.3}$ | $87.5_{\pm 1.5}$ | $90.2_{\pm 1.2}$/$89.6_{\pm 1.2}$ | $71.7_{\pm 1.3}$ | $35.0_{\pm 2.7}$/$33.1_{\pm 2.4}$ | $40.2_{\pm 3.0}$ |
| | FAGCN | $96.9_{\pm 1.0}$/$96.3_{\pm 1.2}$ | $88.9_{\pm 1.6}$ | $94.3_{\pm 1.3}$/$92.6_{\pm 1.6}$ | $81.4_{\pm 2.3}$ | $35.9_{\pm 4.4}$/$32.3_{\pm 3.5}$ | $40.4_{\pm 3.4}$ |
| | Ens. | **97.7**$_{\pm 0.5}$/$97.4_{\pm 0.6}$ | | $93.6_{\pm 0.8}$/$93.4_{\pm 0.8}$ | | **92.9**$_{\pm 0.8}$/$92.7_{\pm 0.8}$ | |
| | MCD | $96.9_{\pm 1.0}$/$96.3_{\pm 1.3}$ | | $92.9_{\pm 1.1}$/$92.9_{\pm 1.1}$ | | $92.3_{\pm 1.1}$/$91.9_{\pm 1.7}$ | |
| | EBM | $96.8_{\pm 1.5}$/$96.0_{\pm 1.5}$ | $88.3_{\pm 1.4}$ | $92.6_{\pm 1.1}$/$92.2_{\pm 1.2}$ | $79.6_{\pm 1.5}$ | $91.9_{\pm 1.1}$/$91.2_{\pm 1.2}$ | $77.1_{\pm 1.8}$ |
| | Safe | $96.8_{\pm 1.5}$/$96.3_{\pm 1.5}$ | | $92.6_{\pm 1.1}$/$92.8_{\pm 1.2}$ | | $91.9_{\pm 1.1}$/$91.8_{\pm 1.2}$ | |
| | GEBM | $96.8_{\pm 1.5}$/$97.0_{\pm 1.2}$ | | $92.6_{\pm 1.1}$/$92.5_{\pm 1.2}$ | | $91.9_{\pm 1.1}$/$91.8_{\pm 1.3}$ | |
| | JLDE | $96.8_{\pm 1.5}$/$96.9_{\pm 1.1}$ | | $92.6_{\pm 1.1}$/$93.4_{\pm 0.8}$ | | $91.9_{\pm 1.1}$/$92.2_{\pm 0.9}$ | |
| PubMed | GPN | $98.7_{\pm 3.4}$/$95.9_{\pm 4.5}$ | $91.2_{\pm 8.1}$ | $95.2_{\pm 0.4}$/$90.1_{\pm 0.7}$ | $84.5_{\pm 0.5}$ | **95.3**$_{\pm 0.4}$/$90.1_{\pm 0.7}$ | **84.5**$_{\pm 0.5}$ |
| | SGCN | **99.4**$_{\pm 0.1}$/**99.3**$_{\pm 0.1}$ | $94.6_{\pm 0.2}$ | **95.4**$_{\pm 0.3}$/$93.0_{\pm 0.7}$ | $85.7_{\pm 0.3}$ | $49.6_{\pm 1.5}$/$49.6_{\pm 1.4}$ | $50.2_{\pm 1.6}$ |
| | BGCN | **99.4**$_{\pm 0.1}$/**99.4**$_{\pm 0.2}$ | $94.3_{\pm 0.5}$ | **95.6**$_{\pm 0.3}$/$95.1_{\pm 0.6}$ | $85.7_{\pm 0.3}$ | $51.2_{\pm 1.5}$/$52.1_{\pm 1.9}$ | $53.4_{\pm 1.4}$ |
| | GPRGNN | **99.4**$_{\pm 0.1}$/**99.4**$_{\pm 0.1}$ | $95.2_{\pm 0.3}$ | **95.6**$_{\pm 0.2}$/$94.4_{\pm 0.4}$ | **86.0**$_{\pm 0.3}$ | $51.2_{\pm 3.0}$/$49.7_{\pm 3.4}$ | $51.5_{\pm 4.5}$ |
| | FSGNN | **99.6**$_{\pm 0.1}$/**99.6**$_{\pm 0.1}$ | **95.4**$_{\pm 0.2}$ | $94.5_{\pm 0.2}$/$91.6_{\pm 0.7}$ | $82.8_{\pm 0.5}$ | $64.8_{\pm 2.1}$/$62.5_{\pm 2.1}$ | $69.9_{\pm 1.7}$ |
| | FAGCN | **99.4**$_{\pm 0.1}$/**99.4**$_{\pm 0.1}$ | $94.7_{\pm 0.2}$ | **95.6**$_{\pm 0.3}$/$93.6_{\pm 0.3}$ | $85.8_{\pm 0.4}$ | $50.8_{\pm 2.2}$/$48.7_{\pm 2.1}$ | $54.2_{\pm 1.8}$ |
| | Ens. | **99.4**$_{\pm 0.1}$/$99.3_{\pm 0.1}$ | | $94.3_{\pm 0.5}$/$94.5_{\pm 0.5}$ | | $94.3_{\pm 0.5}$/$94.4_{\pm 0.5}$ | |
| | MCD | $99.0_{\pm 0.5}$/$99.0_{\pm 0.4}$ | | $94.5_{\pm 0.4}$/$94.4_{\pm 0.4}$ | | **94.5**$_{\pm 0.4}$/$94.3_{\pm 0.3}$ | |
| | EBM | $99.0_{\pm 0.5}$/$98.6_{\pm 1.0}$ | $94.9_{\pm 0.3}$ | $93.4_{\pm 0.7}$/$92.3_{\pm 0.9}$ | $83.9_{\pm 0.5}$ | $93.5_{\pm 0.7}$/$92.4_{\pm 0.8}$ | $82.9_{\pm 0.4}$ |
| | Safe | $99.0_{\pm 0.5}$/$98.7_{\pm 0.7}$ | | $93.4_{\pm 0.7}$/$91.6_{\pm 1.3}$ | | $93.5_{\pm 0.7}$/$91.4_{\pm 1.2}$ | |
| | GEBM | $99.0_{\pm 0.5}$/$99.0_{\pm 0.5}$ | | $93.4_{\pm 0.7}$/$92.2_{\pm 1.0}$ | | $93.5_{\pm 0.7}$/$91.7_{\pm 1.1}$ | |
| | JLDE | $99.0_{\pm 0.5}$/$99.0_{\pm 0.1}$ | | $93.4_{\pm 0.7}$/$91.8_{\pm 0.4}$ | | $93.5_{\pm 0.7}$/$91.8_{\pm 0.4}$ | |

*Table 15.* Expected Calibration Error (ECE) (↓) of different models on clean data (best and runner-up).

| Model | Roman Empire | Amazon Ratings | Chameleon | Squirrel | CoraML | PubMed |
|---|---|---|---|---|---|---|
| GCN | $7.0_{\pm1.6}$ | $19.0_{\pm2.2}$ | $29.3_{\pm12.7}$ | $\mathbf{6.0}_{\pm2.4}$ | $14.8_{\pm1.7}$ | $5.2_{\pm1.8}$ |
| GATv2 | $7.8_{\pm0.6}$ | $36.5_{\pm4.0}$ | $44.1_{\pm10.1}$ | $29.5_{\pm14.6}$ | $15.4_{\pm2.0}$ | $7.2_{\pm2.0}$ |
| GPN | $13.6_{\pm1.7}$ | $9.8_{\pm1.1}$ | $12.1_{\pm4.2}$ | $6.4_{\pm5.6}$ | $23.3_{\pm7.1}$ | $6.7_{\pm1.9}$ |
| FAGCN | $\mathbf{3.6}_{\pm1.1}$ | $22.9_{\pm2.3}$ | $25.9_{\pm7.0}$ | $6.3_{\pm2.1}$ | $8.9_{\pm4.2}$ | $\mathbf{1.7}_{\pm0.6}$ |
| FSGNN | $7.5_{\pm1.8}$ | $33.7_{\pm2.3}$ | $\mathbf{8.4}_{\pm1.9}$ | $7.6_{\pm4.3}$ | $11.9_{\pm2.5}$ | $11.9_{\pm2.4}$ |
| GPRGNN | $6.6_{\pm0.9}$ | $23.0_{\pm2.0}$ | $10.9_{\pm3.9}$ | $8.6_{\pm3.3}$ | $8.8_{\pm5.1}$ | $3.0_{\pm1.7}$ |
| SGCN | $23.3_{\pm1.7}$ | $9.2_{\pm2.6}$ | $8.6_{\pm2.6}$ | $8.5_{\pm1.8}$ | $31.8_{\pm5.6}$ | $10.9_{\pm1.6}$ |
| BGCN | $4.0_{\pm1.6}$ | $3.1_{\pm1.7}$ | $22.5_{\pm7.8}$ | $6.7_{\pm3.2}$ | $5.9_{\pm3.7}$ | $6.8_{\pm1.1}$ |
| Res-GCN | $11.8_{\pm1.0}$ | $37.9_{\pm4.7}$ | $31.2_{\pm14.9}$ | $12.4_{\pm7.9}$ | $12.9_{\pm3.9}$ | $10.2_{\pm0.6}$ |
| Res-GATE | $9.7_{\pm0.5}$ | $43.4_{\pm0.7}$ | $27.4_{\pm13.5}$ | $10.8_{\pm8.6}$ | $12.4_{\pm3.9}$ | $9.7_{\pm0.7}$ |
| Res-GAT-Sep | $10.4_{\pm0.5}$ | $41.0_{\pm0.7}$ | $25.5_{\pm14.4}$ | $29.6_{\pm24.4}$ | $16.0_{\pm2.3}$ | $10.9_{\pm0.7}$ |

*Table 16.* Brier score (↓) of different models on clean data (best and runner-up).

| Model | Roman Empire | Amazon Ratings | Chameleon | Squirrel | CoraML | PubMed |
|---|---|---|---|---|---|---|
| GCN | $0.59_{\pm0.01}$ | $0.72_{\pm0.01}$ | $\mathbf{0.80}_{\pm0.03}$ | $\mathbf{0.80}_{\pm0.01}$ | $0.30_{\pm0.03}$ | $0.24_{\pm0.02}$ |
| GATv2 | $0.19_{\pm0.01}$ | $0.68_{\pm0.01}$ | $0.80_{\pm0.03}$ | $0.85_{\pm0.03}$ | $0.30_{\pm0.03}$ | $0.24_{\pm0.01}$ |
| GPN | $0.87_{\pm0.00}$ | $0.76_{\pm0.00}$ | $0.83_{\pm0.02}$ | $0.87_{\pm0.01}$ | $0.52_{\pm0.05}$ | $0.36_{\pm0.02}$ |
| FAGCN | $0.40_{\pm0.01}$ | $0.68_{\pm0.01}$ | $0.80_{\pm0.03}$ | $0.81_{\pm0.02}$ | $0.38_{\pm0.05}$ | $0.28_{\pm0.01}$ |
| FSGNN | $0.31_{\pm0.01}$ | $0.68_{\pm0.01}$ | $0.82_{\pm0.02}$ | $0.84_{\pm0.01}$ | $0.51_{\pm0.02}$ | $0.40_{\pm0.03}$ |
| GPRGNN | $0.49_{\pm0.00}$ | $0.68_{\pm0.01}$ | $0.84_{\pm0.02}$ | $0.87_{\pm0.02}$ | $0.35_{\pm0.05}$ | $0.25_{\pm0.02}$ |
| SGCN | $0.84_{\pm0.01}$ | $0.75_{\pm0.00}$ | $0.83_{\pm0.02}$ | $0.86_{\pm0.01}$ | $0.58_{\pm0.06}$ | $0.38_{\pm0.02}$ |
| BGCN | $0.78_{\pm0.01}$ | $0.79_{\pm0.01}$ | $0.81_{\pm0.03}$ | $0.87_{\pm0.01}$ | $0.36_{\pm0.04}$ | $0.34_{\pm0.01}$ |
| Res-GCN | $0.28_{\pm0.01}$ | $0.70_{\pm0.01}$ | $\mathbf{0.79}_{\pm0.04}$ | $\mathbf{0.78}_{\pm0.01}$ | $0.29_{\pm0.04}$ | $\mathbf{0.22}_{\pm0.01}$ |
| Res-GATE | $0.18_{\pm0.01}$ | $0.67_{\pm0.01}$ | $0.82_{\pm0.04}$ | $0.86_{\pm0.02}$ | $\mathbf{0.28}_{\pm0.04}$ | $0.21_{\pm0.01}$ |
| Res-GAT-Sep | $\mathbf{0.17}_{\pm0.01}$ | $\mathbf{0.65}_{\pm0.01}$ | $0.84_{\pm0.05}$ | $0.89_{\pm0.02}$ | $0.30_{\pm0.02}$ | $0.22_{\pm0.01}$ |

