# OpenReview forum: "Uncertainty Estimation for Heterophilic Graphs Through the Lens of Information Theory"
_ICML.cc/2025/Conference — ICML 2025 poster_

### Official Review · Reviewer_wjj2 · 2025-02-22

**Overall Recommendation:** 3

**Summary:**

This paper addresses the challenge of estimating epistemic uncertainty on graphs that do not follow the homophily assumption, where neighboring nodes often belong to different classes. The authors provide an information-theoretic analysis of Message Passing Neural Networks (MPNNs) and derive an analog to the data processing inequality that reveals how information can be both lost and (importantly) gained across layers

**Claims And Evidence:**

The paper grounds its approach in an information-theoretic analysis, deriving a novel data processing equality that captures both the loss and gain of information across layers

**Essential References Not Discussed:**

N/A

**Experimental Designs Or Analyses:**

The authors test JLDE on multiple datasets and under various distribution shifts

**Methods And Evaluation Criteria:**

By jointly considering all layer representations, the method intuitively addresses the challenge of information heterogeneity in heterophilic graphs

**Other Comments Or Suggestions:**

N/A

**Other Strengths And Weaknesses:**

Strengths
* The paper brings a new theoretical perspective to uncertainty estimation on graphs by incorporating ideas from information theory
* The derivation of a data processing equality that accounts for both loss and gain of information is novel
* The experiments are extensive, covering multiple datasets, distribution shifts, and network architectures

Weaknesses
* From Definition 4.1 to Proposition 4.2, the authors show that heterophily is advantageous for graph neural networks due to the resulting information gain. Although their proof analyzes this phenomenon using information theory, the conclusion is almost identical to that of [1, 2]
  * [1] Revisiting heterophily for graph neural networks, NeurIPS '22
  * [2] When Do Graph Neural Networks Help with Node Classification? Investigating the Impact of Homophily Principle on Node Distinguishability, NeurIPS '23
* The authors insist that connections to nodes with different semantics (i.e., those providing substantial information) improve overall performance. However, this assumption only holds under the i.i.d. data model, which is neither tractable nor realistic for real-world datasets
* As defined in Eq. 6, the authors propose using the outputs from all layers, a method that has been suggested by many recent studies [3, 4]
  * [3]: Why Do Attributes Propagate in Graph Convolutional Neural Networks?, AAAI '21
  * [4]: Nested graph neural networks, NeurIPS '21

**Questions For Authors:**

Please see the above weaknesses

**Relation To Broader Scientific Literature:**

The paper effectively situates its contributions within the existing literature on uncertainty estimation in both standard neural networks

**Theoretical Claims:**

The paper’s attempt to quantify both information loss and gain is commendable. However, some proofs assume certain conditional independence structures (e.g., ego-graph representations adequately capture all necessary dependencies) which might be too idealized for real-world graph data

---

> ### Author Rebuttal · Authors · 2025-03-31
>
> We thank the reviewer for their thorough review. We want to distinguish our contributions from related work.
>
> ## Performance improvement through Heterophily
> Our theory shows that, from an information perspective, the intermediate layers (i.e. aggregation) in GNNs can provide additional information. This contrasts results for i.i.d. data where information is only lost with network depth. These results explicitly only hold for **interdependent** data (and not the i.i.d. data model). Our analysis exploits that the information a GNN uses to represent a node only depends on its ego graph of a range equal to the GNN depth as can be seen in Figure 3: The latent representation of a node also depends on the features of its $\leq$L-hop neighbors. Therefore, our analysis *explicitly* considers the interdependence and does not require an i.i.d. data model. Theorem 4.1 is valid for *any* MPNN according to the definition of Appendix B.2 without any other assumptions like i.i.d. data.
>
> To clarify this misunderstanding, we change the paper:
> - Changed section title 4.2 from *Information in MPNNs: A Data Processing
> Equality* to *Information for Interdependence: A Data Processing Equalty*
> - We changed the caption of Figure 3 to: *Probabilistic model of how the information is processed in MPNNs given the interdependent data...*
> - The first sentence in 4.2 to: *For problems on graphs, where we deal with interdependent data, the input **X** is twofold as it contains both node features **F** and the graph structure **E**.*
>
>
> ## Discussion of related work
>
> **Heterophily is advantageous:**
> Luan et al. (2022) propose a novel homophily metric through post-aggregation node similarity and develop a novel, powerful GNN: ACM.
> Luan et al. (2023) investigate performance for graphs with various degrees of homophily by studying how node distinguishability is linked to GNN performance and they provide a better performance metric.
>
> While these works are meaningful contributions to heterophilic graphs, we want to stress that our work's conclusion is fundamentally different from 'heterophily is advantageous'. We study how uncertainty can be estimated in heterophilic graphs. Our new theory applies to both homophilic and heterophilic graphs. Our claim is not that certain GNNs benefit from heterophily, but instead, uncertainty in heterophilic graphs can only be quantified when doing joint density estimation. This differs from the related work that uses heterophily to improve accuracy.
>
> **Using Multiple Layers**
> Yang et al. (2021) use the difference between embeddings of the previous layer and the current one to mitigate oversmoothing.
> Zhang and Li (2021) extract local ego-net subgraphs for each node, apply a GNN to those, and finally pool these subgraph representations to obtain node representations.
>
> In contrast, we use an arbitrary MPNN and estimate density from all its latent embeddings to estimate model uncertainty. While other work also considers node representations obtained at different scales, it aims to improve predictive performance by changing the GNN architecture. We apply JLDE post-hoc to an already trained GNN and estimate uncertainty from all its embeddings which we justify both formally and empirically.
>
> However, we agree that it is important to stress the difference to related work and are thankful for these pointers that we discuss in a new paragraph:
>
>
> ```Recent work has focused on refining the inductive biases embedded in GNNs, particularly regarding graph structure, neighborhood similarity, and information aggregation across layers. Luan et al. (2022) revisit GNN performance under heterophily, challenging traditional homophily-based assumptions and proposing adaptive channel mixing to enhance representation learning across diverse neighborhoods. Similarly, Luan et al. (2023) show that GNN performance can not be attributed to homophily alone by introducing the concepts of intra- and inter-class node distinguishability.
>
> Zhang and Li (2021) propose Nested GNNs, which extend beyond rooted subtrees to capture richer local substructures, emphasizing the importance of localized graph context. Complementarily, Yang et al. (2021) avoid over-smoothing by using the difference between a layer's input and output.
>
> In contrast, our work examines the inductive biases of GNNs from an information-theoretic perspective in the context of uncertainty estimation.
> We show the benefits of leveraging information from *all* layers and propose a theoretically grounded uncertainty estimator that achieves state-of-the-art performance. We further demonstrate that heterophilic problems necessitate jointly considering all representations.
> ```
>
> ## Conclusion
> We thank the reviewer for their time and effort to review our work. We hope that we addressed the reviewer's perceived weakness as our theory does not assume i.i.d. data and clarified that we arrive at conclusions that are fundamentally different from existing work and are happy to further discuss this.

---

> > ### Comment · Reviewer_wjj2 · 2025-04-07
> >
> > Thanks for the careful rebuttal. Since the author has addressed most of my concerns, I have updated my score accordingly.

---

### Official Review · Reviewer_n5yN · 2025-03-13

**Overall Recommendation:** 4

**Summary:**

This work addresses the lack of epistemic uncertainty measures on heterophilic graphs by studying the uncertainty of GNNs through information theory. The main contribution is the development of a post-hoc estimate, Joint Latent Density Estimation (JLDE) as a measure of the density of latent embeddings which is useful for out-of-distribution (OOD) node detection for node classification tasks.

**Claims And Evidence:**

Yes.

**Essential References Not Discussed:**

None, to the best of my knowledge.

**Experimental Designs Or Analyses:**

Mostly, ok.
However, one important experimental analysis is missing, which is, studying the sensitivity of the post-hoc uncertainty estimate with respect to varying degrees of homophily ratio. The authors should create synthetic graphs with controlled but varying degrees of homophily ratio and verify if the OOD detection is still effective across various regimes of homophily ratio. See, for instance, [1] (Appendix B) for such an example dataset.

[1] https://arxiv.org/abs/2502.10208

**Methods And Evaluation Criteria:**

Yes.

**Other Comments Or Suggestions:**

1. Line 43, 2nd column => Please elaborate on what JLDE means, since this is the first appearance of this term.
2. Line 135, 2nd column=> I.d.d. => i.i.d
3. Section 4.1 should be shortened to facilitate more discussion on JLDE, in particular Equations 12, and 13 and the associated discussions in the Appendix should be in the main paper.
4. Appendix A, The steps in proofs need to be justified with more supporting statements about known results and references.
5. What is Ens? What do you mean by an “ensemble of 10” in line 346? This needs further clarification.

**Other Strengths And Weaknesses:**

There is a clear gap in the literature where the paper fits. The proposed uncertainty estimate is supported by theoretical motivation, intuition and empirical comparison with recent baselines. No significant weaknesses except those in Comments/Suggestions and Experimental Designs/Analyses.

**Questions For Authors:**

1. Since the focus of the paper is on Epistemic uncertainty, what value does reporting Aleatoric uncertainty in Table 1 add to the paper?
2. In Table 1, what do the boldfaced entries indicate? What is the significance of the grey regions?

**Relation To Broader Scientific Literature:**

The study carries significant interest to the GNN community as well as the uncertainty-estimation community.

**Theoretical Claims:**

Yes. No major issues, but the proof steps need more explanation.

---

> ### Author Rebuttal · Authors · 2025-03-31
>
> We thank the reviewer for their thorough review and suggestions. We want to address their points with the following revisions of our paper:
>
> - **Synthetic Experiments with varying Homophily**: That is a great idea! We investigate how JLDE compares to the state-of-the-art homophilic estimator GEBM on a synthetic graph in [Figure 1](https://figshare.com/s/05c97f1c4314003ce379?file=53331224). The synthetic setting is adapted directly from (Das et al., 2025) with an additional out-of-distribution cluster that the model is not trained on. We observe that while GEBM rapidly deteriorates in performance with increasing heterophily, JLDE is consistently effective in homophilic and heterophilic regimes.
>
> - **Source Code**: There seems to be a misunderstanding as we provide the source code in the supplementary material on OpenReview which is also acknowledged by other reviewers.
>
> - **Other Comments**: 1) We add an explanation of the acronym JLDE to L.43. 2) We fixed this typo, thank you. 3) We agree that the KNN-based realization of JLDE should be discussed more in the main text. In favor of deferring some related work to the Appendix (see response to reviewer is3X), we move Equation 12 to the main text. We also add an explanation and a short discussion about practical concerns like runtime and memory complexity (see feedback to reviewer ejzC). 4) We add verbose explanations for each proof that explain the transition between each line. 5) Ens indeed refers to Ensemble. We clarify this in the main text.
>
> - **Aleatoric Uncertainty**: We report the performance of the associated aleatoric uncertainty estimate for completeness purposes, similar to other work (Stadler et. al., 2021; Fuchsgruber et. al., 2024). This way, we can properly assess the performance of JLDE to *all* other possible uncertainty estimates. We add a clarification that the aleatoric estimate comes from the backbone GNN (see response to reviewer is3X). If the reviewer thinks that these numbers are distracting, we can also defer them completely to the Appendix.
>
> - **Table 1**: Bold-face numbers indicate the best accuracy of the backbone GNN for this dataset and distribution shift. Grey cells highlight values that are associated with JLDE (either accuracy or o.o.d.-detection performance). We update the captions accordingly to indicate this clearly.
>
> We again want to thank the reviewer for their suggestions. We believe that the additional experiments and clarifications underline the merits of our framework and enhance the paper's clarity. If the reviewer has any remaining points that relate to their assessment of our paper we are very happy about additional input.
>
> ## References:
> - Das, Siddhartha Shankar, et al. "SGS-GNN: A Supervised Graph Sparsification method for Graph Neural Networks." arXiv preprint arXiv:2502.10208 (2025).
> - Stadler, Maximilian, et al. "Graph posterior network: Bayesian predictive uncertainty for node classification." Advances in Neural Information Processing Systems 34 (2021): 18033-18048.
> - Fuchsgruber, Dominik, et. al. "Energy-based Epistemic Uncertainty for Graph Neural Networks." arXiv preprint arXiv:2406.04043 (2024).

---

> > ### Comment · Reviewer_n5yN · 2025-04-03
> >
> > Thank you for addressing my questions and concerns. I remain positive about the paper's acceptance.
> >
> > Few questions :
> >
> > 1. Are there practical real-world situations/case studies where epistemic uncertainty estimation for heterophilic graphs is of importance, apart from the OOD detection task?
> >
> > Limitation section:
> >
> > 2.
> > > We propose a design principle for epistemic uncertainty on heterophilic graphs but do not aim to improve aleatoric
> > uncertainty or its calibration
> >
> > Would you elaborate on what are the implications of improving aleatoric uncertainty or its calibration in heterophilic graphs or graphs in general? Also, could there be any unique advantage to estimating aleatoric uncertainty in situations where the graph is noisy?
> >
> > 3.
> > > Also, we focus our evaluation on node classification but the information-theoretic perspective and its implications apply
> > to regression problems and other non-i.i.d. domains that can be cast into graphs as well.
> >
> > I am not clear about what you meant by "other non-i.i.d domains". Would you elaborate on this?
> >
> > ------
> > *Update: I have increased my score after reading other reviews. I encourage the authors to incorporate the discussions regarding additional experiments with varying homophily, differences to other related works, and computational/memory costs into their final version. Good luck!*

---

> > > ### Author Response · Authors · 2025-04-03
> > >
> > > We are happy that we could resolve the questions and concerns of the reviewer. We also want to answer their additional questions:
> > >
> > > 1. Beyond detecting distribution shifts, epistemic uncertainty can be used to abstain from a prediction if the indicated epistemic uncertainty is high. Especially in high-stakes and safety-critical domains like Medicine, this improves the trustworthiness of a model. Disentangling uncertainty is also useful in reinforcement learning tasks (Charpentier et. al., 2022). Furthermore, recent work has shown that accurate epistemic uncertainty leads to an optimal proxy for data acquisition in Graph Active Learning (Fuchsgruber et. al., 2024) which has downstream applications in data-intensive domains. These properties of epistemic uncertainty translate into all application areas in which heterophilic graphs are found, such as fraud detection or recommender systems (Luan et. al., 2024).
> > >
> > > 2. Well-calibrated and accurate aleatoric uncertainty is relevant in domains with ambiguous data. Somewhat simplified, epistemic uncertainty tries to answer the question if I should trust a model's prediction at all. If the answer is yes, model calibration and aleatoric uncertainty are important to make informed decisions that incorporate inherent randomness. Uncertainty estimation for graphs is a recently emerging field and, therefore, the amount of downstream applications in the literature is limited. One example includes molecular data where physical properties can introduce inherent randomness that a model should be able to represent (Wan et. al., 2021; Wollschläger et. al., 2023).
> > >
> > > 3. Our statement entails any domain that can be modeled with a graph even though a structure may not be naturally given. Beyond the aforementioned molecules, also sequential data can be viewed as a line graph. The Roman Empire dataset in fact is constructed as a semantic graph between words. Other examples may include knowledge graphs or relational databases. We mean to express that our framework can be useful in any domain where graphs apply which we will clarify in our paper. We also want to stress that studying specific downstream applications is beyond the more foundational nature of our work.
> > >
> > > We will include parts of this discussion in our introduction to further motivate our work practically. We again want to thank the reviewer for their constructive feedback and we appreciate the time they put into engaging with us in further discussions. If no more concerns or questions remain, and as the reviewer seems to be in favor of acceptance of our paper we would be very happy if they consider raising their score.
> > >
> > > ## References
> > > - Luan, Sitao, et al. "The heterophilic graph learning handbook: Benchmarks, models, theoretical analysis, applications and challenges." arXiv preprint arXiv:2407.09618 (2024).
> > > - Fuchsgruber, Dominik, et al. "Uncertainty for active learning on graphs." arXiv preprint arXiv:2405.01462 (2024).
> > > - Charpentier, Bertrand, et al. "Disentangling epistemic and aleatoric uncertainty in reinforcement learning." arXiv preprint arXiv:2206.01558 (2022).
> > > - Wan, Shunzhou, et. al. "Uncertainty quantification in classical molecular dynamics." Philosophical Transactions of the Royal Society A 379.2197 (2021): 20200082.
> > > - Wollschläger, Tom, et al. "Uncertainty estimation for molecules: Desiderata and methods." International conference on machine learning. PMLR, 2023.

---

### Official Review · Reviewer_ejzC · 2025-03-13

**Overall Recommendation:** 3

**Summary:**

This paper proposes an uncertainty estimation method for heterophilic graphs, primarily through the utilization of multi-layer embeddings. The authors conduct comprehensive analyses, such as examining how information propagates through message passing in neural networks, to validate their claims. They introduce a simple method called Joint Latent Density Estimation (JLDE) that combines outputs from all layers to measure uncertainty. This approach achieves superior results on heterophilic graphs compared to existing methods while maintaining performance on standard graphs. Extensive experiments across multiple datasets demonstrate its effectiveness.

**Claims And Evidence:**

Yes, the claims made in the submission are supported by clear and convincing evidence.

**Essential References Not Discussed:**

No

**Experimental Designs Or Analyses:**

Yes I check the soundness/validity of any experimental designs or analyses. However, the ablation study can be refined.

**Methods And Evaluation Criteria:**

Yes, the proposed methods and evaluation criteria make sense for the problem at hand.

**Other Comments Or Suggestions:**

Please see the above weakness.

**Other Strengths And Weaknesses:**

Strengths:
1. Rigorous theoretical analysis.
2. Logical experiments (heterophilic/homophilic graphs, multiple backbones).
3. Strong results (SOTA on heterophilic graphs).
4. Clear writing.

Weaknesses:

1. Efficiency: No analysis of computational/memory costs from multi-layer embeddings.
2. In ablation gaps:  it is unclear if gains stem from joint embeddings or KNN. Missing comparisons to (a) layer-wise uncertainty averaging, (b) alternative density estimators.

**Questions For Authors:**

NA

**Relation To Broader Scientific Literature:**

1. Heterophilic GNNs: Showing uncertainty estimation requires fundamentally new principles beyond homophily-based methods.
2. Uncertainty in Graphs: Address heterophilic uncertainty, overcoming homophily assumptions in prior work. Introduces post-hoc JLDE, unlike architecture-bound i.i.d. methods.
3. Information Theory: Explaining why deeper MPNN layers gain information in JLDE’s joint layer modeling.

**Theoretical Claims:**

Yes, I check the correctness of any proofs for theoretical claims

---

> ### Author Rebuttal · Authors · 2025-03-31
>
> We thank the reviewer for their thorough review and questions and address them with the following revisions to our paper in addition to the additional experiments shown [here](https://figshare.com/s/05c97f1c4314003ce379?file=53331224).
>
> 1. **Efficiency**: The computational cost of KNN-based JLDE is governed by KNN, which can be implemented in $\mathcal{O}(n_{train} * d_{hidden} + n_{train} * k)$. For smaller training sets that are typical in transductive node classification, this is negligible but can become prohibitive for large training sets. In this case, JLDE can be realized with more efficient density estimators. We opt for KNN due to its simplicity and applicability to estimating a high dimensional density from little data. However, in general, any density estimator can be used (see below). We measure the runtime of KNN-based JLDE in Table 2 (link above). For all datasets with small training sets (all but Amazon Ratings), the cost of JLDE is comparable to competitors. Using multiple layers effectively increases $d_{hidden}$ by a factor of $L$ in terms of runtime complexity. This only incurs a small additional cost (see comparison to 1-layer KNN-based JLDE in Table 2 (link above). The memory complexity is also determined by the density estimator (KNN) and amounts to keeping the embeddings of the training set in the memory, $\mathcal{O}(n_{train} * d_{hidden})$. Since by default, these activations are kept in memory for a forward pass unless explicitly deallocated, there is no overhead in practice. We add both this explanation and the runtime comparison to the Appendix of our paper.
>
> 2. **Ablations**: (a) layer-wise uncertainty: Since the performance of uncertainty estimates is unaffected by monotonic transformations, uncertainty averaging effectively amounts to the same as adding the individual uncertainty estimated from each layer. This variant is already ablated in the paper in Figures 4 and 6-8 as "All (cat)". We will clarify this in the revised paper. (b) Estimating high-dimensional density from little data is challenging which is why we chose KNN to realize JLDE. We also provide ablations using i) an RBF-based Kernel-Density Estimator (KDE) and a Mixture of Gaussians with diagonal covariance (MoG) in Table 1. KDE performs similarly to JLDE but MoG falls short in some settings because of its limited expressiveness. We add this ablation to the revised paper. We also want to highlight that the core message of our paper is not that KNN should be used for uncertainty quantification. Instead, we argue for the merits of estimating the data density in the joint latent node embedding space with *some* suitable density estimator.
>
> We again want to thank the reviewer for suggesting these experiments. We believe they enable a well-rounded assessment of JLDE. Should any other concerns or questions remain, we are happy to discuss them with the reviewer!

---

> > ### Comment · Reviewer_ejzC · 2025-04-03
> >
> > Thanks authors for response. Since my concerns have been addressed, I will keep the positive scoring for the work.

---

> > > ### Author Response · Authors · 2025-04-03
> > >
> > > We again want to thank the reviewer for their time and their constructive feedback that helped us to provide further contextualization for our method and additional ablations. Since there are no remaining concerns and the reviewer in general seems to be positive about the paper's contributions and its acceptance we would be very grateful if they consider raising their score beyond a borderline score if they think it is adequate.

---

### Official Review · Reviewer_is3X · 2025-03-20

**Overall Recommendation:** 3

**Summary:**

The paper explores uncertainty quantification (specifically epistemic uncertainty) in graphs without homophily -- notably previous works in this direction assumed homophily as another source of information about the ground truth probability revealing information about similarity in the conditional label distribution; this assumption is invalid in heterophily. Here the authors adopting an information-theoretic perspective. While in an i.i.d. setting, mutual information decreases by increasing the model's depth, the graph structure introduces potential information gain through message passing (extending receptive field to other nodes). This is well illustrated in the probabilistic model shown in Fig. 3.

Additionally, the authors propose a KNN-based density estimation over all latent variables concatenated (i.e., embeddings from all layers) as a measure of epistemic uncertainty. They also conduct an extensive empirical study to support their approach.

**Claims And Evidence:**

In general yes! The information theoretic view of the problem helps to provide a clear intuition. However, maybe due to my limited understanding of some parts of the paper, I think the theory can be summarized further while preserving the concreteness of the arguments.

The chain of arguments which I understood is: (1) from Tishby & Zaslavsky (2015), and data processing inequality, we know that on i.i.d. data, model looses information across layers (2) despite i.i.d. data, in graphs we can gain information through message passing as we visit new node, and the information gain is provably non-negative. (3) To quantify epistemic uncertainty, we need to process all latent variable (4) we can use KNN density estimation over all latent values.

If my understanding is correct, there are several parts of the paper, that although informative, they are not directly connected to the method. For example the information loss which is quantified in Def 4.1, and Prop 4.2.

Also: I can not connect the practical method and the theoretical discussion. For that please see "Other Strengths And Weaknesses".

**Essential References Not Discussed:**

I think the literature review was comprehensive and covering. I did not find any problems in that area. However I would prefer to see a shorted "related works" section in the manuscript and the current version in the appendix as it shifts from the main storyline of the paper.

**Ethical Review Concerns:**

There are no concerns.

**Experimental Designs Or Analyses:**

Please see "Methods And Evaluation Criteria". There I mentioned one experiment I expect from the authors to show the reliability of the epistemic uncertainty estimates.

**Methods And Evaluation Criteria:**

The evaluation criteria covers some of the mostly used methods in UQ literature; however, I have the following question:

What is the method to quantify aleatoric uncertainty? I can not understand aleatoric uncertainty results in the tables for o.o.d. detection. I also can not understand the results in Table 14, and 15? The authors argue that the calibration is over models, JLDE does not contribute to the ECE, or Brier score. However, one easy way to examine the reliability of the method is to draw a plot to compare the epistemic uncertainty of the nodes compare to the accuracy. Some plot like ECE instead of confidence, the x-axis shows the average epistemic uncertainty.

Also another question:
Does this method easily extend to graphs with homophily? I assume there is no bottleneck for the method to extend to homophily graphs, however I see that the method is outperformed by a significant margin (> 20% in Table 6 for CoraML) compared to SOTA. What is that? What is the intuitive reason behind it?

**Other Comments Or Suggestions:**

Please see "Other Strengths And Weaknesses".

**Other Strengths And Weaknesses:**

### Strength

1. I believe this area (epistemic uncertainty over heterophilic graphs) is both interesting and untapped. Also I found the approach of the authors novel and interesting; they capture the amount of information collectable the receptive field.
2. The results on heterophilic graphs look promising. Also the extensive empirical evaluations makes the paper stronger.
3. The theoretical insights look correct and complete.

### Weaknesses

1. **Context of the paper is inconsistent.** To my perspective the paper looks like two possibly orthogonal directions concatenated. Up to line 317-left is a theoretical study showing that the information (for any general estimator) can possibly gain over increasing layers. Then suddenly it shifts to introducing an uncertainty estimator that is a KNN-density estimator over the concatenation of the latent values. Despite the discussion on the information theoretic point of view, there is no clear chain of arguments leading to the choice of the uncertainty estimator.

    In other words, one can easily follow another chain of argument leading to the same estimator. For instance: since we can not count on homophily as a supplementary modal to encode similarity in the ground truth $p(y\mid x)$, we directly use the latent representation as the input of our uncertainty estimation algorithm. We concatenate the embeddings of all layers since each hierarchically they encode information about $k$-hop neighborhood of the node.

    To me both the example argument stated above and the information theoretic perspective are equally plausible for this estimator.

2. **Writing can be improved.** Several comments on writing: (1) One can not find any footprint of the method (KNN) from the introduction. Also there is no pseudocode that helps to deliver a quick, and at the same time concrete understanding the final practical method. (2) Although a basic understanding of the information theory is necessary for anyone reading ML papers, still I would prefer to see a better intuitive description over notations like conditional mutual information, etc. Specifically Eq. 3, and 6 need more intuitive explanation. (3)The term “JLDE”  (joint latent density estimation) should be mentioned the first time it is used which is line 42-right. (4) The long discussion on the models proposed for heterophilic graphs (related works section) detaches the reader from the main story of the paper. The proposed method is agnostic to the model’s structure, therefore I would suggest a succinct summary in the manuscript and a more detailed related work perhaps in appendix.

**Questions For Authors:**

While addressed in other parts, here I highlight my most important questions:

1. Does this method easily extend to graphs with homophily? I assume there is no bottleneck for the method to extend to homophily graphs, however I see that the method is outperformed by a significant margin (> 20% in Table 6 for CoraML) compared to SOTA. What is that? What is the intuitive reason behind it?
2. **About Eq. 3.** (1) is it even realistic to have this estimator? (2) is it what we need for prediction? For (1) I am unsure if the search space isn’t too restrictive. In other words, maybe even finding one example in that search space is quite difficult. Because the authors are searching over a space of functions such that inputs all possible information from $X$. Which means that a marginally (weighted average over the probability of the input), it should have the same correlation with the invisible ground truth label (or here generative) distribution. I believe this is even harder than having a perfectly calibrated classifier. For (2) if we have such a classifier, even if it includes additional unnecessary information about X from the perspective of uncertainty quantification still this is a very good classifier. Even I can argue that including more information about the generative distribution of X helps generalization.
    1. I assume the questions I asked are maybe due to not completely understanding the goal of Eq. 3. Maybe if authors provide an intuitive explanation, it can help.
    2. Maybe I would be more convinced if the search space was not restricted but the overall optimization was written in form of a min-max problem which maximizes the information about the labels and minimizes the unnecessary information about X.
3. What is the method to quantify aleatoric uncertainty? I can not understand aleatoric uncertainty results in the tables for o.o.d. detection. I also can not understand the results in Table 14, and 15? The authors argue that the calibration is over models, JLDE does not contribute to the ECE, or Brier score. However, one easy way to examine the reliability of the method is to draw a plot to compare the epistemic uncertainty of the nodes compare to the accuracy. Some plot like ECE instead of confidence, the x-axis shows the average epistemic uncertainty.

**Relation To Broader Scientific Literature:**

The paper tends to fill the gap in uncertainty quantification for graphs. Previous works in this area assume the graph has homophily -- the two endpoints of an edge are likely to have the same label. This is potentially used as a source of information helping to signal for higher uncertainty when the prediction is not following this rule.

**Theoretical Claims:**

There are no special theorems in the paper requiring complicated and multi-step proofs. Many of the arguments are directly supported by one line proofs often taken form the information theory literature.

Additionally these are my other problems / questions:

**About Eq. 3.** (1) is it even realistic to have this estimator? (2) is it what we need for prediction? For (1) I am unsure if the search space isn’t too restrictive. In other words, maybe even finding one example in that search space is quite difficult. Because the authors are searching over a space of functions such that inputs all possible information from $X$. Which means that a marginally (weighted average over the probability of the input), it should have the same correlation with the invisible ground truth label (or here generative) distribution. I believe this is even harder than having a perfectly calibrated classifier. For (2) if we have such a classifier, even if it includes additional unnecessary information about X from the perspective of uncertainty quantification still this is a very good classifier. Even I can argue that including more information about the generative distribution of X helps generalization.
    1. I assume the questions I asked are maybe due to not completely understanding the goal of Eq. 3. Maybe if authors provide an intuitive explanation, it can help.
    2. Maybe I would be more convinced if the search space was not restricted but the overall optimization was written in form of a min-max problem which maximizes the information about the labels and minimizes the unnecessary information about X.

---

> ### Author Rebuttal · Authors · 2025-03-31
>
> We thank the reviewer for their in-depth feedback and questions. We are happy they find our paper to provide novel insights into an interesting and untapped research area. We provide additional material [here](https://figshare.com/s/05c97f1c4314003ce379?file=53331224).
>
> ## Connection between Theory and Method
>
> **Context of the Paper**: JLDE is intuitively motivated since, in heterophilic graphs, a node's neighbors are semantically different from itself which leads to diverse latent embeddings. While we agree that this method could have been proposed without formal justification, we believe that our analysis provides a theoretical basis that not only supports the intuition behind JLDE's effectiveness but is also valuable for further advancements in uncertainty quantification under heterophily. We utilize it to justify that for heterophilic graphs, each embedding provides different information. To that end, we develop an information-based framework for MPNNs that describes how the information in latent embeddings relates to each other (Theorem 4.1). The key insight that links this analysis to JLDE is discussed in L.295: The information gain is governed by the information that the k+1-th hop neighbors add that is not already contained in the $\leq$k-hop neighbors. For heterophilic data, the semantic differences between adjacent nodes induce information gain while for homophilic graphs the *additional* information diminishes. Estimating a node's data density to quantify uncertainty therefore *must* rely on all hidden embeddings. While Theorem 4.1 marks a significant contribution by itself it does not explicitly target uncertainty quantification. We view this not as a limitation. It fits well into the paper as it justifies our main claim: Uncertainty quantification in heterophilic graphs greatly benefits from considering all hidden node representations as it utilizes the information gain term. We clarify this by revising L.161 to explain that we propose JLDE as an intuitive uncertainty quantification framework that we formally justify from an information-theoretic angle. Def 4.1 and 4.2 make Theorem 4.1 more digestible by separately discussing its components.
>
> ## Uncertainty Calibration Experiment
>
> That is a great idea! We conducted this experiment (Figure 1, link above). For 5 of 6 datasets, JLDE's uncertainty correlates well with accuracy.
>
> ## Writing
>
> 1. We mention the KNN-based realization of JLDE in L.44 and provide pseudocode in Algorithm 1 (link above).
> 2. We elaborate on Equation 3 (see below) and Equation 6 and an intuition for MI in L.83.
> 3. We introduce the acronym JLDE in L.42.
> 4. We defer the first paragraph of Section 3 to the Appendix to shorten Related Work.
>
> **Conciseness of Theory**: We agree that Prop. 4.2 does not directly relate to uncertainty quantification and we move it to the Appendix.
>
> ## Questions
> **Aleatoric Uncertainty** is quantified as $1 - max_c p_c$ (L.838) from the backbone MPNN and is the same for all post-hoc estimators. Like related work, we report it for completeness. We move its definition and this explanation to L.348.
>
> The **ECE and Brier** metrics (Table 14, 15) depend only on the classifier. Therefore, we do not explicitly restate it for each post-hoc uncertainty estimator and report them only for completeness.
>
> **Equation 3** is directly taken from the seminal work of Tishby et. al. that established an information-theoretic perspective on NNs. It is referenced consistently throughout the literature. In practice, the optimal estimator of Eq. 3 is not attainable and methods instead optimize a trade-off between compression and predictive capabilities similar to the reviewer's min-max formulation. For example, the "Deep Variational Bottleneck", (Alemi et. al., 2016) proposes a varational optimization problem. Compression is necessary because making predictions from inputs that contain semantically irrelevant information is challenging (and hence, representation learning is used). We see how this can be difficult to understand without knowledge of the prior work on information theory and NNs and add an explanation to Section 4.1.
>
> **Homophilic Graphs**: Estimators like GPN, GEBM, or Safe explicitly exploit the homophily through homophilic graph diffusion to improve uncertainty. While JLDE is particularly useful for heterophily, it does not make structural assumptions and applies to homophilic and heterophilic graphs. As a drawback, it can not exploit the homophily in graphs directly and falls short of methods that do. Nonetheless, JLDE performs competitively to other post-hoc methods that do not use homophilic graph diffusion like EBMs or Ensembles. We believe that future work can optimize JLDE for homophilic graphs.
>
> We again thank the reviewer for their thoughtful and in-depth feedback which helped us improve our manuscript and hope we adequately addressed their points. We are happy to discuss further open questions.

---

> > ### Comment · Reviewer_is3X · 2025-04-02
> >
> > I thank the authors for replying to my comments. Yes indeed the authors have already pointed to an interesting problem with an insightful approach.
> >
> > Here are my concerns:
> >
> > - About the **context of the paper** part unfortunately I am still not convinced. The definition of homophily and heterophily (also as noted in the paper) is based on label similarity and not the feature similarlity. Therefore (1) one can simply design a dataset with high homophily and diverse feature space on endpoints of an edge. Simply assume an image dataset, and a noisy similarity kernel that perfectly connect similar labels 90% of the time, so for instance majority of images with class tree are connected to each other while the feature space includes information about the light, average color, etc of the image. On the oposite side one can easily create a heteophily graph where the edge is sampled via a kernel on feature similarity. Therefore the statement claiming “we should see all levels of embedding” is intuitively right but to me it does not pass through features when homophily is in the space of labels. Therefore I can not follow the sentence “while for homophilic graphs the *additional* information diminishes”.
> > - Building on the previous comment, shouldn't we expect that JDLE reconstructs the information about homophily (that is used as a side modal of similarity by the other graph uncertainty quantification approaches)?
> > - Thank you for drawing the ECE plot, and even the box-based reliability chart is even more illustrative (while unfortunately many other works don’t do that. Question is why no Cora and PubMed there is no visible correlation? Since those are homophily graphs, I would still expect a correlation. My intuitive explanation is that fixed bin interval causes a low support inside the bin and therefore on low support many statistics would be volatile. Can you please check the number of nodes falling inside each bin? Also can you try plotting the same plot but this time instead of fixed bin interval, using fixed number of points inside each bin?
> > - Looking at the algorithm: Does the norm-2 distance suffer from the high dimensionality caused by combining all the embeddings? Do you use l-2 distance always? Can you use a model to learn the low-embedding distances, or maybe even a PCA/LDA?
> > - Thank you for explaining Eq. 3, I would also suggest having somewhat similar description in the paper around the equation to increase readability.
> > - About the empirical comparison with GPN, etc on homophily graph I admit that the authors are right. Those methods exploit the homophily as a side modality indicating the similarity in graph. Therefore, on those datasets, surely JLDE should be compared with structure agnostic methods. Interestingly I see somehow similar behaviour compared to ensemble models, while presumably your method requires less runtime.
> >
> > Surely I like the paper, but still the problems I mentioned (l-2 norm on high dimensionality, difference of the framework's claim for heterophily and homophily graph, and the reliability plot in fig.2 of the attachment) are fundamental. To me the fact that JDLE and ensemble models tie in homophily graphs, while ensembles are expensive to compute, is interesting. Therefore if the problems I pointed are solved surely I'll read the paper for another pass and consider to increase my score.

---

> > > ### Author Response · Authors · 2025-04-02
> > >
> > > ## Context
> > > The reviewer raises an important point regarding the definition of homophily that needs disambiguation. Recent work (e.g., Luan et al., 2022, see reviewer wjj2) argues for concepts beyond label similarity incorporating features. Our work (previously only implicitly) adopts an information-theoretic perspective that aligns with these newer definitions.
> > >
> > > Traditionally, homophily is defined by whether the labels at edge endpoints agree. However, we define homophily through the semantic information of *features that matter for the label*. This is quantified by the mutual information between a node’s label and features at different hops.
> > >
> > > We consider a graph to be heterophilic when a node's neighbors provide additional, semantically distinct information about the node's label compared to the node’s own features. Formally, the homophily of the i-hop neighbors to the (i-1)-hop ego graph is:
> > >
> > > $h_v^{(i)} = I(G^{(i)}_v; G^{(0:i-1)}_v) - I(G^{(i)}_v; G^{(0:i-1)}_v | Y_v)$
> > >
> > > This definition aligns with a post-aggregation metric of Luan et. al. It is composed of
> > > - the *entire* semantic redundancy in the i-th order and up-to-(i-1)-th order ego graph
> > > - and subtracts all parts that are not relevant to the label (i.e. task-irrelevant information in the features).
> > >
> > > This definition also nicely ties homophily to our analysis regarding the *realizable information gain* of Def 4.2:
> > >
> > > $I(Y_v; G^{(i)}_v | G^{(0:i-1)}_v) = I(Y_v; G^{(i)}_v) - h_v^{(i)}$
> > >
> > > It shows why heterophily leads to larger information gain and motivates JLDE. In homophilic graphs, the largely redundant semantic information about the label in a node's neighbors reduces the potential information gain.
> > >
> > > This definition of homophily also clarifies the role of uninformative features like brightness, which do not contribute semantic information about the label and to which our notion of homophily is invariant.
> > >
> > > To relate this to the reviewer’s examples:
> > >
> > > - Example 1 (i.i.d. classification with similar images): While the graph shows label homophily, the neighbors provide no new information beyond a node's own features. Hence, the graph is homophilic under our definition.
> > > - Example 2 (edges based on uninformative features): The information overlap between adjacent node features does not influence our homophily and the graph is heterophilic. However, the neighbors provide no information about the label
> > >  in the first place, i.e. $I(Y_v; G_v^{(i)}) = 0$. JLDE can not exploit even a heterophilic structure *if the structure itself is semantically irrelevant*. The benefits of JDLE are not unrealized because of homophily but because neighbors are uninformative overall.
> > > - An example of a semantically heterophilic graph is the word graph of Roman Empire: An adverb is qualified by being connected to a verb. The adjacent 'verb'-information is semantically different compared to the 'adverb'-information of the node itself and, thus, provides information about the label that can be used by JLDE.
> > >
> > > The empirical success of JLDE confirms that this notion of homophily applies well to standard benchmarks.
> > >
> > > We also notice that the definition in L.083f. is misleading in that regard as we do not define homophily only in the label space. We will instead formally introduce our information-theoretic notion of homophily. Thank you for spotting this -- it definitely helps the consistency of the paper and makes our assumptions more clear.
> > >
> > > ## Other
> > > We update our [rebuttal pdf](https://figshare.com/s/05c97f1c4314003ce379?file=53331224).
> > >
> > > - **Can JLDE reconstruct homophily**: In general, yes! However, it is expected that methods that are hard-coded to do so will do that better than an agnostic method. JLDE recovers homophily as well as Ensembles or EBMs -- both of which are also not explicitly designed to use homophily.
> > > - **ECE plot:** JLDE is well calibrated for Cora and PubMed, but less so for Chamaleon and Squirrel. This likely relates to the worse performance of the backbone for those datasets which deteriorates with the embedding quality JLDE is based on. We provide the bin size for Figure 2 in Figure 3 and show the same plot for bins of the same size in Figure 4. The trend is more or less the same and reveals a potential drawback of JLDE as it relies on sufficiently useful embeddings.
> > > - **Dim. Reduction:** You are right -- in fact, we use PCA (L.851) before we run JLDE. We updated Algorithm 1. In our experiments, we always use the L2-norm, but as per Table 1, different density works as well. Our main claim regards the merits of Joint Latent Density, not how to best estimate this density.
> > > - **Incorporating feedback into manuscript:** Yes, we already added an extensive description that follows our rebuttal. We believe that the discussions greatly enhance the clarity of our work which is why we incorporate all the suggestions into the manuscript.
> > >
> > > We again thank the reviewer for these helpful discussions and the time & effort to give us the opportunity for further clarification.

---

### Decision · Program_Chairs · 2025-05-01

**Decision:**

Accept (poster)

**Comment:**

This paper studies uncertainty estimation on heterophilic graphs. The key idea is to analyze MPNNs from information-theoretic perspective and to realize that embeddings provide information about data distribution. Then uncertainty quantification is done through density estimation which leads to the proposed method named JDLE. The main concerns from the reviewers were mainly about similar observations to existing works that analyze GNNs on heterophilic graphs, extensions to homophilic graphs, additional analysis/experiments on JDLE, as well as experimental details regarding aleatoric uncertainty quantification (as the main focus of this paper is on epistemic uncertainty). I appreciate the very engaging discussions between authors and reviewers. The authors have addressed these concerns, and all reviewers remain positive about this paper. Thus, I recommend acceptance.